# Formation of a Composite Albian–Eocene Orogenic Wedge in the Inner Western Carpathians: P–T Estimates and $^{40}$Ar/$^{39}$Ar Geochronology from Structural Units

Marián Putiš [1,*], Ondrej Nemec [1], Martin Danišík [2], Fred Jourdan [2,3], Ján Soták [4], Čestmír Tomek [4], Peter Ružička [1] and Alexandra Molnárová [1]

[1] Department of Mineralogy, Petrology and Economic Geology, Faculty of Natural Sciences, Comenius University, 842 15 Bratislava, Slovakia; ondrej.nemec@uniba.sk (O.N.); peter.ruzicka@uniba.sk (P.R.); molnarova140@uniba.sk (A.M.)

[2] John de Laeter Centre, Curtin University, Perth, WA 6102, Australia; M.Danisik@curtin.edu.au (M.D.); F.Jourdan@exchange.curtin.edu.au (F.J.)

[3] School of Earth and Planetary Sciences, Curtin University, Perth, WA 6102, Australia

[4] Earth Science Institute, Slovak Academy of Sciences, 840 05 Bratislava, Slovakia; sotak@savbb.sk (J.S.); cestmir.tomek@savba.sk (Č.T.)

[*] Correspondence: marian.putis@uniba.sk

**Abstract:** The composite Albian–Eocene orogenic wedge of the northern part of the Inner Western Carpathians (IWC) comprises the European Variscan basement with the Upper Carboniferous–Triassic cover and the Jurassic to Upper Cretaceous sedimentary successions of a large oceanic–continental Atlantic (Alpine) Tethys basin system. This paper presents an updated evolutionary model for principal structural units of the orogenic wedge (i.e., Fatricum, Tatricum and Infratricum) based on new and published white mica $^{40}$Ar/$^{39}$Ar geochronology and P–T estimates by Perple_X modeling and geothermobarometry. The north-directed Cretaceous collision led to closure of the Jurassic–Early Cretaceous basins, and incorporation of their sedimentary infill and a thinned basement into the Albian–Cenomanian/Turonian accretionary wedge. During this compressional D1 stage, the subautochthonous Fatric structural units, including the present-day higher Infratatric nappes, achieved the metamorphic conditions of ca. 250–400 °C and 400–700 MPa. The collapse of the Albian–Cenomanian/Turonian wedge and contemporary southward Penninic oceanic subduction enhanced the extensional exhumation of the low-grade metamorphosed structural complexes (D2 stage) and the opening of a fore-arc basin. This basin hemipelagic Coniacian–Campanian Couches-Rouges type marls (C.R.) spread from the northern Tatric edge, throughout the Infratatric Belice Basin, up to the peri-Pieniny Klippen Belt Kysuca Basin, thus tracing the south-Penninic subduction. The ceasing subduction switched to the compressional regime recorded in the trench-like Belice "flysch" trough formation and the lower anchi-metamorphism of the C.R. at ca. 75–65 Ma (D3 stage). The Belice trough closure was followed by the thrusting of the exhumed low-grade metamorphosed higher Infratatric complexes and the anchi-metamorphosed C.R. over the frontal unmetamorphosed to lowest anchi-metamorphosed Upper Campanian–Maastrichtian "flysch" sediments at ca. 65–50 Ma (D4 stage). Phengite from the Infratatric marble sample SRB-1 and meta-marl sample HC-12 produced apparent $^{40}$Ar/$^{39}$Ar step ages clustered around 90 Ma. A mixture interpretation of this age is consistent with the presence of an older metamorphic Ph1 related to the burial (D1) within the Albian–Cenomanian/Turonian accretionary wedge. On the contrary, a younger Ph2 is closely related to the late- to post-Campanian (D3) thrust fault formation over the C.R. Celadonite-enriched muscovite from the subautochthonous Fatric Zobor Nappe meta-quartzite sample ZI-3 yielded a mini-plateau age of 62.21 ± 0.31 Ma which coincides with the closing of the Infratatric foreland Belice "flysch" trough, the accretion of the Infratatricum to the Tatricum, and the formation of the rear subautochthonous Fatricum bivergent structure in the Eocene orogenic wedge.

**Keywords:** Inner Western Carpathians; Albian–Eocene wedge; tectono-thermal overprinting; $^{40}$Ar/$^{39}$Ar geochronology; evolutionary model

## 1. Introduction

Subduction–collision processes and the formation of accretionary wedges [1] are usually related to the closure of oceanic basins. The Inner Western Carpathians (Slovakia, Central Europe) provide an example of a complex accretionary wedge, which in addition to oceanic basins, incorporated a riftogenic continental basin system.

The Inner Western Carpathians (IWC) comprise two principal orogenic wedges: (i) the southern, so-called Meliatic–Gemeric–Veporic wedge of Late Jurassic–Early Cretaceous age, derived from the Neotethys Ocean and its northern passive continental margin, and (ii) the northern, so called Fatric–Tatric–Infratatric wedge derived from the Atlantic Tethys Ocean (Alpine Tethys, sensu [2]) southern (active) continental domain; it formed in two stages during Albian–Cenomanian/Turonian and Eocene times [3,4] and references therein.

A polystage evolution of the northern (Fatric–Tatric–Infratatric, or FTI) wedge is not adequately explained and continues to be a matter of discussion. For example, some authors argued that in the pre-collision stage, a continuous cover succession without any tectono-metamorphic events was formed within the Infratatric Unit (Infratatricum) of the FTI wedge during Triassic to Late Cretaceous times ([5] and references therein). In contrast, some authors argued that metamorphosed olistolithic to clastogenic material, including the Lower Cretaceous slates found in the Upper Cretaceous "flysch" of the Infratatricum, provide evidence for a tectono-metamorphic event prior to the Late Cretaceous [3,6,7]. This tectono-thermal event was constrained to ca. 100 Ma based on white mica $^{40}$Ar/$^{39}$Ar geochronology applied to the Infratatricum [3,7].

To address this controversy, we present an updated evolutionary model for the northern FTI wedge. The model is based on the tectono-thermal events recognized in the reappraised basement–cover structural complexes by published and new P–T estimates and white mica $^{40}$Ar/$^{39}$Ar geochronology from the major shear zones. We focus on the subautochthonous Fatricum (or an inferred tectonically reduced basement–cover structural complexes of the Fatricum) and the Infratatric basement–cover structural complexes which show much more distinct tectono-thermal overprint than the central Tatric sliver.

*Review of the Western Carpathians Tectonic Units*

The principal tectonic units of the Western Carpathians (Figure 1) are briefly characterized from north to south as follows:

The Pieniny Klippen Belt (PKB) is a Cenozoic structure separating the Outer (OWC) and Inner Western Carpathians (IWC). The OWC are limited by the Neogene Carpathian Foredeep deposits in the North which are overridden by the Cretaceous to Palaeogene flysch rootless nappes of the inner Magura and outer Silesian units. The Stramberk Jurassic–Cretaceous klippen may be a prolongation of the Helveticum, and the Magura Unit is equivalent to the Rhenodanubian Flysch Zone of the northern Penninicum [8].

The inferred southern Penninicum pendant is the Váhic (Pieninic–Váhic) Ocean between the PKB and the IWC [3,5,9]. A metamorphic core complex assigned to the Penninic Iňačovce(–Krichevo) Unit was identified by deep boreholes in the Transcarpathian Basin floor (eastern Slovakia) from deep borehole materials [10]. This complex contains Jurassic(?)–Cretaceous Bündnerschiefer type meta-sediments (variegated phyllites with marbles), black phyllitic schists, greenschists, chloritoid schists and Paleogene flysch overlain by serpentinites. The youngest phyllitic formation contains Nummulites sp. bearing meta-sandstones of the Eocene age. Importantly, ophiolitic meta-basites, which are typical members of the Penninicum cf. [11] were also identified [10]. The complex most likely shows late Eocene MP/LT metamorphism (l.c.).

The IWC northern wedge is composed of the Fatric–Tatric–Infratatric–peri-PKB basement–cover structural complexes ([3] and references therein). The IWC Tatricum, Veporicum and Gemericum are equivalent units to the Lower, Middle and Upper Austroalpine units, respectively, in the Eastern Alps [8,12,13].

An intra-continental branch of the Atlantic Tethys (or Alpine Tethys after [2]) called the Fatricum [14] was located between the Tatricum and Veporicum. The Fatric nappes are presumed to have formed from a Jurassic to Lower Cretaceous sedimentary infill of the Fatric (Zliechov) Basin which originated on an extensionally thinned basement (~Variscan crystalline basement with a Permian–Triassic pre-rift cover) in a transitional area of the present-day Tatricum and Veporicum. The Fatric (Zliechov) Basin was closing from the late Albian to Cenomanian at ca. 100–95 Ma and the Fatric nappes were thrust onto the Tatricum during the Turonian to Santonian ([5,8,13–17] and references therein) and overlain by the Hronic nappes [8].

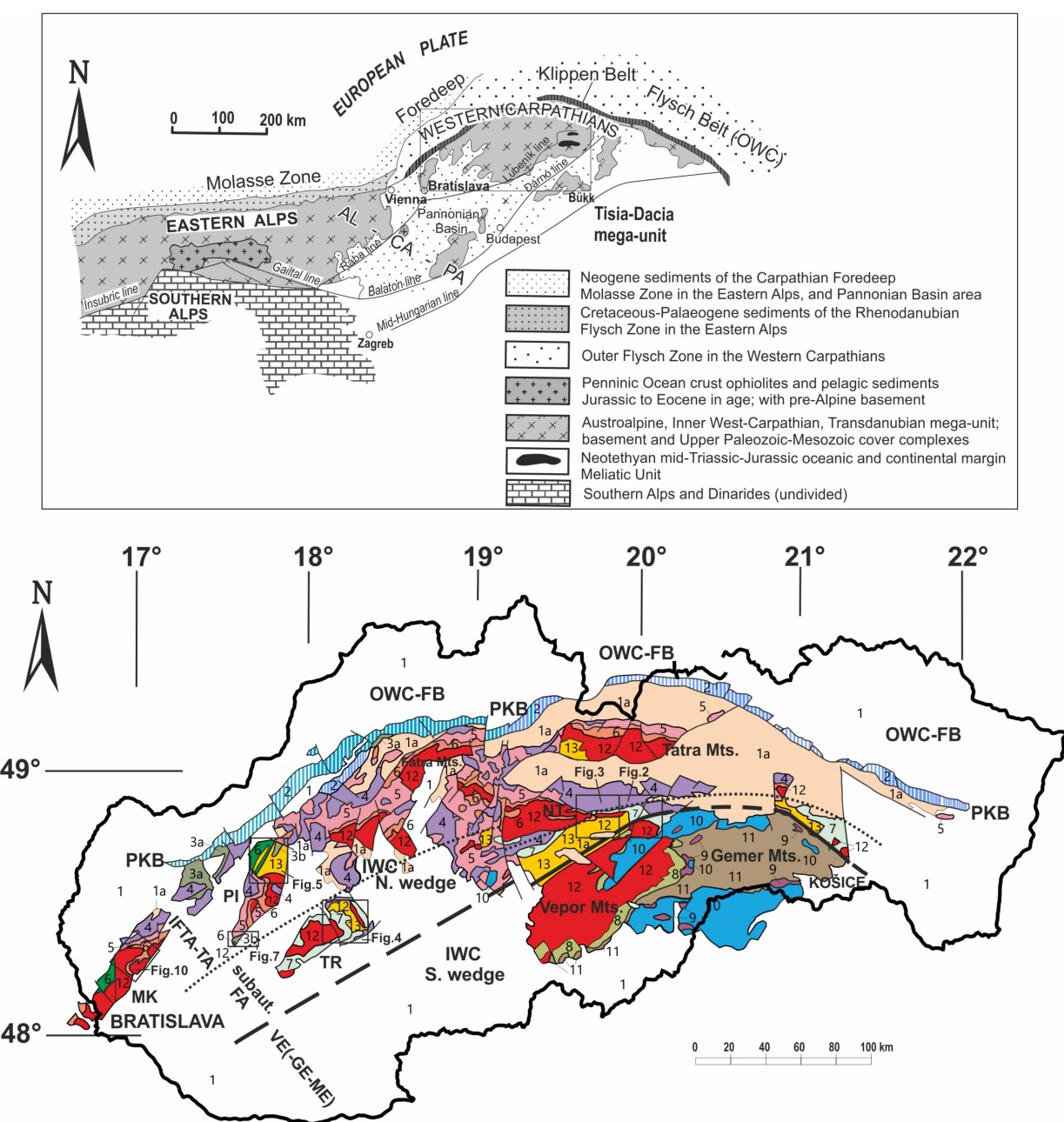

**Figure 1.** Position of the Western Carpathians in Cenozoic ALCAPA microplate (upper figure). Schematic tectonic map of the Inner Western Carpathians (IWC) on the lower figure (modified from [4,18]). Added thick dashed line indicates Late Cretaceous tectonic boundary between the southern and northern wedges, and the dotted line separates the Tatricum from subautochthonous Fatricum. 1—Quaternary and Cenozoic deposits; Outer Western Carpathians Flysch Belt (OWC-FB); 1a—Paleogene deposits of the IWC; 2—Pieniny Klippen Belt (PKB); 3a—Upper Cretaceous to Eocene Gosau-type sediments; 3b—Upper Cretaceous Infratatric Succession; 4—Hronic nappes; 5—Fatric nappes; 6—Infratatric/Penninic successions (green) and the Tatric cover; 7—subautochthonous Fatric cover; 8—Veporic cover; 9—Meliatic nappes; 10—Silicic nappes (including Turnaicum); 11—Gemeric Early Paleozoic basement (Variscan Lower Unit) and cover; 12—Variscan Upper Unit; and 13—Variscan Middle Unit. NT–Nízke Tatry Mts. (with frames of Figures 2 and 3), TR–Tribeč Mts. (frame of Figure 4), PI–Považský Inovec Mts. (frame of Figure 5 from northern Selec Block and the northern margin of the central Tatric Bojná Block, frame of Figure 7 from the southern Hlohovec Block); MK–Malé Karpaty Mts. (frame of Figure 10), IFTA–TA–Infratatric–Tatric zone, subaut. FA–subautochthonous Fatric zone, VE(-GE-ME)–Veporic(-Gemeric-Meliatic) zone.

The Late Cretaceous Čertovica thrust-fault separates the Tatricum and Fatricum (including the so-called north-Veporic area), the structural complexes of which show different

grade of the Alpine metamorphic overprint (e.g., [19,20]). The central Tatric sliver consists of the relatively thicker basement rocks with the meso-Variscan granitic plutons intruded in the higher-grade metamorphics, such as gneisses, amphibolites and migmatites [21], and covered by the Triassic platformal and Jurassic–Cretaceous deep-water Šiprúň or shallow-water Tatra successions of the Tatric Basin. The closure of this basin occurred in the Cenomanian to Turonian as documented by the termination of sedimentary successions by pelagic "flysch" sediments (e.g., [8] and references therein).

The Infratatricum developed from the deep-water Middle Jurassic to Lower Cretaceous successions which were deposited on a distant thinned continental margin basement [3,22]. The Infratatric deep-water Orešany Succession appears to have been formed in a transitional area to the south-Penninic oceanic basin. The southward Penninic subduction of Penninic Ocean opened the Infratatric fore-arc Late Cretaceous basin system on the upper plate [3,6].

A wide range of the blastomylonitic white mica $^{40}$Ar/$^{39}$Ar ages from ca. 100 to 50 Ma have been reported from the northern IWC wedge [3,7,23–28]. The Early Cretaceous zircon fission-track (ZFT) ages of ca. 144–135 Ma and an Eocene age of ca. 45 Ma obtained for basement rocks prove a very low-grade thermal overprinting in the Tatricum during the Early Cretaceous and Eocene extensions [29]. Zircon (U–Th)/He (ZHe) and apatite fission-track (AFT) ages between ca. 50 and 40 Ma are interpreted as erosional and tectonic exhumation of the basement units due to the extensional collapse of the Carpathian orogenic wedge [30].

The IWC southern (Meliatic–Gemeric–Veporic or MGV) wedge formed due to the closure of the Middle Triassic–Late Jurassic Neotethyan oceanic Meliata Basin and southward subduction of its oceanic and continental margin crust at ca. 160–150 Ma [4,23,24,31–33] and references therein. The MGV wedge also comprises a decoupled part of the Gemeric and Veporic thinned continental margin which accreted to the Meliatic subduction-related accretionary wedge at ca. 140–110 Ma ([34,35] and references therein). Contemporaneous exhumation and cooling of the overthrusting high-pressure metamorphosed Meliatic fragments is constrained by the zircon (U–Th)/He thermochronology [34] and the monazite in situ EPMA dating in the underlying southern Gemericum [36,37]. The Meliaticum overlies the Gemericum and is overlain by Silicic nappes [8] and references therein.

## 2. Geological Setting with Reappraisal of Tectonic Units, and Terminology

### 2.1. Allochthonous vs. Subautochthonous Fatricum

The Tatric Unit (Tatricum) appears to have weaker Alpine tectono-metamorphic overprinting than the Veporic Unit (Veporicum). Internally, the marginal Tatric domains show a more distinct tectono-metamorphic overprinting grade than its central, mostly unmetamorphosed sliver. These marginal zones were defined by [22] as individual structural units of the Tatricum, termed Infratatricum and Supratatricum. For example, the north-Tatric low-grade tectono-metamorphosed structural units in the Považský Inovec Mts., underlying the unmetamorphosed central Tatricum were called "Infratatricum". Similarly, the greenschist facies tectono-metamorphosed basement–cover structural units in the area south of the central Tatricum (i.e., between the Čertovica and Pohorelá lines) in the Tribeč Mts. and the eastern Nízke Tatry Mts. were called "Supratatricum" and were considered the root zone of the Mesozoic Fatric nappes ("Fatricum" after [14]) overriding the Tatricum. However, the compressed basement of the Fatricum root zone south of the Čertovica line is also traditionally termed "Northern Veporicum" in the eastern Nízke Tatry Mts. or "Tatricum" in the Tribeč Mts. Zobor block (e.g., [5,8,18,38]). To remove duplicate terminology of Fatricum/Northern Veporicum/Supratatricum, we choose to use the following principal Late Cretaceous structural units of the IWC with the Variscan basement participation: Infratatricum (N of Hrádok–Zlatníky line), Tatricum (between Hrádok–Zlatníky and Čertovica lines), Fatricum (between Čertovica and Pohorelá lines), Veporicum (between Pohorelá and Lubeník lines), Gemericum (between Lubeník and Rozňava lines) and the Hronicum, Meliaticum and Silicicum as Upper Paleozoic–Mesozoic rootless nappes. The Fatricum [14]

has an allochthonous supratatric position presented by the Krížna nappe system, and this was introduced approximately 20 years before the introduction of Supratatricum [22], therefore, we prefer Fatricum to Supratatricum although the latter would be an equivalent term to the Infratatricum from a structural viewpoint.

An important question is—what was the basement of the detached Mesozoic Fatric nappes finally bivergently overriding the Tatricum and Veporicum? Paleogeographically, there was a large domain of the Jurassic–Lower Cretaceous basement extension and the basin formation in the northern part of the IWC which, after the inversion, transformed into the main Late Cretaceous structural units such as the Infratatricum, Tatricum and Fatricum, including the newly defined subautochthonous Fatricum in this paper. We do not overlap a pre-Fatricum with the pre-Tatricum or pre-Veporicum but consider it as an individual paleogeographic domain that developed during the Jurassic–Early Cretaceous intracontinental extension. It is inferred that the central deep-water Middle Jurassic to Lower Cretaceous Fatric Basin Zliechov Succession formed on a strongly thinned pre-rift basin basement, while the laterally shallowing successions formed toward partly thinned continental margins. Thus, the south-Fatric continental margin is represented by the Veľký Bok, Lučatín and Veľké Pole type successions, and the north-Fatric margin by the Vysoká, Manín and Veľký Tribeč type successions of a large at least first-hundred-km wide Fatric Basin [8,13,14,39].

Following the Fatric Basin closure, the thinned basin basement and a part of the sedimentary infill underwent compressional southward underthrusting below the Veporicum in the front of the Neotethyan wedge. The exhumed tectono-metamorphosed basement-cover structural units from this suture zone are termed here the subautochthonous Fatricum to stress their close relationship to allochthonous or detached Fatric Mesozoic cover nappes sensu [14]. This area of the Fatricum is the homeland or the root zone of the Fatric nappes, finally bivergently overriding the Tatricum and Infratatricum, and the Veporicum. The southeast-vergent part of the subautochthonous Fatricum is represented by the Vápenica Nappe [21,40,41] with the Veľký Bok type Permian–Lower Cretaceous cover succession in the eastern Nízke Tatry Mts. (Figures 2 and 3). This nappe with a high-angle discordance is overlying the compressed south-Fatric (called also north-Veporic) structural units, such as the Ľubietová and Krakľová after [42]. The Staré Hory Nappe in the western Nízke Tatry Mts., and the Razdiel (an equivalent of the Vápenica Nappe) and Zobor nappes in the Tribeč Mts. of subautochthonous Fatricum are north-vergently overlying the Tatric sliver, and thus the Čertovica thrust fault is inferred to the north of the Tribeč Mts., although covered by the Cenozoic sediments.

The flat-lying Vápenica Nappe on the compressed WSW–ESE striking and mostly steeply south-dipping structure of the south-Fatric (former north-Veporic) phyllonite zones indicates postponed back-thrusting in relationship to the northward Fatric nappe formation. The Late Cretaceous sinistral transpression zone between the Tatricum and newly defined Veporicum (i.e., former southern Veporicum) or between the Čertovica and Pohorelá thrust faults formed after the expulsion of the Fatric nappes over the Tatricum.

The Vápenica Nappe consists of metamorphosed Veľký Bok type Permian–Lower Cretaceous succession overlying the tectonically reduced and sheared-off 100–300 m thick granitoid basement. Blastomylonitized granodiorites to tonalites and the sedimentary cover show the same low-grade tectono-metamorphic overprinting in the whole area of the eastern Nízke Tatry Mts. and the Upper Hron Valley. The thrust plane of the Vápenica Nappe may be a reactivated originally north-dipping extensional fault of the southern Fatric Basin continental margin. The SE/ESE-vergency of the Vápenica Nappe still under ductile conditions over the south-Fatric Ľubietová and Krakľová nappes is indicated by the WNW–ESE to E striking stretching lineation of granitoid blastomylonites. Similarly, the Krakľová Nappe is back-thrust over the Veporic front in the Kráľova Hoľa Massif also with a south-vergent duplex structure [40,41,43,44] and Geological Map 1:50,000, and Figure 2.

The Čertovica thrust fault separates the sutured subautochthonous Fatricum from the Tatricum. While the detached cover successions were included in the superficial Fatric

nappes overlying the Tatricum [8,13–15], the subautochthonous Fatricum was incorporated in the Albian–Cenomanian/Turonian accretionary wedge. The early evolutionary stage of this northern wedge formation is preserved toward the southwest in the Tribeč Mts. Razdiel and Zobor nappes (Figure 4) which were called blocks so far. Both nappes are distinctly tectono-metamorphosed, thus indicating inferred rear remnants of this wedge.

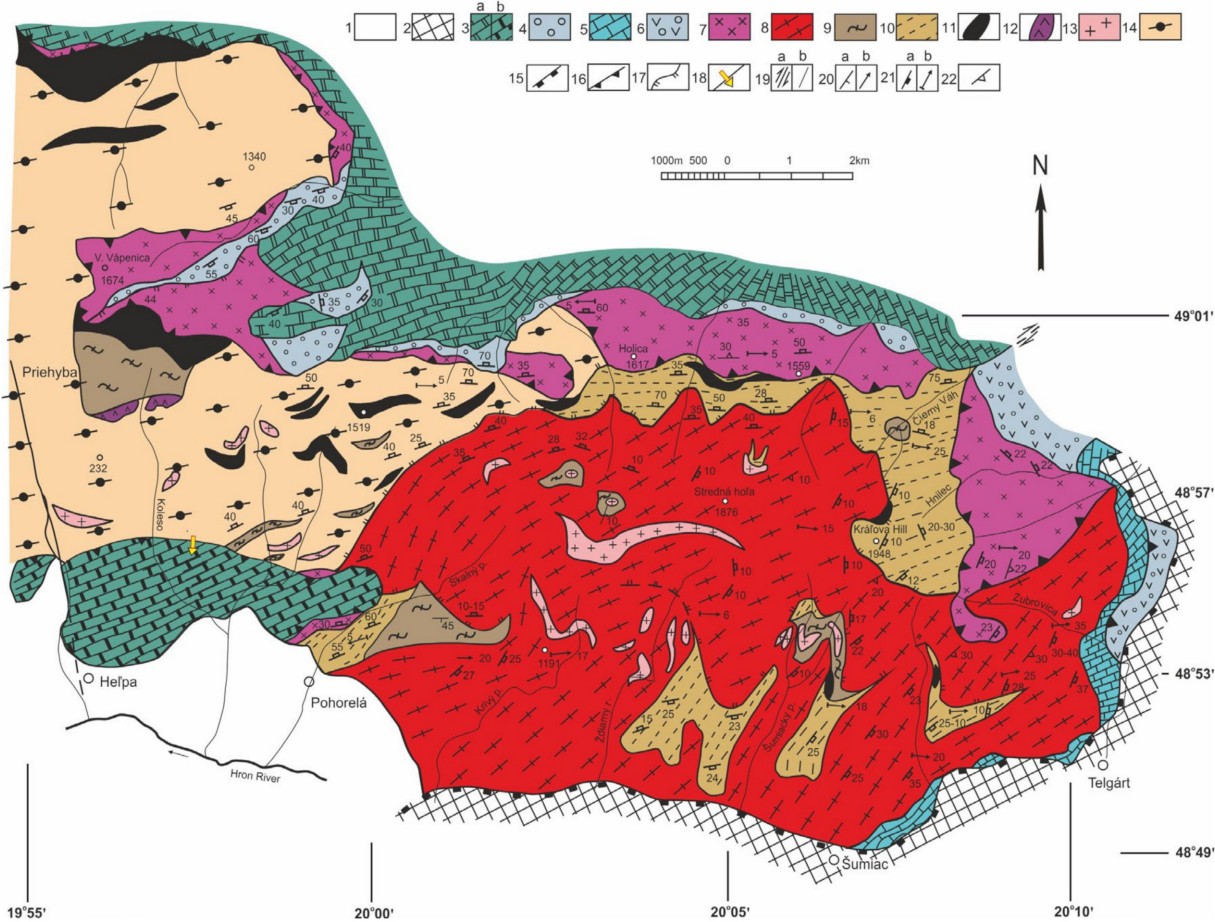

**Figure 2.** Geological map of the eastern termination of the Nízke Tatry Mts. (modified from [41]). Contact of the subautochthonous Fatric Vápenica and Krakľová nappes with the Veporic Kráľova Hoľa Nappe in a southvergent structure. 1—Quaternary and Cenozoic deposits; 2—Muráň Nappe (Silicicum, undivided); 3a—SE-vergent Vápenica Nappe on the northern slopes of the Nízke Tatry Mts. overlying the southernmost subautochthonous Fatric Krakľová Nappe; 3b—Vápenica Nappe in the Upper Hron Valley synform; 4—Permian siliciclastic sediments of the Veľký Bok Succession (a part of the Vápenica Nappe); 5—Triassic Foederata cover succession of the Veporicum; 6—Permian volcano-sedimentary succession of the Veporic(?) cover; 7—Variscan tonalites to granodiorites of the Vápenica Nappe hanging wall; 8—Variscan granites to granodiorites of the Veporic Kráľova Hoľa Nappe; 9,11,12—Gneisses (9), amphibolites to eclogites (11) and meta-peridotites (12) of the Upper Variscan structural unit overlying the micaschist gneisses of the Middle Variscan structural unit, included in Cretaceous Krakľová Nappe; 10—Cretaceous phyllonites of the Variscan metamorphics; 13—Leucocratic granites, aplites and pegmatites of unknown age; 14—Micaschist gneisses of the Krakľová Nappe; 15—Thrust plane of the Mesozoic Muráň Nappe (Silicicum); 16—Thrust plane of the Vápenica Nappe over the Krakľová Nappe in Fatricum; 17—Important thrust planes subsequent to subhorizontal nappe thrusting; 18—Thrust planes probably influenced by gravitational sliding; 19—Faults, strike-slips (a) and geological boundaries (b); 20—Pre-Alpine metamorphic schistosity/foliation (a) or lineation (b); 21—Alpine schistosity (a) or lineation (b); 22—Joint cleavage.

The Zobor and Razdiel nappes of the Tribeč Mts. show lithological differences in the Mesozoic cover successions [45]. The Variscan granitoid basement of the Zobor Nappe has the Triassic to Lower Jurassic sedimentary cover which continues upwards with a shallow-water Middle to Upper Jurassic succession. In contrast, the Razdiel Nappe

granitoid–micaschist–amphibolite basement has the pre-rift Permian to Lower Jurassic sedimentary cover which continues upwards with a deeper-water Middle to Upper Jurassic succession [45]. These nappes, however, possess similar lithofacies from the Tithonian up to the Cenomanian [13,45]. These differences in the basement and cover rocks suggest tectonic juxtaposition of these nappes within the subautochthonous Fatric domain although later separated by the Skýcov fault zone at the surface [21,41,45].

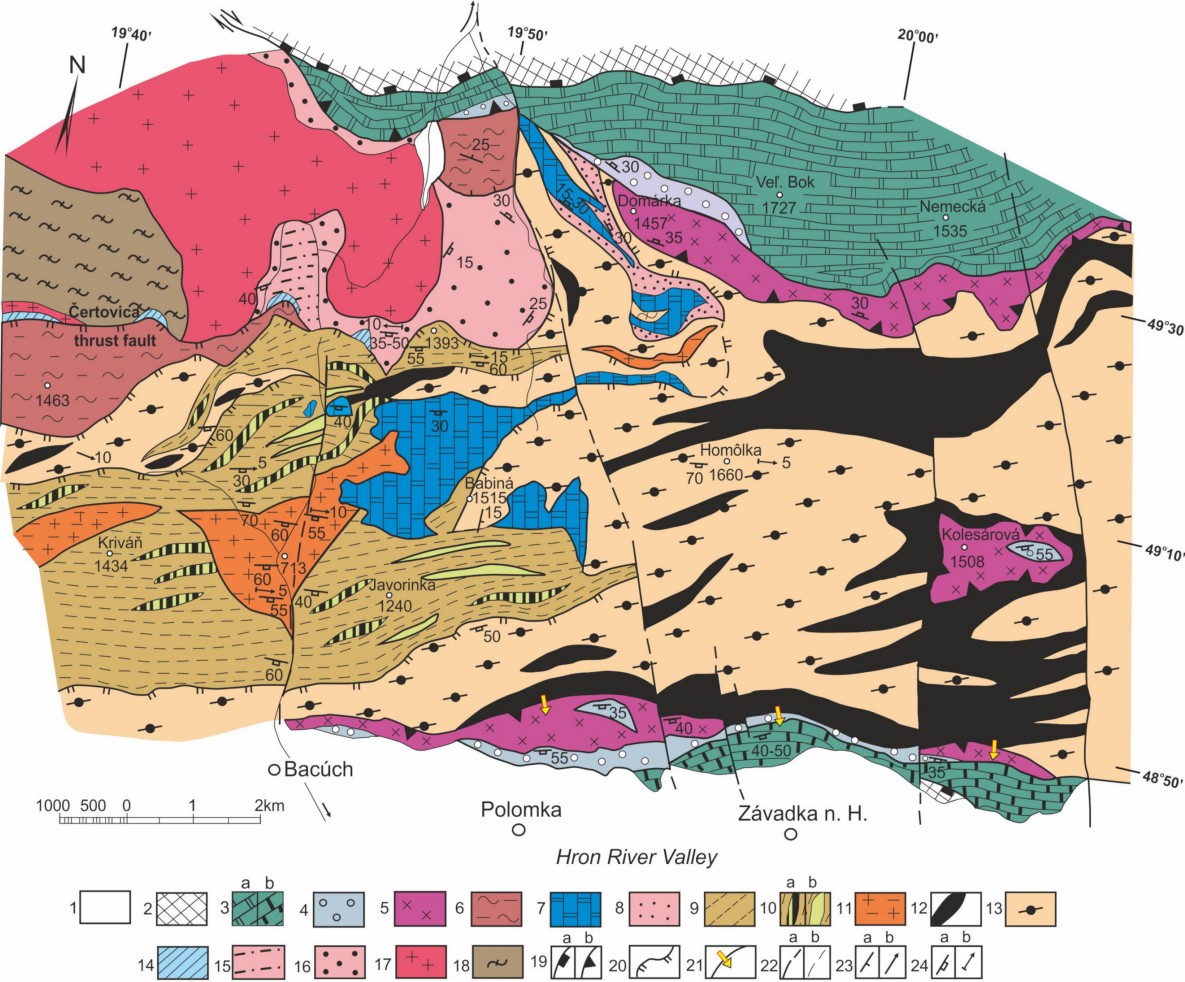

**Figure 3.** Geological map of the middle part of the Nízke Tatry Mts. (modified from [41]). Contact of the Tatricum and subautochthonous Fatric Ľubietová, Krakľová and Vápenica nappes around the Čertovica thrust fault. 1—Quaternary and Cenozoic deposits; 2—Choč Nappe (Hronicum, undivided); 3a—SE-vergent Vápenica Nappe on the northern slopes of the Nízke Tatry Mts. overlying the southernmost subautochthonous Fatric Krakľová Nappe; 3b—Vápenica Nappe in the Upper Hron Valley synform; 4—Permian siliciclastic sediments of the Veľký Bok Succession (a part of the Vápenica Nappe); 5—Variscan tonalites to granodiorites of the Vápenica Nappe hanging wall; 6—Ľubietová basement-cover nappe; 7—Triassic carbonates of the Krakľová Nappe; 8—Lower Triassic quartzites of the Krakľová Nappe; 9—11—Permian: siliciclastic sediments (9), acidic (10a) and basic (10b) volcanics, and granite porphyry (11); 12—Amphibolites of the Upper Variscan structural unit overlying the micaschist gneisses (13) of the Middle Variscan structural unit, included in Cretaceous Krakľová Nappe; 14—16—Triassic cover of Tatricum: carbonates (14), shales (15) and quartzites (16); 17—18—Variscan basement of Tatricum: granites to tonalites (17), gneisses to migmatites (18); 19a—Thrust plane of the Choč Nappe (Hronicum); 19b—Thrust plane of the Vápenica Nappe over the Krakľová Nappe in Fatricum; 20—Important thrust planes subsequent to subhorizontal nappe thrusting; 21—Thrust planes probably influenced by gravitational sliding; 22— Faults, strike-slips (a) and geological boundaries (b); 23—Pre-Alpine metamorphic schistosity/foliation (a) or lineation (b); 24—Alpine schistosity (a) or lineation (b).

A sheet of the blastomylonitized granitoids and the Triassic to Lower Cretaceous metamorphosed cover, exposed at the surface in the Zobor Nappe is terminated by the Skýcov NW–SE striking fault zone in the middle of the mountains. The Razdiel Nappe behind the Skýcov fault zone reveals the core of an open NW merging antiform composed of the phyllonitized micaschists and amphibolites. The limbs of this structure are formed by a blastomylonitized granitoid sheet composed of granodiorites to tonalites on the top, and the porphyric granites to fine-grained granites and leucogranites at the bottom. The SW part of the antiform at the Skýcov fault zone is compressed and the granitoids and the Permian cover rocks are underthrust below the micaschists and amphibolites of the antiform core.

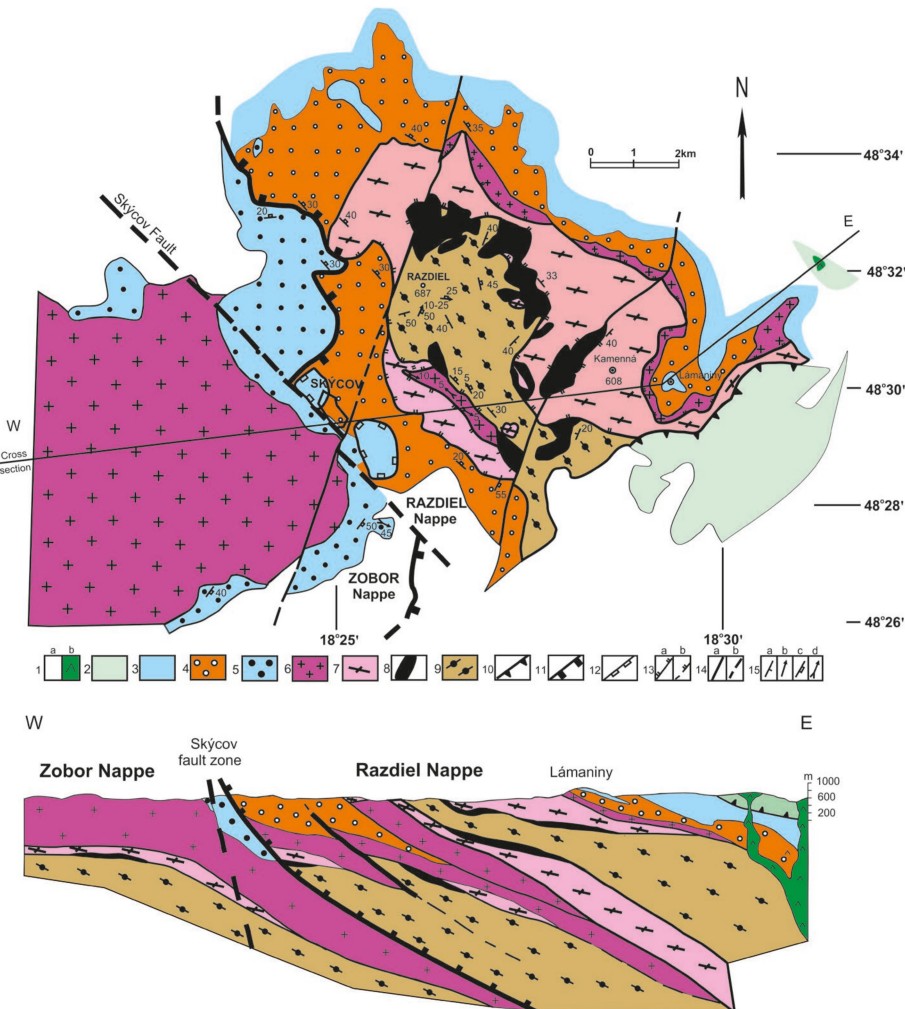

**Figure 4.** Geological map and cross-section of the subautochthonous Fatricum Zobor and Razdiel nappes in middle part of the Tribeč Mts. (modified from [41]). 1a—Quaternary and Cenozoic deposits; Outer Western Carpathians Flysch Belt (OWC-FB); 1b—Neogene andesites; 2—Choč Nappe (Hronicum); 3—Triassic—Lower Cretaceous Veľké Pole (Veľký Bok type) Succession [45] of the Razdiel Nappe; 4—Permian volcano-sedimentary succession of the Razdiel Nappe; 5—Lower Triassic quartzites of the Zobor Nappe (a part of the Veľký Tribeč Massif Triassic—Lower Cretaceous Succession); Variscan basement of the Razdiel and Zobor nappes: 6—Tonalites to granodiorites; 7—Granites mostly leucocratic; 8—Amphibolites; 9—Micaschists; 6–8—Variscan Upper Unit; 9—Variscan Middle Unit. Structural elements: 10—Thrust plane of the Hronic Choč Nappe; 11—Cretaceous thrust fault of the Razdiel Nappe over the Zobor Nappe in the subautochthonous Fatricum; 12—Detached fragments of the Mesozoic cover; 13—Important thrust planes after subhorizontal nappe thrusting; 14—Faults, strike-slips (a) and geological boundaries (b); 15—Pre-Alpine metamorphic schistosity/foliation (a) or lineation (b), Alpine schistosity (a) or stretching lineation (b).

The sedimentary Permian to Lower Cretaceous metamorphosed cover rocks is systematically on top of the upper granitoid sheet, and this situation indicates partly preserved Variscan basement tectono-stratigraphy in the core of the Razdiel Nappe Alpine structure [22,46–48]. The upper part of the Razdiel Nappe granitoid sheet, composed of the 300–400 thick blastomylonitized granitoids and the Permian–Lower Cretaceous cover of the Veľký Bok type Veľké Pole Succession [45], resembles the structure of the Vápenica Nappe in the eastern Nízke Tatry Mts. (Figure 3).

The NW-ward vergency of the Zobor Nappe was determined from the asymmetry of the S–C structures and NW–SE striking and mostly shallowly SE-dipping stretching lineation of the tonalite blastomylonites in the Nitra and Zlatno areas. The last extensional meso-structures indicate an opposite SE-vergent sliding. The same meso- and micro-structures and shear-sense were observed in the Razdiel Nappe granitoid blastomylonites which occur, for example, in the Chudá Valley and Kamenná Hill areas (see also the Section 4.1.2.).

### 2.2. The Infratatricum

The Selec Block in the Považský Inovec Mts. north of the Hrádok–Zlatníky thrust-fault was called "Carpathian Penninic" by [49] because of the Alpine tectono-metamorphic overprinting [50,51] and submerging below the central Považský Inovec Mts. Bojná Block of a typical unmetamorphosed Tatricum. The latter is considered analogous to the Lower Austroalpine Unit of the eastern Alps (e.g., [9,12,13]). The Selec Block structure was redefined by [22] as the Infratatricum, directly underlying the Tatricum (Figure 5). The general orogenic structure is documented in an interpretative reflection seismic cross-section by Tomek in [3].

The reconstructed deep-water Middle Jurassic to Lower Cretaceous Humienec Succession in the northern part of the Považský Inovec Mts. was considered a representative of the Penninic Ocean distal continental margin [3,6,52]. In contrast, [8,53] and Plašienka [5,54] considered that this succession, including the Upper Cretaceous sediments [55], exposed oceanic Váhicum [9] or the oceanic southern Penninicum pendant. However, the calc-alkaline meta-basalt fragments in the Upper Cretaceous "flysch" were derived from the Permian volcano–sedimentary succession of the higher Infratatric Inovec Nappe, and the inferred oceanic crust was not confirmed [3,6,51,52,56,57]. Because of the absence of Jurassic–Cretaceous ophiolites, we consider that this Jurassic–Lower Cretaceous continental margin Humienec Succession, the remnants of which were discovered in the lower Infratatric Humienec thrust sheet, is either an integrated part of the south-Penninic Realm [3,6] or an unknown basin, probably of the Fatric Realm (this paper, and the reasons see in the Discussion Section 5.2). In contrast, the higher Infratatric Inovec Nappe and the Tatric Panská Javorina Nappe are considered pendants of the Lower Austroalpine units s.l.

The micaschist-gneiss basement and the cover rocks of the higher Infratatric Inovec Nappe show distinct Alpine tectono-metamorphic overprinting in the higher anchimetamorphic to lower greenschist facies [3,50,51,56]. These overlie the lower Infratatric Humienec thrust sheet which also incorporates Upper Cretaceous Couches-Rouges-type (C.R.) reddish marlstones and "flysch" pelagites (Figure 5). The stratigraphic extent of the marls at least up to the Santonian was confirmed by Marginotruncana coronata Bolli foraminifera [6,52]. These sediments contain thin fine-grained detrital layers which are most likely turbiditic in origin. The extent of the C.R. was significant, from the Tatric edge up to the peri-PKB (see also [17,58,59]).

The inferred Penninic subduction (ca. 90–75 Ma) termination and the change from an extension to compression regime is recorded in the lower anchi-metamorphic overprinting of the C.R. during the D3 stage inferred at ca. 75–65 Ma, when the Belice Basin rebuilt to a trench-like "flysch" trough infilled by the Late Campanian(?)–Maastrichtian siliciclastic and calciclastic pelagites. Contemporarily, the C.R. underthrust below the Inovec Nappe. Most of the lower Infratatric frontal Humienec thrust sheet members occur in the "flysch", often in the form of olistoliths and scarp breccias which shaded southwards in the basin [3,6].

In contrast, the "flysch" sources from the higher Infratatric Inovec Nappe were derived from the south, or from the exhumed rear Inovec Nappe as a part of the collapsing Albian–Cenomanian/Turonian wedge front (Figure 6A,B).

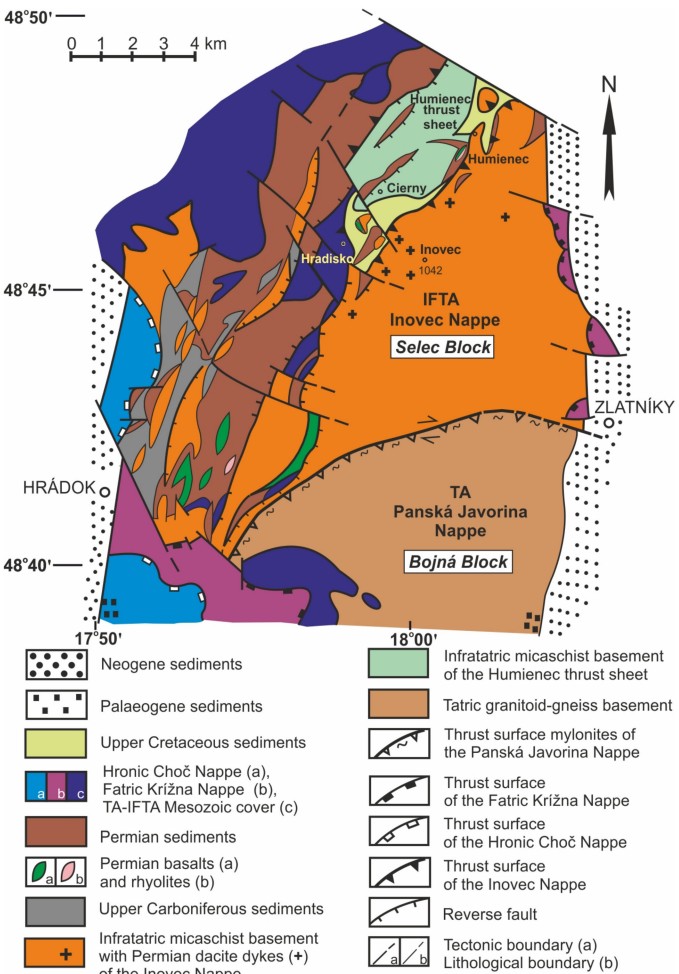

**Figure 5.** Geological sketch-map of the northern and middle part of the Považský Inovec Mts. (compiled from [3,6,50]; higher Mesozoic Krížna and Choč nappes from [60]. TA—Tatricum, IFTA—Infratatricum.

Part of the Infratatric structure is also exposed in the relatively small Hlohovec Block in the southern part of the Považský Inovec Mts. [39,61]. While the Selec Block structural units were described in more detail by [3] and references therein, the structure of the Hlohovec Block (Figure 7) is less known, and it will therefore be described here in greater detail and include our new field observations.

The HPJ-1 Jašter structural–geological drill-hole at Hlohovec revealed steeper 50–70° direct tectonic contact of the Upper Cretaceous Couches-Rouges-type reddish marls (C.R.) with the deformed granodioritic basement. This situation was interpreted as marls sliding into the adjoining Neogene basin along an extension fault, and contemporary exhumation of the basement granodiorite and the Triassic–Jurassic cover of the Tatricum [62].

A typical feature of the Hlohovec Block tectonic structure is the Middle Triassic marble thrust sheet overlying the C.R. and "flysch" [39,61,63]. Similar Upper Cretaceous rocks were also reported in the northern Selec Block and the Soblahov-1 drill-hole north of the Považský Inovec Mts. [3,6,53,55].

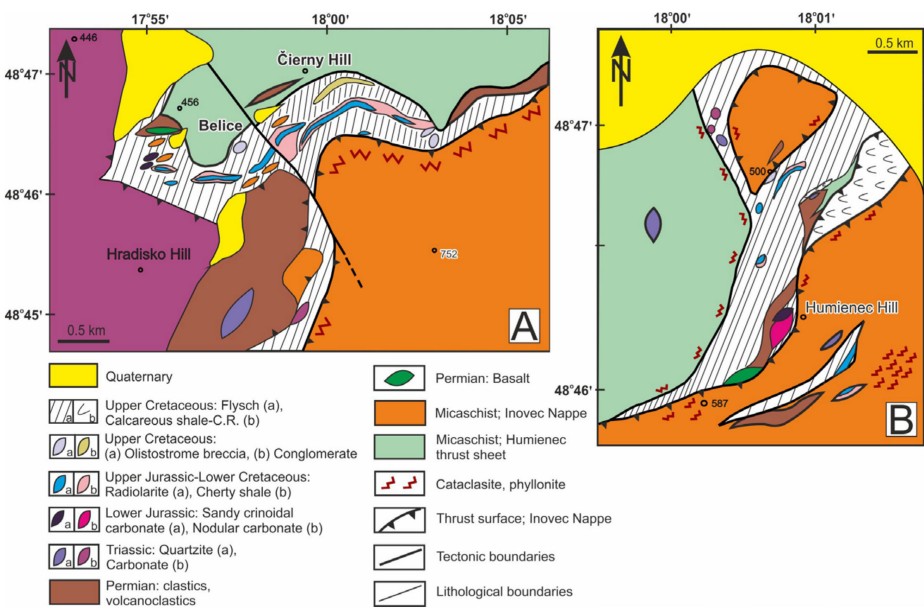

**Figure 6.** Detailed map of the Upper Cretaceous flysch sediments with olistoliths of metamorphosed Permian to Lower Cretaceous rocks in the Čierny vrch Hill (**A**) and Humienec Hill (**B**) areas [3,6], (modified).

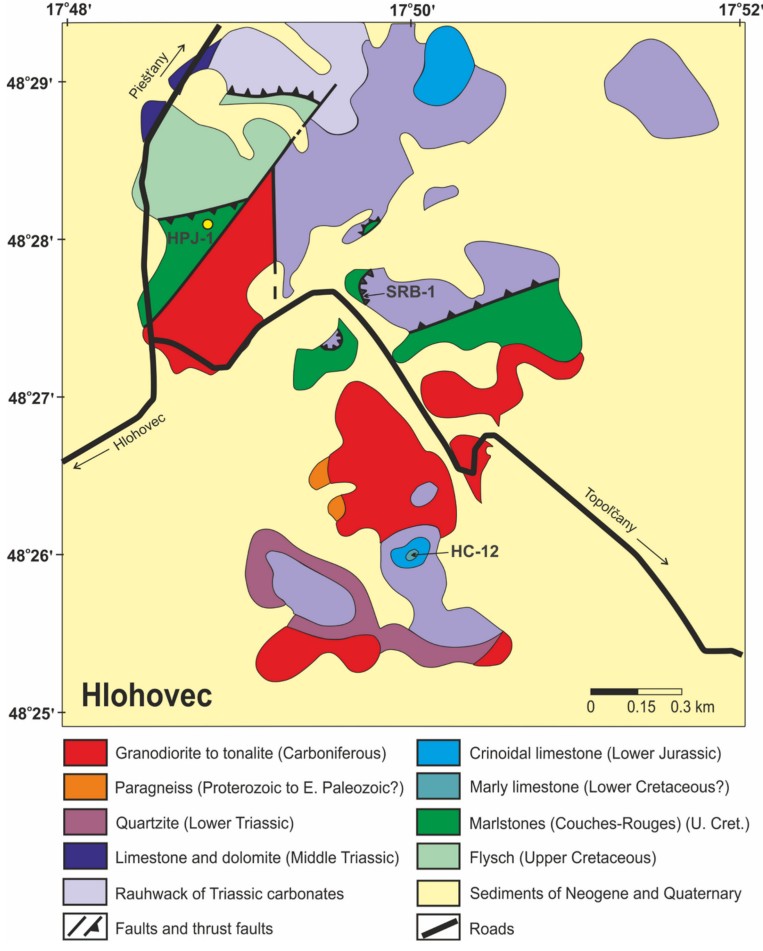

**Figure 7.** Geological map of the Infratatric Považský Inovec Mts. Hlohovec Block (modified from [60,62], and our study). Position of the Ar/Ar dated samples SRB-1 and HC-12.

Although the large area of the Hlohovec Block is under the Quaternary sediments [60], the deeply eroded valleys and quarries yield sufficient information on the geological structure. The Soroš quarry near Hlohovec township reveals direct tectonic contact or the thrust plane of the Middle Triassic marble sheet over the anchi-metamorphosed C.R. (Figures 7 and 8A). The metamorphosed hemipelagic marly limestone (Lower Cretaceous?) in association with the recrystalized Jurassic sandy crinoidal limestones, Middle Triassic marbles and Lower Triassic meta-arkoses and meta-quartzites complete the cover succession of a small granodiorite massif near Hlohovec. While the Lower Triassic rocks are closely bound to the underlying granodiorite, the Middle Triassic–Lower Cretaceous(?) marble sheet was detached from the basement and thrust over the C.R. as an independent tectonic body. This is assigned here as the higher Infratatric Hlohovec Nappe, with equivalent structural position to the higher Infratatric Inovec Nappe in the northern Selec Block, particularly the Hradisko Hill thrust sheet of this nappe (Figures 5 and 7).

The laminated grey calcitic and calcitic–dolomitic marbles have metamorphic schistosity covered with clearly visible white mica aggregates. The schistosity planes (S1) are shallowly ESE dipping. The stretching lineation on the S1 planes is WNW–ESE striking and 10–15° ESE dipping. Dark-gray marbles are transformed into yellow white carbonatic schists at the C.R contact (D3 stage). These new dense schistosity planes (S2) contain newly formed very fine-grained white-mica aggregates. The secondary schistosity of the marble mylonites/ultramylonites (S2) is equivalent to the anchi-metamorphic schistosity (S1) of the underlying C.R. The intrafolial folds of the C.R. marls are cosscut with the secondary cleavage S2 which may be related to the later (D4) thrusting of the marly slates over the "flysch" sediments in the foreland.

A younger recrystalized fine-grained white-mica2 aggregates formed in the hanging wall marble ultramylonites of the Triassic–Jurassic–Early Cretaceous(?) marble thrust-sheet, most likely during the underthrusting of the footwall C.R. type reddish marls during the D3 compression stage. This may explain the differences in the metamorphic overprinting compared to the C.R. slates with "flysch" sediments after the D4 stage at ca. 65–50 Ma when the whole Infratatric Upper Cretaceous sedimentary succession was included in the IWC Eocene accretionary/orogenic wedge [3]. Although the fold-cleavage structures also occur in the "flysch" sediments, they exhibit negligible metamorphic overprinting at the boundary of the diagenesis and lower anchi-metamorphism (see the Section 4.1.1).

The advanced D4 stage occurred in the brittle-ductile regime. Kink folds with NE–SW axis and SE-ward vergency occur in the C.R. slates directly below the marble thrust-plane in the Soroš quarry. Moreover, the marble thrust sheet hanging wall contains cataclasite and rauhwacke layers which may have formed during overthrusting of the almost unmetamorphosed "flysch" sediments (Figure 7). Steepened thrust-faults and back-thrusts are typical in the NW part of the Selec Block (Figure 5), thus indicating an important collision–transpression event most likely related to the PKB Cenozoic structure formation.

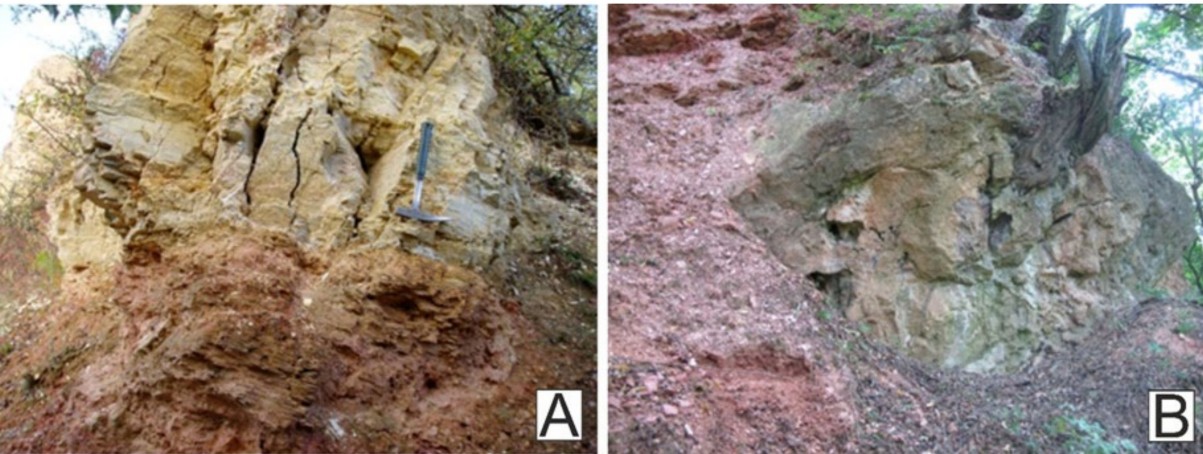

**Figure 8.** Tectonic structure in the Soroš quarry at Hlohovec town: (**A**) thrust plane of the Triassic marbles of the higher Ifratatric **Hlohovec Nappe** over the reddish Couches Rouges type marly slates (C. R.) of the lower Infratatric **Jašter thrust sheet**; (**B**) infolded lens-shape marble fragments in C.R.

Part of the rugged Albian–Cenomanian/Turonian wedge basement, including the tectonized granodiorites, were tectonically emplaced in the anchi-metamorphosed C.R. sediments forming a mélange. Similarly, the distinctly tectono-metamorphosed marble fragments of the Hlohove Nappe thrust sheet in the form of lensoidal blocks (xm–x10m) were infolded into the soft reddish marly slates and this mélange-like structure is also documented in the Soroš quarry near Hlohovec (Figure 8B).

*2.3. Tatricum*

The Tatric Basin shallow—(Tatra) and deep-water (Šiprúň) Mesozoic cover successions terminate with the Albian–Cenomanian flyschoid Poruba Formation (e.g., [8,39]). Turonian age of the formation was confirmed only in the Tatra and Veľká Fatra Mts. [64,65]. The youngest Turonian–Santonian age of the Tatric Mesozoic cover was determined in the Hubina Formation which discordantly overlies the Poruba Formation in the Bojná Block of the Považský Inovec Mts. [17]. This formation partly resembles the Infratatric Upper Cretaceous succession of the Selec and Hlohovec blocks in the Považský Inovec Mts.

The Tatric basement–cover complexes show indistinct tectono-thermal overprinting, except for localized blastomylonitic and phyllonitic zones in the granitic and metamorphic rocks, respectively. These Alpine reactivation zones often associate with cataclasites, ultracataclasites and pseudotachylytes [66–68].

The phyllonite zone of the Tatricum's western margin in the Lúčanská Malá Fatra Mts. merges SE-wards below the crystalline massif and appears in the Valča and Bystrička valleys tectonic windows [7]. The newly formed white mica yielded $72.4 \pm 2.7$ Ma $^{40}$Ar/$^{39}$Ar plateau age [26], and a similar $^{40}$Ar/$^{39}$Ar step age cluster was obtained by [7]. The newly formed white mica of Ms to Ph composition from the phyllonites and underthrust marble mylonites (Figure 9) yielded higher anchi-metamorphic to lower greenschist facies temperatures at lower to medium pressures [69] (Table 1).

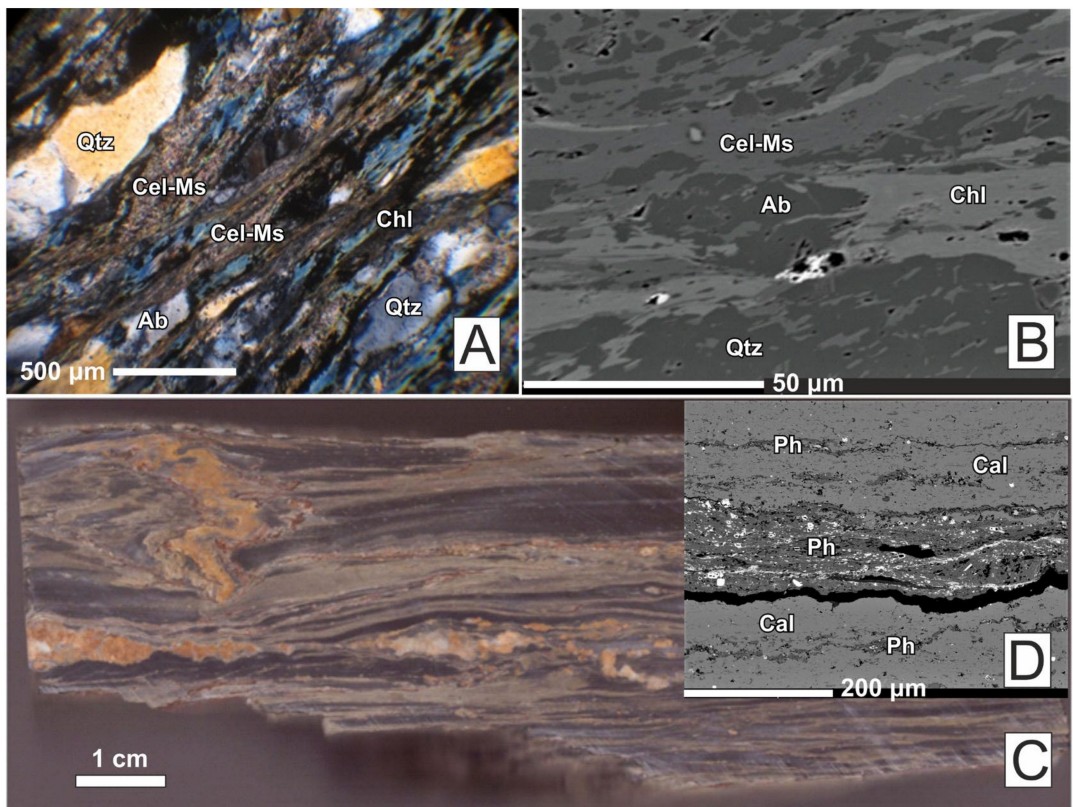

**Figure 9.** Gneiss phyllonites (**A**,**B**) overlying the Middle Triassic marble mylonites (**C**,**D**) along the thrust plane of the Malá Fatra Mts. Tatricum over the inferred Infratatricum of the Kozol antiform. (**A**) Chl–Ph phyllonites in the Kunerád Valley (MF-KU4) overlying Triassic marble mylonites (C); (**B**) phyllonites (s. MF-84) of the Tatricum hanging wall in the Bystrička Valley tectonic window [7]; (**C**) marble mylonite with NW-vergent infrafolial folds (MF-KU3); (**D**) BSE image of marble mylonite in C.

**Table 1.** Estimated metamorphic conditions from the Tatricum thrust fault zone over the inferred Infratatricum of the Kozol antiform to the west. Supplementary data to [7,69].

| Sample | Sample Description GPS Coordinates | Chl(C) T (°C) ±50 °C | Chl(J) | Chl(V) | T* (°C) | Ph(M) P (MPa) ±200 MPa | Chl-Ms(V) T (°C) ±50 °C | P (MPa) ±200 MPa |
|---|---|---|---|---|---|---|---|---|
| MF-84 | gneiss phyllonite N 49°02.893′ E 18°50.894′ | 319–338 | 321–339 | 309–327 | 314 | 450 | 283–294 | 400–450 |
| MF-KU4 | gneiss phyllonite N 49°04.981′ E 18°44.897′ | 311–329 | 318–336 | 287–314 | 305 | 600 | 241–253 | 550–560 |
| MF-KU1 | gneiss phyllonite N 49°05.086′ E 18°44.936′ | 263–311 | 235–286 | 245–250 | 265 | 300 | - | - |

Chlorite thermometry by [70]—(C), [71]—(J). Si-in-Ph barometry after [72]—(M) based on average Si-in-Ph content. Chl thermometry and Chl–Ms thermobarometry after [73]—(V). *—Average of calculated T or estimated T for samples with no suitable minerals for thermometry.

The most typical are common fold and thrust structures of the basement and the cover rocks in the Tatra and Nízke Tatry Mts. which formed under brittle–ductile conditions. Despite very low-temperature deformation conditions, the Upper Variscan Unit granodiorite kalifeldspar at Velické Pleso Lake in the Tatra Mts. yielded minimum measured $^{40}$Ar/$^{39}$Ar step-heating ages of ca. 48–36 Ma [28]. This documents basement overheating before Paleogene sedimentation, and this was most likely due to the late Eocene basement extension and exhumation.

The zircon (U–Th)/He thermo-chronology ages of 48–22 Ma in the Tatra Mts. are consistent with the extension and the Paleogene fore-arc sedimentation [74,75]. The apatite

fission track ages range from 20 to 15 Ma, and apatite (U–Th)/He ages range from 18 to 14 Ma (l.c.). These ages overlap with the cooling ages of Tatricum and Infratatricum in the Považský Inovec and Malá Fatra Mts. [29,76,77].

The kalifeldspar $^{40}$Ar/$^{39}$Ar age of 48 Ma from the Tatra Mts. [28] is compatible with $48 \pm 2$ Ma white mica plateau age from the Tatricum hanging wall blastomylonites in the Hrádok–Zlatníky shear zone (Figure 5) at the contact of the Tatric Bojná and the Infratatric Selec blocks in the Považský Inovec Mts. [3,7]. Granodiorite blastomylonites at the Žiarska Valley pass in Tatra Mts., at the reactivated contact of the Upper and Middle Variscan units, could be of similar age. These $^{40}$Ar/$^{39}$Ar ages of ca. 48–36 Ma [28] suggest final Tatricum and Infratatricum structuring in the Eocene.

### 2.4. Penninic, Infratatric and Tatric Units of the Malé Karpaty Mts.

The extremely compressed tectonic structure of the Malé Karpaty Mts. units resembles mélange type zones along the contact of the Penninic and Austroalpine units in the Eastern Alps; for example, from the Engadine and High Taurs tectonic windows (cf. [11] and references therein).

The south-western Malé Karpaty Mts. reveal the bottom Borinka Unit, likely representing a marginal part of the oceanic southern Penninicum (~Váhicum, after [8,9,78]). The northern-most inferred higher Infratatric Bratislava–Modra basement–cover nappe overlies the Penninic Borinka and the inferred lower Infratatric Orešany units, and in turn is overlain by the Fatric and Hronic nappes ([16,79], and 1:50,000 Geological Map). Tatricum presence is not clear. Therefore, the Hainburg/Bratislava–Modra Nappe may be a part of the SW termination of the extended Tatric sliver, or this is already part of the higher Infratatricum. The structural position of the Borinka Unit partly resembles the Matrei Zone Penninic margin in the Tauern Window (cf. [11]).

The inferred Penninic oceanic rift likely continued along the front of the present-day Infratatricum [22]. The continental margin material occurs in the Jurassic scarp-breccias of the Borinka Succession [78]. The Infratatricum of the Malé Karpaty Mts. thus appears to have had the closest paleogeographic position to the oceanic southern Penninicum (~Váhicum after [9]).

The internal structure reveals the Hainburg/Bratislava and Modra–Orešany nappe units, including their deep-water cover successions (Devín and Kuchyňa or Solírov–Orešany, respectively), which overlie the Jurassic succession of the inferred Penninicum oceanic margin (Figure 10). The higher, likely Tatric Kadlubka Succession is attached to the Fatric Nappe hanging wall. The general orogenic nappe structure was reported from deep reflection seismic cross-sections through the Malé Karpaty Mts. by [80].

The basement and cover rocks were mostly anchi-metamorphosed [7,20,69,81]. The Bratislava–Devín road-cut and the Prepadlé Valley at Borinka reveal granitic blastomylonites with newly formed phengitic white mica dated at ca. 80 Ma by K/Ar [82] and $^{40}$Ar/$^{39}$Ar [7] methods.

Phengitic white mica from quartzitic meta-sandstones from the Hrubá Dolina Valley quarry near Pezinok township was $^{40}$Ar/$^{39}$Ar-dated at ca. 73 Ma from the upper overturned limb of a recumbent fold in Figure 10. The dynamically recrystalized Qtz aggregates indicate a minimum temperature of around 250–300 °C for this common fold-thrust basement–cover structure [82]. This implies the closure of this inferred southern Penninic branch at ca. 90–80 Ma, followed by the north-Infratatric accretionary wedge formation.

The final structure in Figure 10 includes the SE-vergent thrust faults, and the Hainburg/Bratislava–Kuchyňa segment was back-thrust over the Modra–Orešany one [7,69,79,82]. The Modra shear zone tonalite blastomylonites merge below the Hainburg/Bratislava–Kuchyňa segment. There are also remarkable ultracataclastic zones accompanying the different lithological and tectonic boundaries [66].

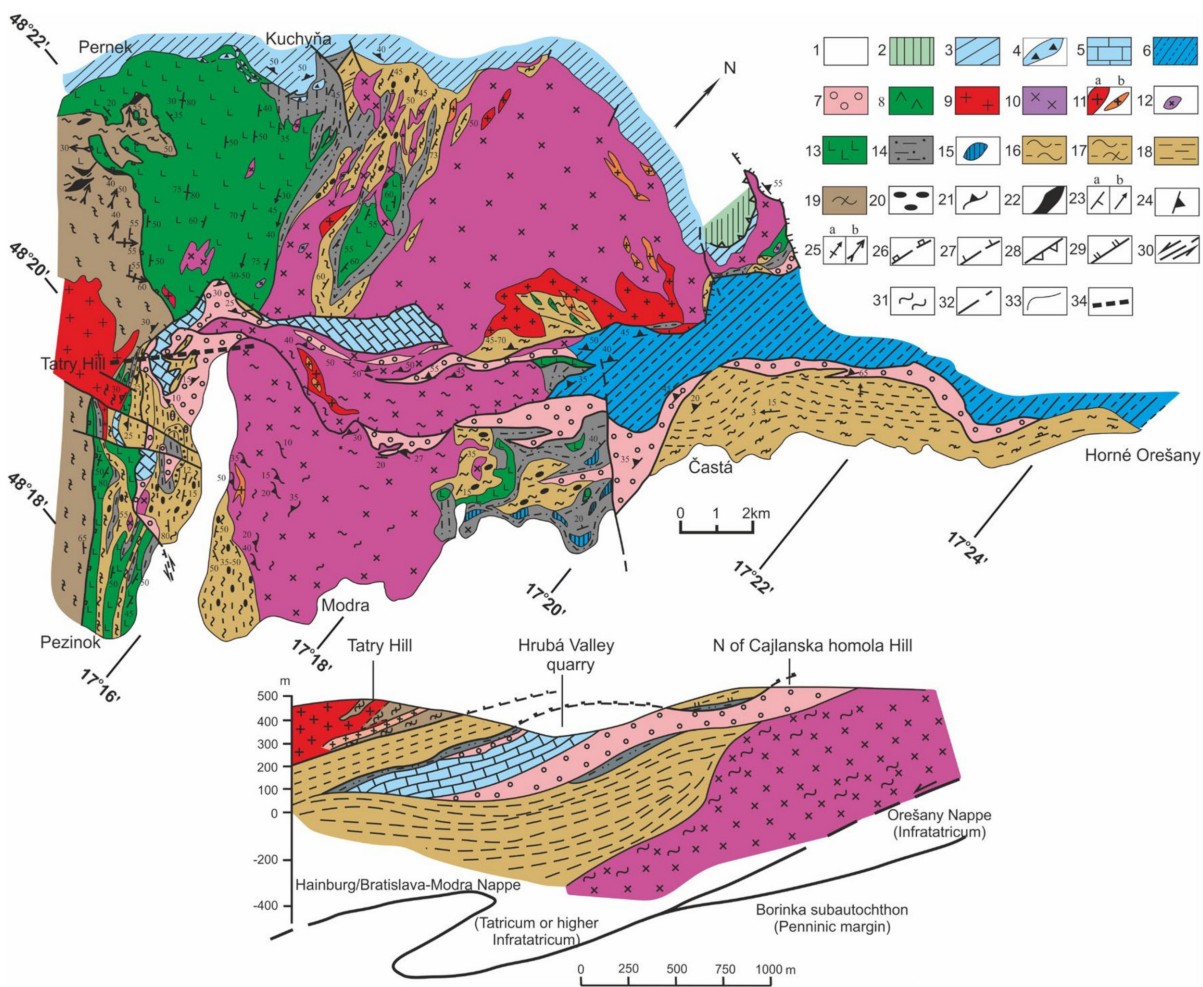

**Figure 10.** Geological map (modified from [7,82]) and a cross-section in central part of the Malé Karpaty Mts. (modified from [41,82] with inferred underlying Infratatric and Penninic tectonic units. 1—Quaternary and Tertiary undivided; 2—Meosozoic rocks of Krížna Nappe (Fatricum); 3–5—Mesozoic cover rocks of the Hainburg/Bratislava–Modra Nappe (Tatricum or higher Infratatricum); 6—Mesozoic rocks of the (lower) Infratatric Orešany Nappe; 7—Permo-Scythian metasediments; 8—Amphibolites; 9—Granites; 10—Tonalites to granodiorites; 11—Veins of leucocratic granites, aplites and pegmatites; 12—Diorites; 13—Metabasites (Devonian ophiolites); 14—Black schists; 15—Calc-silicate marbles 16—Meta-sandstones, sandy schists; 17—Clayey-sandy schists; 18—Phyllites (14–18 Devonian); 19—Gneisses (Ordovician–Silurian?); 20—contact-metamorphic hornfelses; 21—Alpine (Cretaceous) blastomylonitic planes; 22—Cataclasites, ultracataclasites; 23—Pre-Alpine (Early Carboniferous, Variscan) schistosity/foliation (a), metamorphic lineation (b); 24—Alpine (Cretaceous) schistosity; 25—Alpine metamorphic lineation (a), kink fold axis (b); 26—Pre-Alpine post-granitic thrust plane (Late Variscan); 27—Alpine (Cretaceous) thrust plane; 28—Krížna Nappe thrust plane; 29—Alpine thrust planes; 30—Strike-slip fault; 31—Cretaceous blastomylonites; 32—Faults; 33—Lithological boundaries; 34—Cross-section line.

**Abbreviations of the rock-forming mineral names used in the text, tables and figures are**: Ab—albite; Act—actinolite; Amp—amphibole; An—anorthite; Ap—apatite; Aug—augite; Bt—biotite; Cal—calcite; Cel-Ms—celadonite-enriched Ms; Chl—chlorite; Cld—chloritoid, Cpx—clinopyroxene; Czo—clinozoisite; Dol—dolomite; Ed—edenite; Ep—epidote; Fsp—feldspar; Grt—garnet; Ilt—illite; Ilm—ilmenite; Kfs—potassium feldspar; Lws—lawsonite; Mnz—monazite, Ms—muscovite; Pg—paragonite; Ph—phengite (Cel-rich Ms); Pl—plagioclase; Pre—prehnite; Pu—pumpellyite; Qtz—quartz; Rt—rutile; Spl—spinel; Sps—spessartine; St—staurolite, Stlp—stilpnomelane; Tr—tremolite; Ttn—titanite; Vsv—vesuvianite; Zrn—zircon.

## 3. Materials and Methods

The field investigation focused on the Infratatric, marginal Tatric and the inferred subautochthonous Fatric basement–cover structural complexes. Meso- and microstructures of representative rock-types which showed tectono-metamorphic overprinting were subjected to more detailed petrological and geochronological investigations.

The mineral composition and textures of the studied rocks were first investigated in polished sections by polarised light microscope. The mineral element compositions were measured by electron probe micro-analysis (EPMA) on a Cameca SX–100 electron microprobe at the State Geological Institute of Dionýz Štúr in Bratislava, and by JEOL Super-probe JXA 8100 at the Earth Science Institute of Slovak Academy of Sciences in Banská Bystrica (Slovakia). The EPMA was applied on white micas, chlorite, actinolite, epidote, apatite, titanite and albite. The voltage was accelerated by 15 kV, with a beam current of 20 nA focused to 3–5 μm, and the following standards and measured lines were used: Si (TAP, Kα, wollastonite), F (LPCO, Kα, LiF), Cl (LPET, Kα, NaCl), Al (TAP, Kα, $Al_2O_3$), Ca (LPET, Kα, apatite), Fe (LLIF, Kα, fayalite), Ti (LLIF, Kα, $TiO_2$), K (LPET, Kα, orthoclase), Na (TAP, Kα, albite), Mg (TAP, Kα, forsterite), Mn (LLIF, Kα, rhodonite), Cr (LLIF, Kα, Cr), Ni (LLIF, Kα, Ni). Detection limits were within 0.01–0.05 wt.% of oxide.

Chlorite crystal formulae calculation was based on 14 anions and performed by Windows program WinCcac [83]. Micas were re-calculated based on 11 oxygens. The Chlorite and Mica classification is according to [84] and [85], respectively. Amphibole analyses were re-calculated by Excel Spreadsheet of [86] based on the classification of [87]. The metamorphic overprinting conditions were estimated by chlorite and paragonite-muscovite thermometers [70,71,73,88] and chlorite–phengite [73] and phengite [72] barometers combined with Perple_X pseudosection modeling.

Chl–Mica–Qtz–$H_2O$ multi-equilibria geothermobarometry [73] was carried out by XmapTools [89] under the NCKFMASHO ($Na_2O$–CaO–$K_2O$–FeO–MgO–$Al_2O_3$–$SiO_2$–$H_2O$) chemical system using α-quartz, mica end members: muscovite, Mg and Fe-celadonite and chlorite end members: clinochlore, daphnite, sudoite, Mg and Fe-amesite, pyrophyllite. Standard state and solid solutions of [73] for chlorites and [90] and [91] for mica were used. Water activity of $H_2O$ = 1 was used for calculations of all samples. Results from all geothermobarometry methods can be found in Section 4.1.

Three meta-basite samples were used in the Perple_X modeling: One from the Infratatricum of the Považský Inovec Mts. Selec Block, and two from the subautochthonous Fatricum; one from the Tribeč Mts. Razdiel Nappe and one from the eastern Nízke Tatry Mts. Vápenica Nappe.

The P–T pseudosections were calculated by the Perple_X computer program package (ver.6.9.0) in 4–7 kbar and 200–400 °C ranges [92,93]. Calculations were performed using the thermodynamic dataset of [94]. The solid-solution models of [95] and [96] were chosen: cAmph (G) for amphiboles, Chl (W) for chlorites, Mica (W) for potassic white mica and Bio (W) for biotite. In addition, Pu was used for pumpellyite and Ep (HP11) for clinozoisite–epidote. However, no solid-solution model was selected for plagioclase because this phase is almost pure albite in very-low-grade meta-basic rocks. Samples were calculated under the NCKFMASHTO chemical system ($Na_2O$–CaO–$K_2O$–FeO–MgO–$Al_2O_3$–$SiO_2$–$H_2O$–$TiO_2$).

The studied rocks' original compositions were slightly modified to fit this 9-component system, as follows: (1) CaO was reduced according to bulk-rock phosphorous content, and assuming these elements were bound exclusively to ideally composed apatite and (2) the $H_2O$ contents were set in excess to enable free hydrous fluid phase formation at relatively low P–T. The pseudosections were contoured with isoplets for Mica and Chl chemical parameters using Perple_X werami and pstable sub-programs to supply raw data. The whole-rock composition of samples used for thermodynamic modeling are in Table S1 (Supplementary Materials).

Three samples and five mineral fractions were selected for the $^{40}Ar/^{39}Ar$ dating: Meta-quartzite sample (ZI-3) containing a homogeneous aggregate of metamorphic white mica in recrystalized quartz aggregate; metamorphosed marly limestone (HC-12) containing white-

mica rich layers in recrystallized calcite aggregates; schistose marble (SRB-1) containing two grain-size generations of white mica and one biotite generation in the recrystalized calcite matrix. Phengitic white mica aggregates of samples ZI-3, HC-12 and SRB-1 were chosen for constraining the age of Alpine metamorphism and related deformation stages. Muscovite and biotite fractions from sample SRB-1 were used for dating the inferred basement rocks' sources.

Optically fresh, unaltered and homogeneous grains of muscovite, biotite and phengite for $^{40}Ar/^{39}Ar$ dating were hand-picked from a 125–212 μm-size fraction under a stereomicroscope, then ultrasonicated and thoroughly rinsed in MilliQ water to remove impurities from the grains. $^{40}Ar/^{39}Ar$ dating was performed at the Western Australian Argon Isotope Facility at John de Laeter Centre (Curtin University, Perth). Multi-grain aliquots of the samples were loaded on an aluminium disc, with the GA1550 biotite standard (99.738 ± 0.100 Ma; [97] used as a flux monitor and irradiated for 40 h in the Oregon State University nuclear reactor (USA). All samples were unpacked after cooling for several weeks and individually loaded into copper (Cu) planchettes for analysis on the ARGUS-VI multi-collector mass spectrometer outfitted with four Faraday collectors and an ion-counting CuBe electron multiplier (CDD) at Curtin University. All analyses were conducted on single-grain aliquots, and this enabled direct determination of $^{40}Ar$ concentrations in single grains and prevented mixing of different age populations. Mica grains were analysed by incremental heating using a continuous 100 W PhotonMachine© CO2 laser GA1550. After heating, reactive gases were purified in an extra low-volume stainless steel extraction line of 240cc volume, using one SAES AP10 and one GP50 getter. The Ar isotopes were measured in static mode using a low volume 600cc Thermofisher© ARGUS VI mass spectrometer set with a permanent resolution of around 200 [98]. Measurements were performed in multi-collection mode using 4 Faraday cups with 1012 ohm resistors to measure 40, 38 and 37 masses and a 1013-ohm resistor to measure the 39 mass. The 36 mass was then measured by ultra-low background compact discrete dynode ion counter. The relative abundance of each mass was measured simultaneously using 10 cycles of peak-hopping and 16 s integration time for each mass. Raw mass-spectrometer data were reduced using the Argus program written by M.O. McWilliams and run under a LabView environment. The raw data were processed using the ArArCALC software [99] and the ages were calculated using the decay constants of [97]. A neutron fluence parameter J-value (0.0109134 ± 0.06%) was calculated from standard grains within the disc. mass discrimination was monitored regularly through the analysis using an automated air pipette, and this provided mean values of 0.993719 (±0.05%) per Dalton atomic mass unit relative to an atmospheric or trapped $^{40}Ar/^{36}Ar$ ratio of 298.56 ± 0.31 [100]. The correction factors for interfering isotopes were ($^{39}Ar/^{37}Ar$) Ca = 6.95 × $10^{-4}$ (±1.3%), ($^{36}Ar/^{37}Ar$) Ca = 2.65 × $10^{-4}$ (±0.84%) and ($^{40}Ar/^{39}Ar$) K = 7.30 × $10^{-4}$ (±12.4%; [101]). All parameters and relative abundance values are provided in Table S2 following the recommendations of [102], and corrected for blank, mass discrimination and radioactive decay.

## 4. Results

### 4.1. Petrography, Mineral Chemistry and Tectono-Metamorphic P–T Conditions

4.1.1. Infratatricum

Považský Inovec Mts. Selec Block (Inovec Nappe and Humienec Thrust Sheet)

**Petrography**

Our research focused on: (1) the Alpine metamorphic overprinting textures in the retrogressed basement phyllonites and the infolded Upper Paleozoic rocks (Figure 11A,B) of the higher Infratatric **Inovec Nappe**; (2) Mesozoic, Jurassic and Lower Cretaceous rocks (Figure 11C–E) of the lower Infratatric **Humienec thrust sheet**, the metamorphosed fragments of which occur only in the Upper Cretaceous "flysch"; (3) the Upper Cretaceous **Couches-Rouges** type slightly metamorphosed marls to marly carbonates (Figure 11F,G) and (4) the Upper Cretaceous "**flysch**" shales and conglomerates (Figure 11H,I) with indistinct metamorphic overprinting.

The most distinct metamorphic textures were observed in the basement and cover rocks of the higher Infratatric Inovec Nappe. The more or less schistose Permian meta-basalts are composed of Ep, Czo, Chl, Act, Ab, Cal, Ttn (leucoxene), Ap, Rt-sagenite, Mag and Cel-rich Ms (Ph). They have relics of magmatic Hbl, Cpx and Cr-Spl (Figure 12A). Equivalent metamorphic textures, rich in newly formed white mica (mostly Ph), occur in Permian meta-rhyolites (Figure 12B), meta-dacites (Figure 12C), arkosic meta-sandstones (Figure 12D) and Upper Carboniferous meta-sandstones (Figure 12E).

The signatures of a low-temperature tectono-thermal overprinting are also visible in the Tatricum hanging wall granite blastomylonites according to newly formed aggregates of dynamically recrystalized Qzt, Ab, Chl and fine-grained white mica (Figure 12F).

Less distinct but clear metamorphic textures were observable in Middle to Upper Jurassic cherty slates to meta-radiolarites and Lower Cretaceous cherty–clayey slates of the reconstructed deep-water Humienec Succession. These occur as clast-to- olistolith or tectonic fragments in the Upper Campanian to Maastrichtian "flysch" and they may have been derived from the frontal lower Infratatric Humienec thrust sheet.

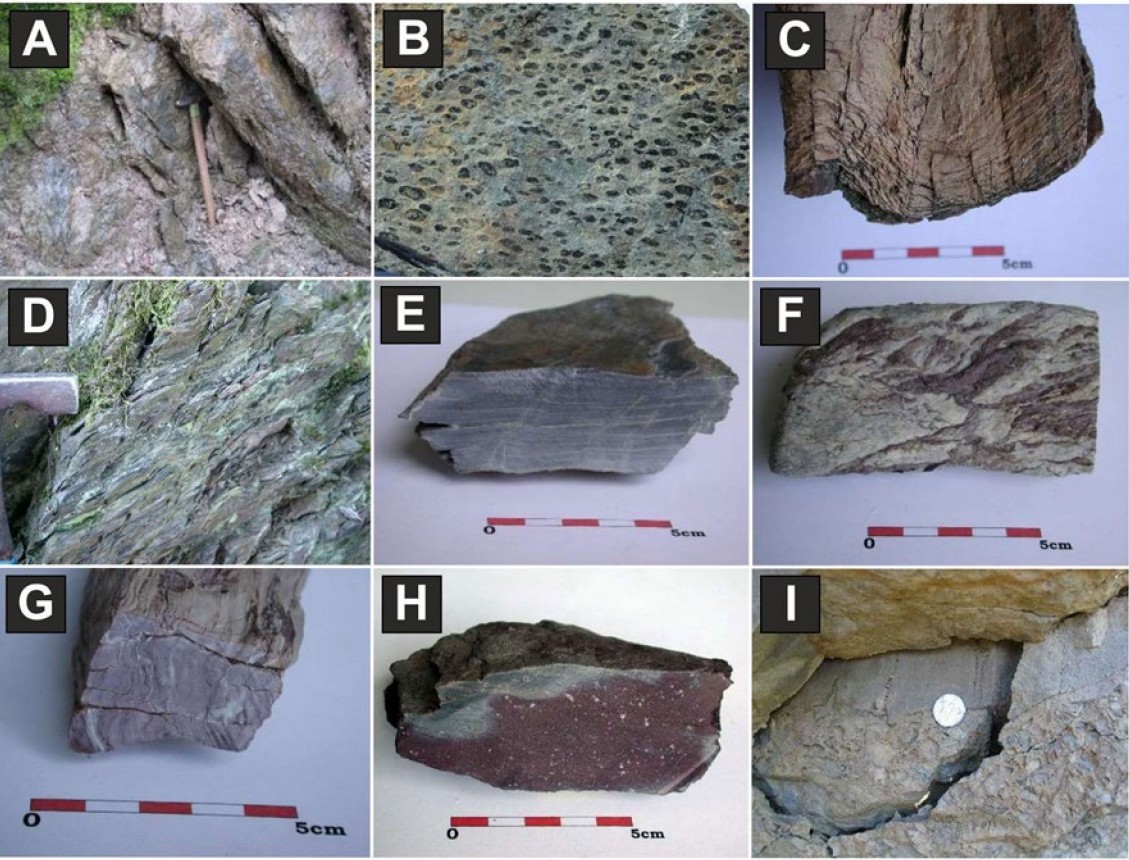

**Figure 11.** Macroscopic features of tectono-metamorphic overprinting in the Považský Inovec Mts. **Selec Block**: (**A**) Phyllonite of micaschist-gneisses in area of Humienec Hill to Krásna Valley; (**B**) Permian meta-basalt (PI-B3, Hôrka Valley). Coarse-grained Chl pseudomorphs are visible after amygdaloids (view into metamorphic schistosity plane in XY cut); (**C**) Middle to Upper Jurassic cherty slate to meta-radiolarite (PI-18, SE of Čierny Hill); (**D,E**) Lower Cretaceous cherty-clayey slate (PI-22, SE of Čierny Hill); (**F,G**) Upper Cretaceous Couches-Rouges type marly slates with isoclinal syn-metamorphic folds (F, PI-105) and crenulation cleavage, (G, PI-39, N of Humienec Hill), and (**H,I**) Upper Cretaceous "flysch" SE of Čierny Hill: reddish clayey shale (H, PI-89), sandstone and conglomerate (I, PI-17). Samples A,B from **Inovec Nappe**, and samples C–G from inferred **Humienec thrust sheet** in **"flysch"** (H,I).

The very fine-grained white mica (Ilt-Ph) aggregates define S1 metamorphic schistosity planes which are often crenulated and crosscut with the S2 cleavage planes (Figure 13A,B). Relic radiolarians are flattened and replaced by Qtz aggregates surrounded by newly

formed white-mica (Ilt-Ph) aggregates in metamorphic S1 schistosity planes (Figure 13C,D). Flattened and stretched coarse-grained Cal grains of Lower Jurassic crinoidal limestones are separated by the dynamically recrystalized Cal aggregates which contain newly formed metamorphic Ab and Qtz (Figure 13E,F).

Similar distinct metamorphic textures were found in Couches-Rouges type marly slates. The samples were taken from a tectonic slice overlying the "flysch" in the **Humienec thrust sheet** directly in the footwall of the higher Infratatric Inovec Nappe. The Cal rich layers show dynamic recrystalization and newly formed Ab (Figure 14A). Despite recrystalization and deformation, the foraminifers were preserved in some places (Figure 14B). The presence of partly regenerated clastogenic Ms is characteristic in the newly formed fine-grained white mica (Ilt-Ph) aggregates in the siliciclastic layers (Figure 14C,D).

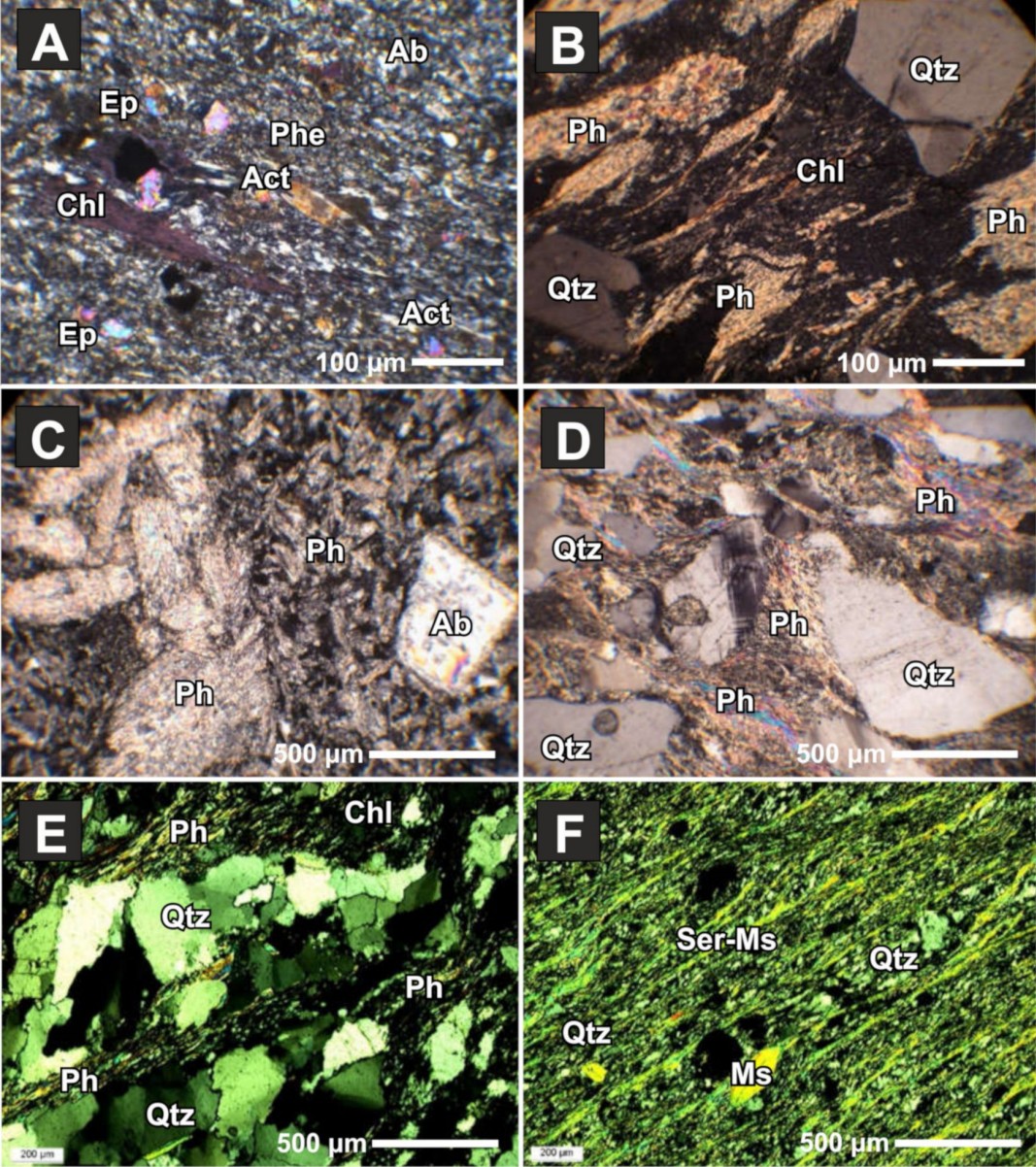

**Figure 12.** Microscopic textures from the higher **Infratatric Inovec Nappe** (**A–E**) and the hanging wall of the **Tatric Panská Javorina Nappe**: (**A**) Permian meta-basalt (PI-B3, Hôrčanská Valley); (**B**) Permian meta-rhyolite (PI-R11, Hôrčanská Valley); (**C**) Permian meta-dacite (PI-D1, NE of Inovec Hill); (**D**) Permian arkosic meta-sandstone from an olistolith in "flysch" (PI-24, N of Hradisko Hill); (**E**) Upper Carboniferous meta-sandstone (PI-5, Hôrčanská Valley); and (**F**) granite blastomylonite (PI-17HZ, NE of Panská Javorina Hill, from [99]). Asymmetric S–C microstructure indicates a dextral or top-to-the W shear along the Hrádok–Zlatníky thrust fault. All pictures at *X* N.

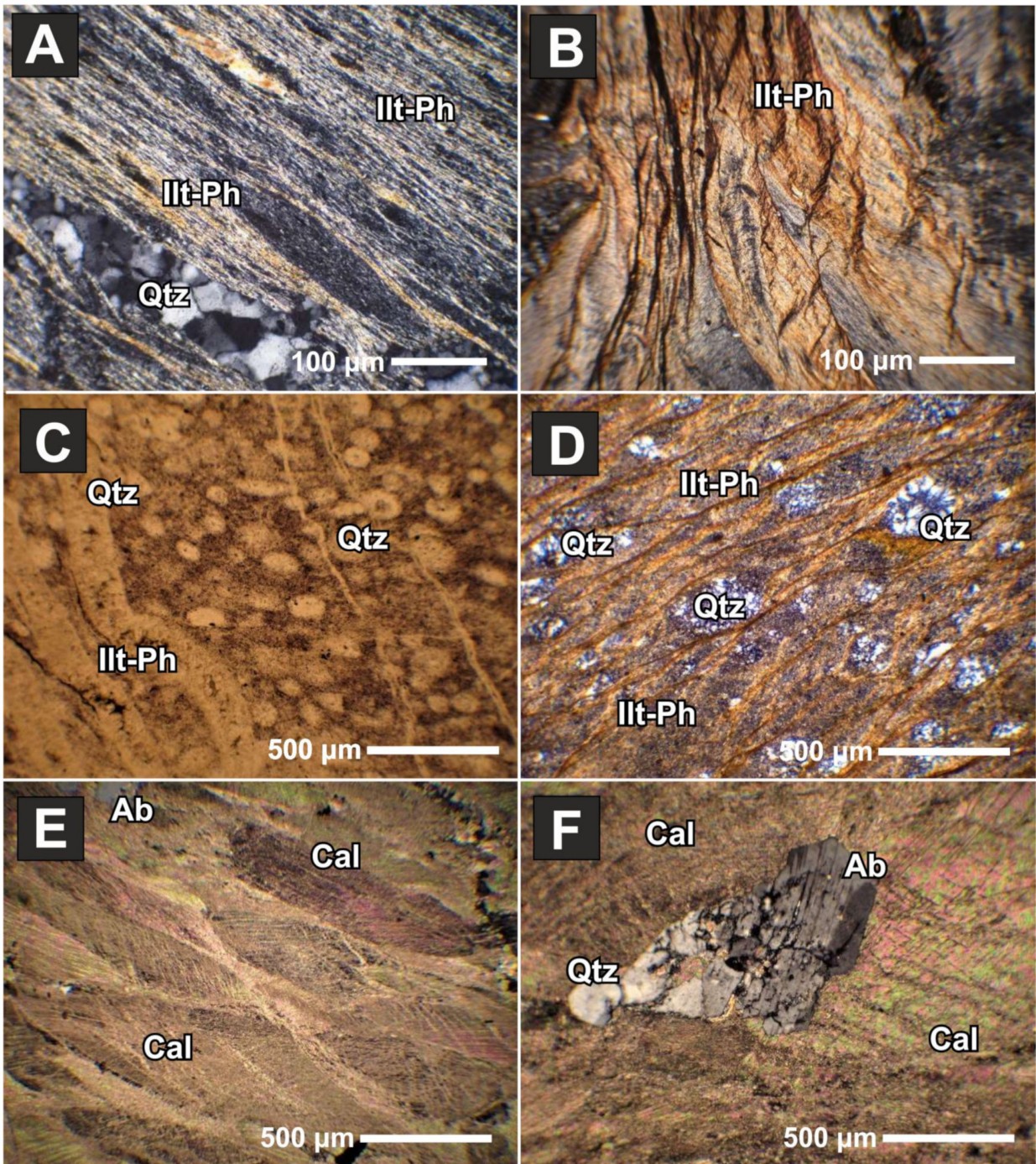

**Figure 13.** Deformational-metamorphic micro-structures of inferred **Humienec thrust sheet** clastogenic and olistolithic rock fragments in the Upper Cretaceous flysch: (**A**,**B**) Lower Cretaceous cherty-clayey slates (PI-21, PI-22, SE of Čierny Hill); (**C**,**D**) Middle to Upper Jurassic cherty slates to meta-radiolarites (PI-18 SE of Čierny Hill, PI-41 NW of Humienec Hill); and (**E**,**F**) sheared and partly dynamically recrystalized Lower Jurassic crinoidal limestones with flattened and stretched coarse Cal grains and newly formed Ab-Qtz aggregates (PI-110, below Humienec Hill). Picture C at *II* N; A, B and D–F at *X* N.

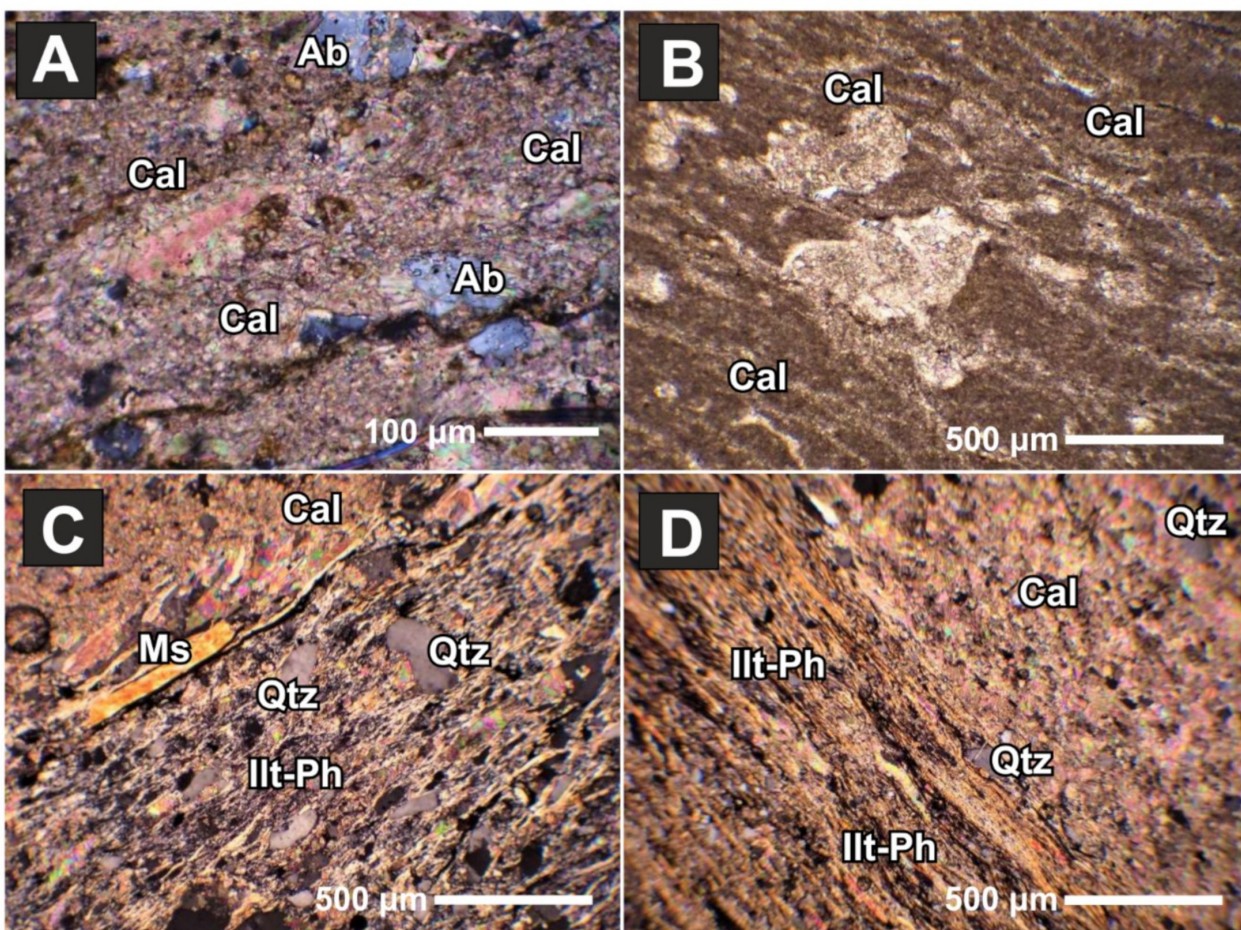

**Figure 14.** Couches-Rouges type marly slates from a tectonic slice in the **Humienec thrust sheet**: (**A**) recrystalized Cal layers with newly formed Ab (PI-105, N of Humienec Hill); (**B**) deformed and recrystalized foraminifers in cleavage planes in recrystalized Cal matrix (PI-39a, N of Humienec Hill); (**C**) alternation of carbonatic and siliciclastic layers with Ms clasts and newly formed Ilt-Ph stripes (PI-39b, N of Humienec Hill); (**D**) Ilt-Ph rich layers with relics of clastogenic Ms (PI-39c, N of Humienec Hill). Picture B at *II* N; A and C,D at *X* N.

The transitional sedimentary–metamorphic textures are typical of the Upper Cretaceous flysch sediments hosting the clast to olistolithic fragments derived from the frontal Humienec thrust sheet and the rear Inovec Nappe, respectively. The "flysch" shales, sandstones and conglomerates are weakly deformed (Figure 11H,I and Figure 15A–C). The Ms, Bt, Qtz, Kfs, Pl, Ch and Zrn clastogenic material in shales, sandstones and conglomerates is well preserved; for example, Bt is pleochroic and not chloritized (Figure 15C). Rarely, fold-cleavage structure formed (Figure 15D). Calciclastic layers contain carbonatic clasts of variable grain-size and the well preserved clastogenic texture shows negligible Cal recrystalization (Figure 15D). Newly formed very fine-grained white mica (Ilt to Ilt-Ph) aggregates are rarely observable (Figure 15E). The "flysch" sandstone layers contain pebbles of micaschists and Middle-Upper Jurassic to Lower Cretaceous slates (Figure 15F).

**BSE images, mineral chemistry and P–T estimates**

BSE images of investigated rock textures (Figure 16) document newly formed metamorphic phases which were used for the P–T estimates from the higher Infratatric **Inovec Nappe** and the lower Infratatric **Humienec thrust sheet**.

The higher Infratatric **Inovec Nappe** Permian arkosic meta-sandstones (PI-2, PI-4, PI-7, PI-24) show newly formed fine-grained aggregates of Ph (±Chl) and relics of clastogenic coarse-grained Qtz, Fsp and Ms (Figure 16A,B). Permian meta-basalts consist of Chl, Act, Ab, Ep–Czo, ±Ph, Qtz, Cal and relics of magmatic Hbl and Spl (Figure 16C). Permian meta-rhyolites contain Pl, Kfs, Qtz and chloritized Bt porphyroclasts surrounded by newly

formed Ph–Chl aggregates (Figure 16D). Upper Carboniferous meta-sandstones contain newly formed Ph, Chl and Ab, but also Qtz, Fsp and Ms porphyroclasts (Figure 16E).

Lower Cretaceous cherty–clayey slates of the lower Infratatric **Humienec thrust sheet** show crenulated S1 schistosity planes crosscut by S2 syn-metamorphic cleavage planes, and both plane systems are defined by newly formed Ilt-Ph and Chl fine-grained aggregates (Figure 16F).

Upper Cretaceous flysch shales and sandstones of the **Humienec thrust sheet** show negligible metamorphic overprinting compared to the rocks of the higher Infratatric In-ovec Nappe or the Jurassic–Lower Cretaceous clastogenic to olistolith fragments of the Humienec thrust sheet within the "flysch" matrix. There are observable mico-crenulated shales in some places, and these are composed of very fine-grained white-mica (Ilt to Ilt-Ph) aggregates (Figure 16G). Well-preserved sedimentary structures of calciclastic and siliciclastic layers with Ms, Chl, Bt, Qtz, Fsp, Cal and Dol porphyroclasts are typical (Figure 16H).

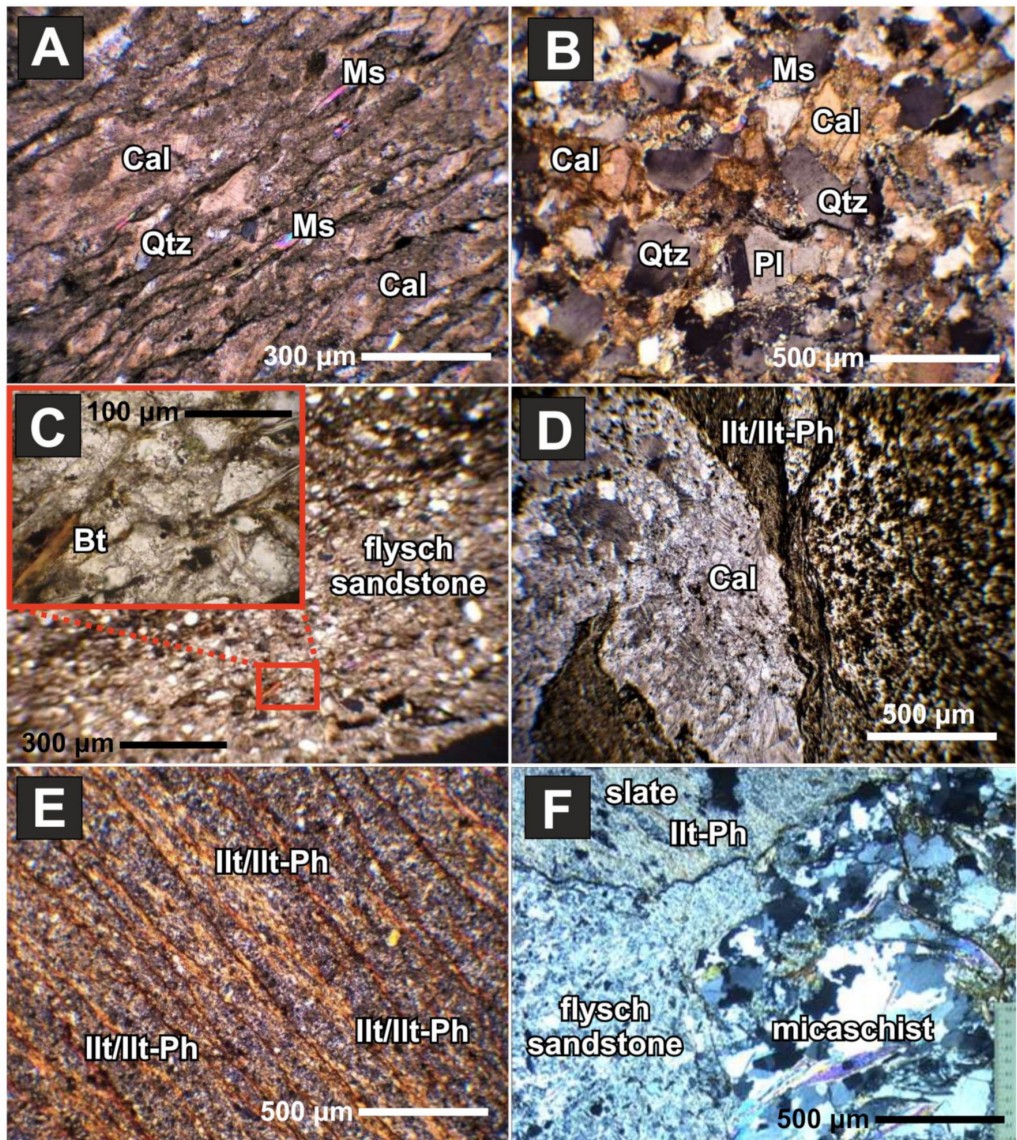

**Figure 15.** Rock microstructures of Upper Cretaceous **"flysch"** sediments: (**A**) calciclastic sandstone (PI-89, SE of Čierny Hill); (**B**) siliciclastic–calciclastic sandstone (PI-19, SE of Čierny Hill); (**C**) sandstones with well-preserved clastogenic material, including Bt (PI-88b, SE of Čierny Hill); (**D**) folded calciclastic layers with clastogenic texture show weak Cal recrystalization (PI-33, N of Hradisko Hill); (**E**) shale with newly formed Ilt to Ilt-Ph (PI-88a, SE of Čierny Hill); (**F**) sandstone matrix (PI-88c, SE of Čierny Hill) with pebbles of micaschists and Lower Cretaceous(?) cherty-clayey slates. Pictures C,D at *II* N; A,B and E,F at *X* N.

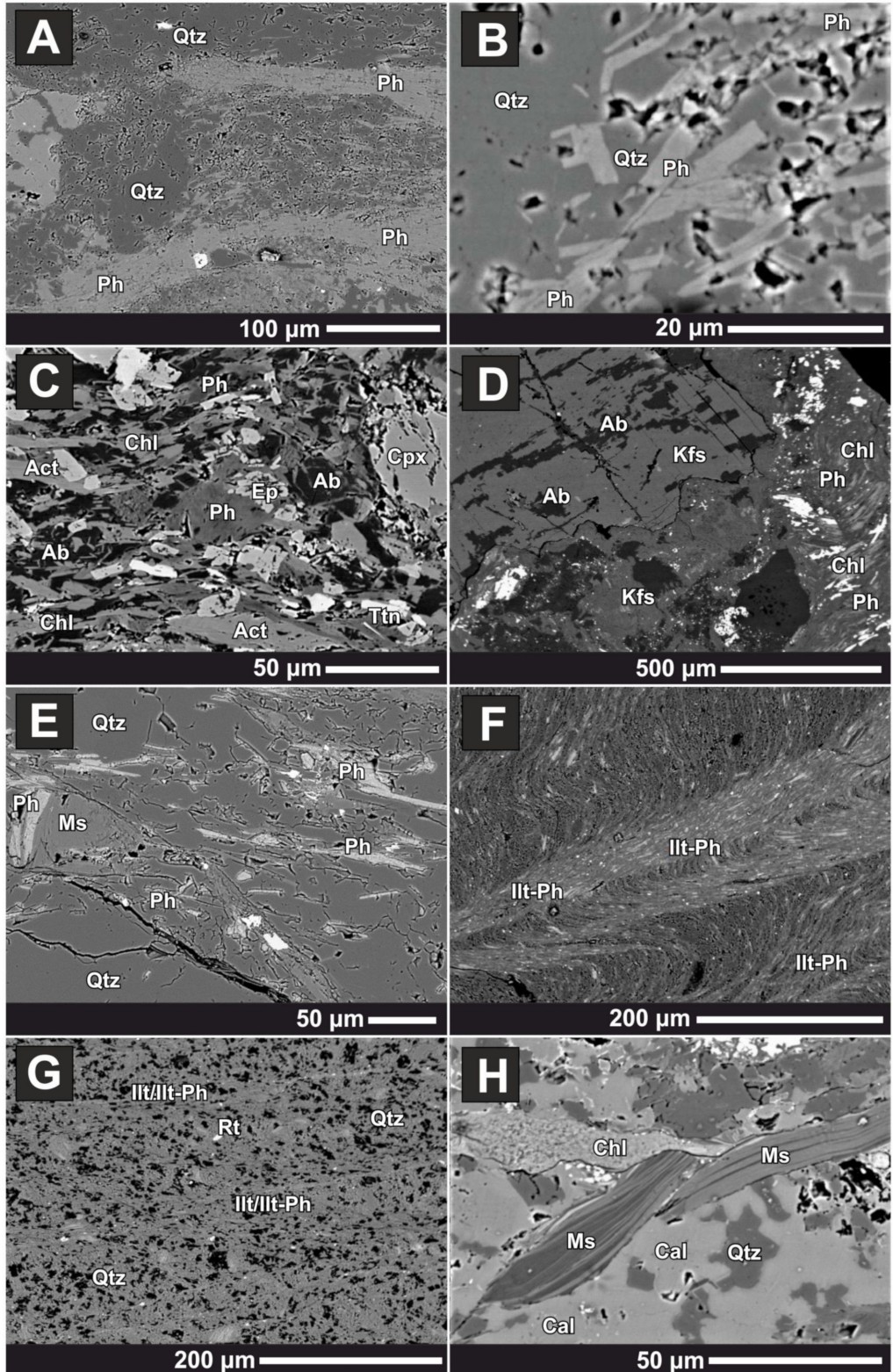

**Figure 16.** Examples of BSE images of white mica aggregates (±Chl) from the Infratatric rocks the Považský Inovec Mts. **Selec Block**: (**A**,**B**) Permian meta-sandstone (PI-2, Hôrčanská Valley); (**C**) Permian Act meta-basalt (PI-B3, Hôrčanská Valley); (**D**) Permian meta-rhyolite (PI-R11, northern branch of Hôrčanská Valley); (**E**) Upper Carboniferous meta-sandstone (PI-5, Hôrčanská Valley); (**F**) Lower Cretaceous cherty–clayey slate (PI-22, SE of Čierny Hill); (**G**) Upper Cretaceous "flysch" shale (PI-88, SE of Čierny Hill); (**H**) Upper Cretaceous "flysch" sandstone (PI-136, SE of Humienec Hill). Samples A–E from **Inovec Nappe**, and sample F from lower Infratatric **Humienec thrust sheet** in "flysch" (**G**,**H**).

The results of geothermobarometric methods are based on metamorphic Chl and Ph or Chl–Ph pairs. The chemical composition of minerals including those used in Perple_X pseudosection modeling are available in Table S3.

Chlorite in the Permian meta-basalts, meta-dacites and meta-sandstones of the higher Infratatric **Inovec Nappe** has a composition of clinochlore, while Chl from meta-rhyolites is chamosite in the classification diagram (Figure 17). Mg–Chl (clinochlore) has a composition of 22.2–24.0 wt.% FeO and 16.4–18.8 wt.% MgO with a Fe/(Fe + Mg) ratio 0.28–0.31. Fe–Chl (chamosite) from meta-rhyolites contains 27.6 wt.% FeO and 11.6 wt.% MgO with Fe/(Fe + Mg) ratio 0.57. The $^{VI}$Al content reaches 1.14–1.47 *p.f.u.* in all studied samples from the Infratatric Inovec Nappe (Table S3). Chlorite from the lower Infratatric Humienec thrust sheet was not identified.

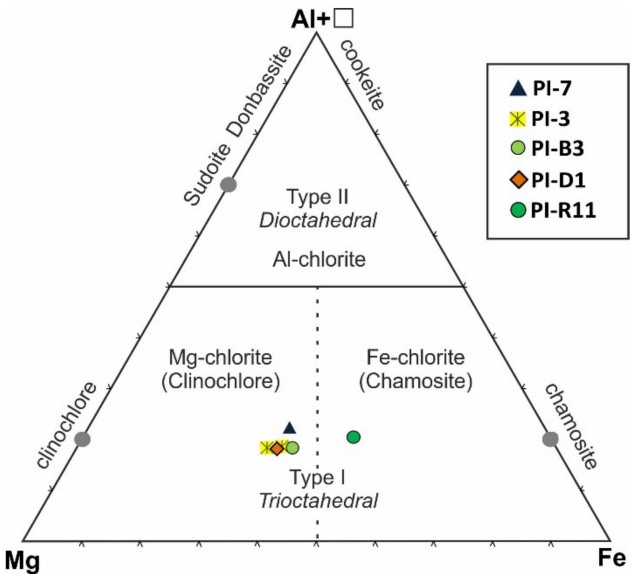

**Figure 17.** Al + □–Mg–Fe classification diagram of chlorite after [84] from the higher Infratatric **Inovec Nappe** Permian samples: PI-3, PI-B3 meta-basalt, PI-R11 meta-rhyolite, PI-D1 meta-dacite, PI-7 meta-sandstone.

The metamorphosed basement and cover rocks of the **higher Infratatric Inovec Nappe** contain Cel-Ms (Ph). All chemical analyses of the white mica project along the muscovite–celadonite mixing-line in the classification diagram (Figure 18A). White mica from higher Infratatric Inovec Nappe rocks shows $SiO_2$ content from 47.9 to 51.5 wt.% (3.21–3.44 *p.f.u.*). FeO and MgO vary between 1.5–6.1 and 1.0–3.2 wt.%, respectively, with the highest values linked to meta-basalts and the lowest to meta-sandstones. $TiO_2$ content is generally low; up to 0.6 wt.%. Permian meta-basalts and meta-rhyolites (PI-B3, PI-R11) contain higher alkalis (0.9–1.0 K + Na *p.f.u.*) and 9–10 wt.% $K_2O$ (Figure 18B, Table S3). The newly formed Ilt-Ph metamorphic aggregate in Upper Carboniferous to Permian meta-sandstones show relatively variable $K_2O$ contents between 9.4 and 11.0 wt.% (0.8–1.0 K + Na *p.f.u.*).

The Ilt-Ph metamorphic aggregate in Jurassic–Cretaceous slates of the inferred **Humienec thrust sheet** (PI-21, PI-22) has a relatively homogeneous composition. The variable $K_2O$ between 8.7 and 9.0 wt.% in all samples documents transitional Ilt-Ph composition with slightly decreased Na + K and $K_2O$. The presence of Ilt-Ph (3.3–3.4 Si *p.f.u.*) with 0.7–0.8 *p.f.u.* K + Na values and 8–9 wt.% $K_2O$ is consistent with the lower-T anchi-metamorphic conditions in Figure 18B.

Similarly, the Upper Cretaceous Couches Rouges type marly slates (PI-39) has Ilt-Ph (3.4 Si *p.f.u.*) with 0.6–0.7 *p.f.u.* K + Na values and 7–8 wt.% $K_2O$ (Figure 18B; Table S3). This is consistent with lower-T anchi-metamorphic conditions.

"Flysch" sediments (PI-88) contain newly formed white micas (3.1–3.2 Si *p.f.u.*) of Ilt to Ilt-Ph composition with much lower alkali content (0.5–0.7 *p.f.u.*) and $K_2O$ (5–7 wt.%) than in the Inovec Nappe-type basement rocks, including the Lower Cretaceous slates. Figure 18B and Table S3 document these low values typical of advanced diagenesis and/or lowest-T anchi-metamorphism at approximately 150–200 °C.

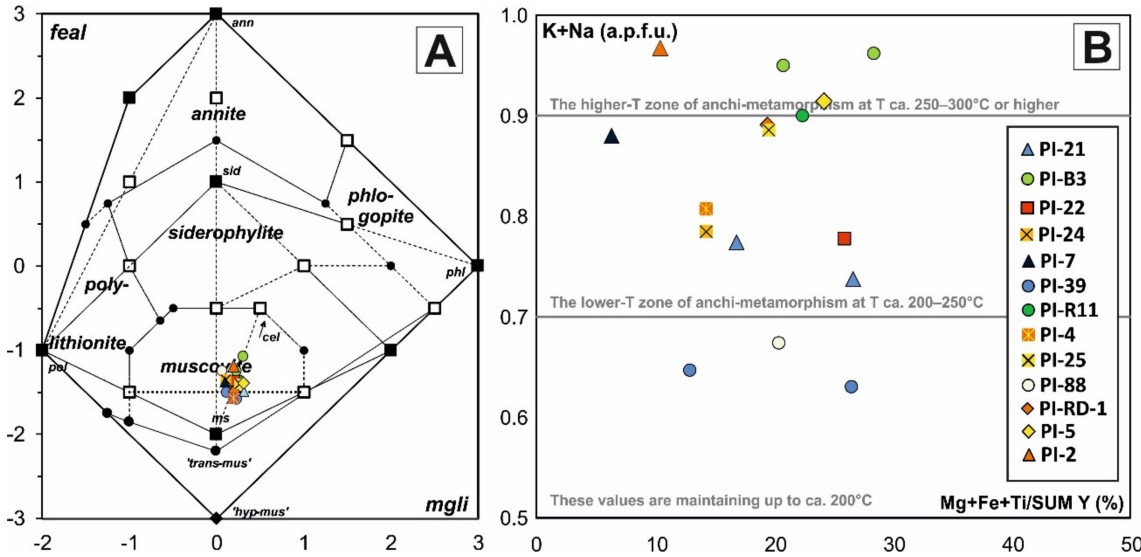

**Figure 18.** Classification diagram of white micas from the higher Infratatric **Inovec Nappe** (PI-2, PI-4, PI-5, PI-7, PI-B3, PI-R11, PI-D1), and from lower Infratatric **Humienec thrust sheet** (PI-21, PI-22, PI-24, PI-25, PI-39 and PI-88). (**A**) after [85]. Parameters represent: *mgli* = Mg-Li, *feal* = $(Fe^{2+} + Fe^{3+} + Mn + Ti) - {}^{VI}Al^{3+}$. (**B**) K + Na vs. Mg + Fe + Ti/SUM Y (%) compositional trend of white micas. Values for lines taken from the works of [69,103,104].

Amphiboles with Act composition were identified in meta-basalts (PI-B3) with a $SiO_2$ content up to 48.5–54.8 wt.%. The content of $Al_2O_3$ and MgO in Act varies between 0.7–5.9 wt.%. and 14.7–18.2 wt.%, respectively. CaO and $Na_2O$ content is generally low up to 12.5 and 2.8 wt.%, respectively. FeO varies between 9.2 and 13.9 wt.%. Distinct core to rim crystal zoning is visible in the bigger grains with commonly higher contents of $Al_2O_3$, MgO, $TiO_2$ and $Na_2O$ and lower contents of FeO and $SiO_2$ in the core. Plagioclase of investigated samples corresponds to Ab.

The geothermometry applied to Permian meta-basalts and meta-rhyolites (PI-B3, PI-R11) of the **Inovec Nappe** gives an average Chl crystalization temperature of 277 and 285 °C, respectively. Slightly higher average temperature of 303 °C was obtained for meta-dacite sample (PI-D1) (Table 2). Permian meta-sandstone (PI-7) gives average Chl temperature of 309 °C.

The newly formed Ilt-Ph metamorphic aggregate in the Jurassic–Cretaceous slates of the inferred **Humienec thrust sheet** (PI-21, PI-22) yields 600 MPa metamorphic pressure at the estimated 250 °C temperature. Slightly higher pressures were estimated for the Upper Cretaceous Couches Rouges type marly slates (PI-39), with metamorphic pressure at 750 MPa by Si-in-Ph geobarometry [72] (Figure 19, Table 2).

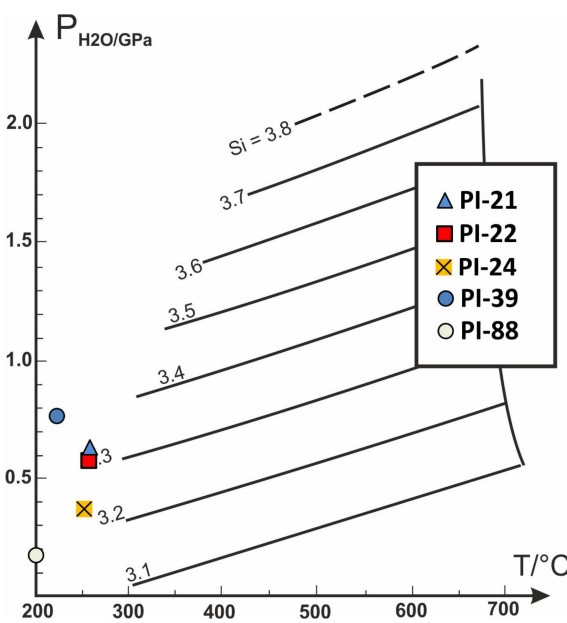

**Figure 19.** Pressure estimates (diagram after [72]) from lower Infratatric **Humienec thrust sheet** (PI-21, PI-22, PI-24 and PI-39) in flysch (PI-88). Average temperatures derived from Chl thermometry [70,71,73] or estimated from mechanics of quartz dynamic recrystalization [105] in case of missing Chl in sample.

Figure 20 highlights that the inferred peak metamorphic P–T conditions of Permian meta-dacites and meta-sandstones of higher Infratatric **Inovec Nappe** accord with Chl–Ph pairs at 323 °C at 600 MPa and 290 °C at 400 MPa (Table 2). This was accomplished with multi-equilibria Chl–Ph–Qtz–$H_2O$ geothermobarometry [73].

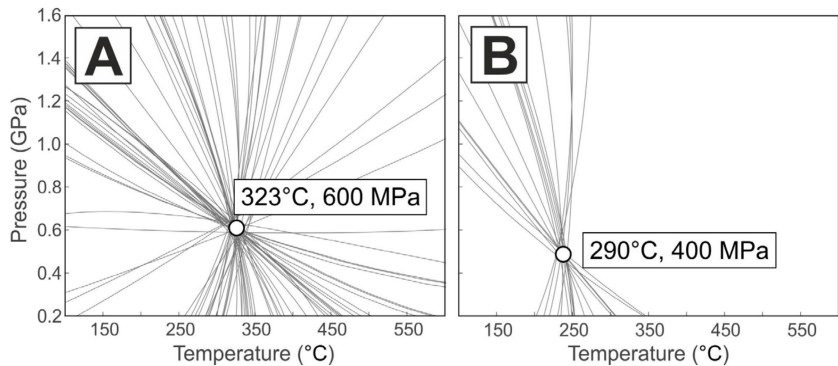

**Figure 20.** P–T diagrams with univariant curves and points intersecting with Chl–Ph pairs at equilibrium P–T conditions 323 °C at 600 MPa and 290 °C at 400 MPa. (**A**) Permian meta-dacite (PI-D1); (**B**) Permian meta-sandstone (PI-7). Higher Infatatric **Inovec Nappe**. Based on Chl–Ph–Qtz–$H_2O$ geothermobarometry [73].

**Perple_X pseudosection modeling** was used to estimate D1 stage P–T conditions of the Permian meta-basalts in the greenschist-facies metamorphic overprinting at 330–345 °C and 460–580 MPa pressure (Figure 21, Table 2). These estimates also apply to the meta-basalt olistoliths of the higher Infratatric Inovec Nappe in the Upper Cretaceous "flysch", and they agree with the results obtained by conventional Chl and Ph geothermometry within the given error.

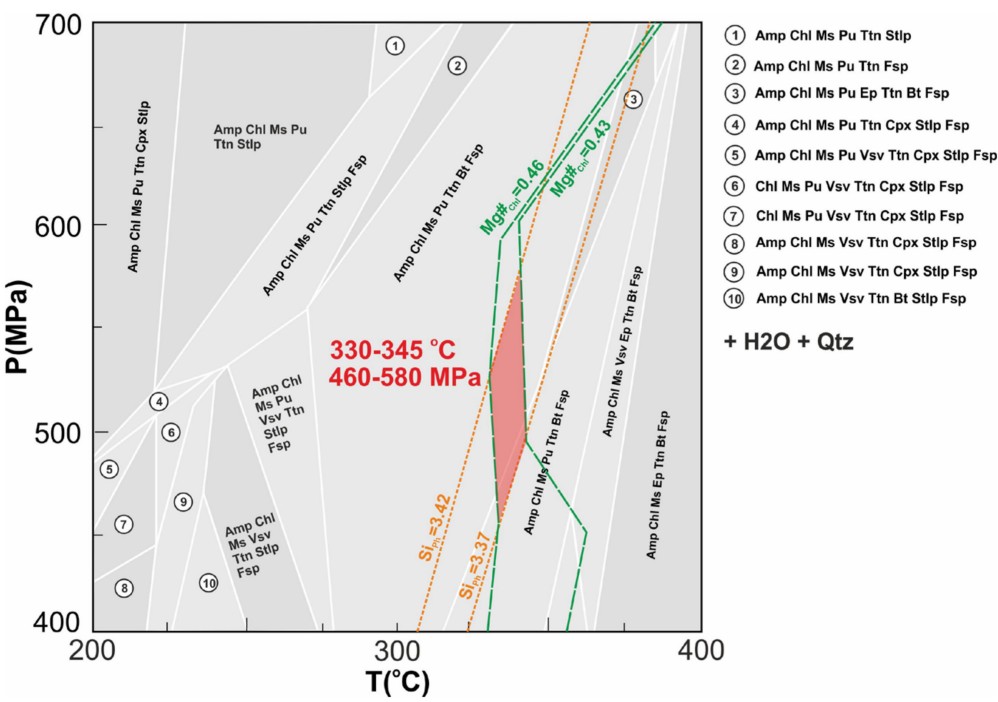

**Figure 21.** P–T pseudosection calculated by Perple_X thermodynamic software [92,93] for the D1 peak metamorphic stage of Permian meta-basalt (PI-B3) of the higher Infratatric **Inovec Nappe**, using isopleths intervals of Si-in-Ph 3.37–3.42 *p.f.u.* and Chl$_{Mg\#}$ 0.43–0.46. Whole-rock composition used for modeling can be found in Table S1.

**Table 2.** Estimated P–T conditions from the Považský Inovec Mts. **Selec Block**.

| Sample | Sample Description GPS Coordinates | Chl[C] T (°C) ±50 °C | Chl[J] | Chl[V] | T* (°C) | Ph[M] P (MPa) ±200 MPa | Chl-Ms[V] T (°C) ±50 °C | P (MPa) ±200 MPa | Perple_X T (°C) ±50 °C | P (MPa) ±100 MPa |
|---|---|---|---|---|---|---|---|---|---|---|
| | | | | | **Higher Infatatric Inovec Nappe** | | | | | |
| PI-7 | Permian meta-sandstone N 48°43.726′ E 17°56.606′ | 329 | 332 | 283 | 309 | - | 290 | 400 | - | - |
| PI-B3 | Permian meta-basalt N 48°42.46′ E 17°56.142′ | 284 | 288 | 259 | 277 | - | - | - | 330–345 | 460–580 |
| PI-R11 | Permian meta-rhyolite N 48°43.021′ E 17°57.406′ | 296 | 288 | 259 | 285 | - | - | - | - | - |
| PI-D1 | Permian meta-dacite N 48°47.578′ E 18°3.182′ | 283 | 287 | 318 | 303 | - | 323 | 600 | - | - |
| | | | | | **Lower Infratatric Humienec Thrust Sheet** | | | | | |
| PI-39 | Upper Cretaceous Couches-Rouges type marly slate clayey-rich layer N 48°48.95′ E 18°3.823′ | - | - | - | 220 | 750 | - | - | - | - |
| PI-88 | Upper Cretaceous "flysch" shale N 48°47.128′ E 18°1.972′ | - | - | - | 200 | 200 | - | - | - | - |
| PI-24 | Permian meta-sandstone olistolith in "flysch" N 48°47.305′ E 18°0.759′ | - | - | - | 250 | 400 | - | - | - | - |
| PI-21 | Middle Jurassic cherty slate in "flysch" N 48°47.349′ E 18°1.887′ | - | - | - | 250 | 600 | - | - | - | - |
| PI-22 | Lower Cretaceous clayey slate in "flysch" N 48°47.356′ E 18°1.938′ | - | - | - | | 600 | - | - | - | - |

Chlorite thermometry by [70]—(C), [71]—(J). Si-in-Ph barometry after [72]—(M) based on average Si-in-Ph content. Chl thermometry and Chl–Ms thermobarometry after [73]—(V). *—Average of calculated T or estimated T from mechanics of quartz dynamic recrystalization [105] in case of missing Chl.

Považský Inovec Mts. Hlohovec Block (Hlohovec Nappe and Jašter Thrust Sheet)

**Petrography**

Our research focused on the metamorphic overprinting textures in: (1) the granitoid basement meta-aplites (Figure 22A); (2) Lower Triassic arkosic meta-sandstones (Figure 22B), Lower Triassic meta-quartzites (Figure 22C), Middle Triassic marbles (Figure 22D,E), Lower Jurassic crinoidal limestones—marbles (Figure 22F) and inferred Lower Cretaceous marly marbles (Figure 22G); (3) Upper Cretaceous Couches Rouges type marly slates (Figure 22H) and (4) the Upper Cretaceous "flysch" sandy shales with indistinct metamorphic overprinting (similar to Figure 15).

The most distinct metamorphic textures were observed in the **higher Infratatric Hlohovec Nappe** basement and cover rocks. The meta-aplite vein in granodiorite to tonalite is weakly deformed, with book-shelf structures of chloritized Bt, recrystalized Qtz and albitized feldspars (Figure 22A). Newly formed white mica aggregates (Ph) occur in Lower Triassic meta-arkoses (Figure 22B) and meta-quartzites (Figure 22C). The Middle Triassic marble mylonites to ultramylonites contain very fine-grained white-mica (Ph) and partly recrystalized Dol porphyroclasts (Figure 22D,E). The microfold core of inferred Lower Cretaceous marly marble contains white mica (Ph)-rich layers alternating with Cal layers, where the mechanical twins are oriented in axial-plane cleavage (Figure 22G).

Couches-Rouges type marly slates of the **lower Infratatric Jašter thrust sheet** contain fine-grained siliciclastic layers composed of fine-grained aggregates of newly formed white mica (Ilt-Ph) and rarely Chl (Figure 22H).

**BSE images, mineral chemistry and P–T estimates**

BSE images of the investigated rock textures (Figure 23) document the newly formed metamorphic phases used in P–T estimates.

The higher Infratatric **Hlohovec Nappe** Lower Triassic arkosic meta-sandstones show newly formed fine-grained aggregates of Ph and relics of clastogenic coarse-grained Ms (Figure 23A). Lower Triassic meta-quartzite contains newly formed Ph, but also Ms clasts recrystalized into Ph + Pg aggregates (Figure 23B). Middle Triassic marble mylonites contain coarse-grained Ph1 in the first S1 metamorphic schistosity planes. These planes with the segmented Ph1 are rotated in shear bands and crosscut by the mylonitic shear (C) planes defined with the recrystalized fine-grained Ph2 aggregates (Figure 23C). Inferred Lower Cretaceous marly marbles are rich in one generation of metamorphic Ph aggregates with tiny relics of Ms. Fold-cleavage structure is characteristic (Figure 23D).

Upper Cretaceous Couches Rouges marly slates of the lower Infratatric **Jašter thrust sheet** show less distinct metamorphic overprinting than the higher Infratatric Hlohovec Nappe rock group. However, the secondary very fine-grained white mica generation of the Hlohovec Nappe hanging wall marble mylonites appears comparable to the newly formed white micas of the underlying Jašter thrust sheet Couches-Rouges marly slates (Figure 23E–G).

Carbonatic sandstones of the Upper Cretaceous **"flysch"** succession show neither macroscopic nor microscopic signatures of tectono-metamorphic overprinting (Figure 23H).

The geothermobarometric methods are applied to metamorphic Chl and Ph or Chl–Ph, and Ph–Pg pairs. The chemical analyses of representative minerals from the higher Infratatric Hlohovec Nappe and the lower Infratatric Jašter thrust sheet of the Hlohovec Block are available in Table S4.

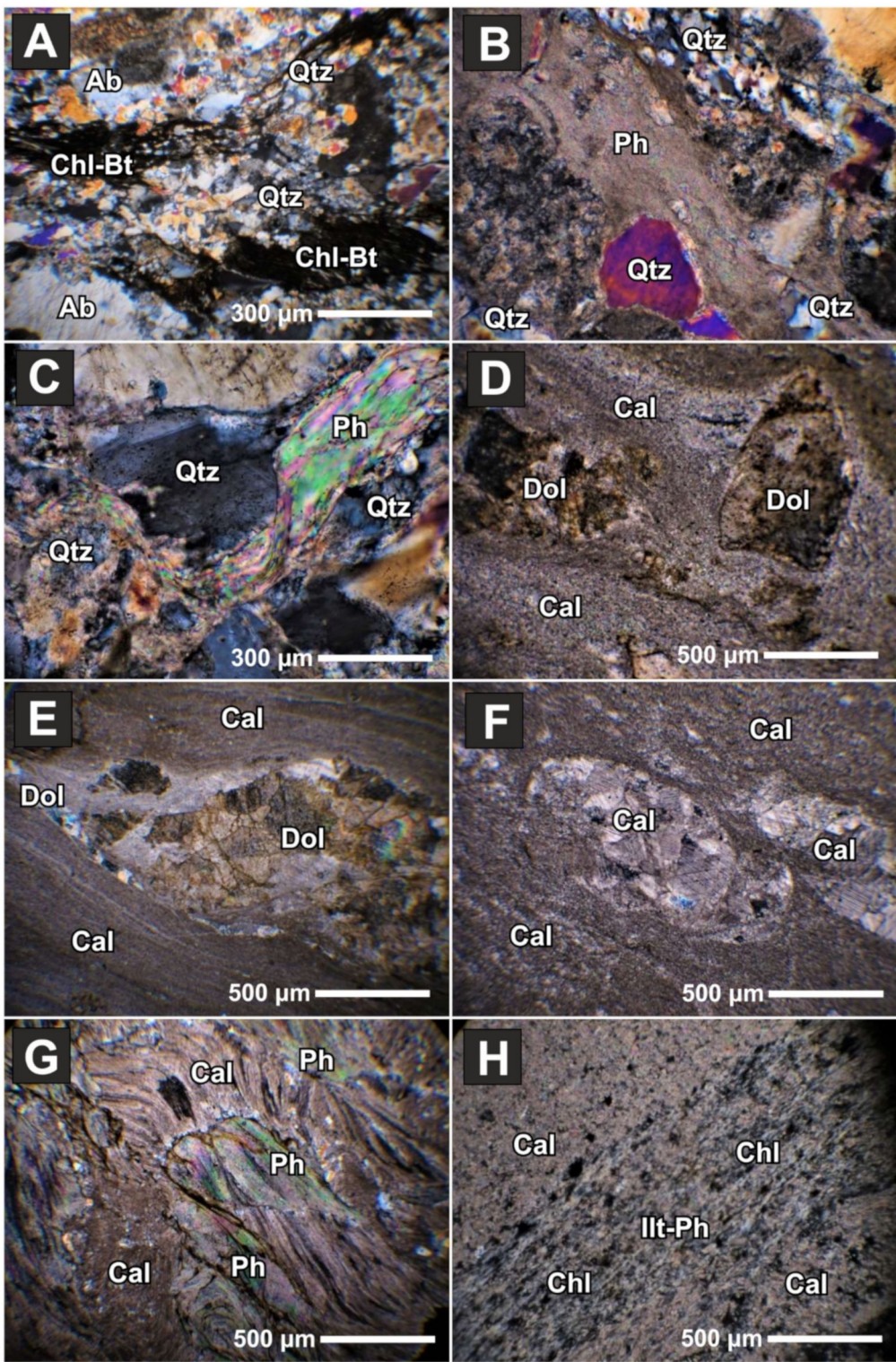

**Figure 22.** Macroscopic features of tectono-metamorphic overprinting in the Považský Inovec Mts. **Hlohovec Block**: (**A**) granitic basement meta-aplite proto-mylonite (HC-19, at Hlohovec); (**B**) Lower Triassic meta-arkose (HC-15, at Hlohovec); (**C**) Lower Triassic meta-quartzite (HC-14, at Hlohovec); (**D,E**) Middle Triassic mylonitic marbles with Dol porphyroclasts (SRB-1 dated sample, Soroš quarry at Hlohovec); (**F**) Lower Jurassic crinoidal limestone-marble (HC-6, at Hlohovec); (**G**) Lower Cretaceous marly marble (HC-12 dated sample, at Hlohovec); (**H**) Upper Cretaceous Couches-Rouges type marly slate (JS-4, at Hlohovec-Jašter). A–G samples from the higher Infratatric **Hlohovec Nappe**. H sample from the lower Infratatric **Jašter thrust sheet**. All pictures at *X* N.

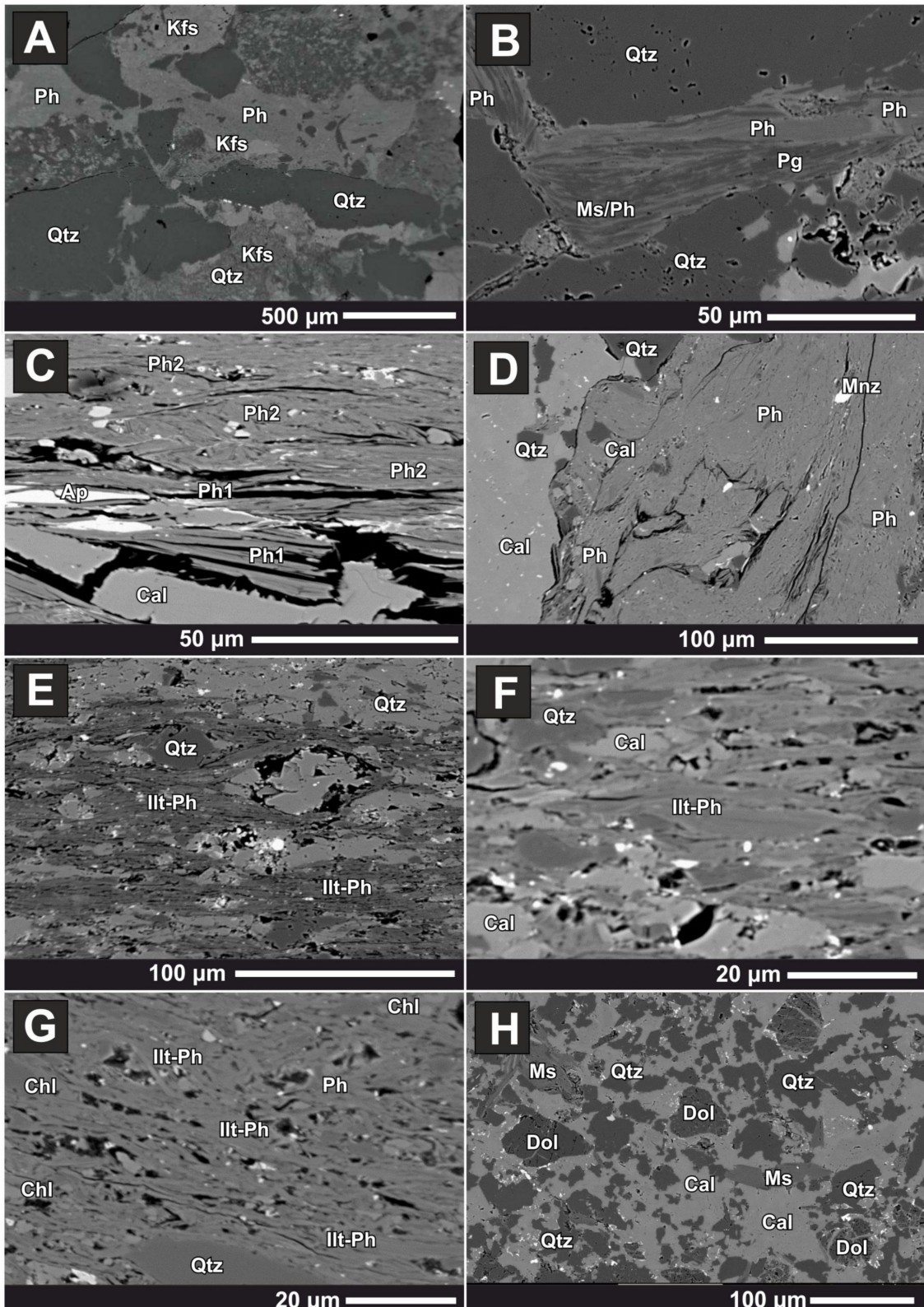

**Figure 23.** Examples of BSE images of white mica aggregates (±Chl) from the Infratatric rocks the Považský Inovec Mts. **Hlohovec Block**: (**A**) Lower Triassic meta-sandstone (HC-15, at Hlohovec); (**B**) Lower Triassic meta-quartzite (HC-14, at Hlohovec); (**C**) Middle Triassic marble mylonite with two Ph generations (SRB-1 dated sample, Soroš quarry at Hlohovec); (**D**) Lower Cretaceous marly marbles (HC-12 dated sample, at Hlohovec); (**E–G**) Upper Cretaceous Couches Rouges marly slates (HC-2, JS-2, JS-4, Hlohovec-Jašter); (**H**) Upper Cretaceous "flysch" carbonatic sandstone (JS-5, Hlohovec-Jašter).

Chlorite from the **Jašter thrust-sheet** Upper Cretaceous Couches Rouges type marly slates (JS-2) is mostly clinochlore in the classification diagram (Figure 24). Chlorites contain up to 10.9–24.2 wt.% FeO and 14.0–19.3 wt.% MgO. The Fe/(Fe + Mg) ratio is 0.24–0.52. The f $^{VI}$Al content reaches 1.18–1.81 *p.f.u.* in the studied sample (Table S4). Chlorite from the Hlohovec Nappe was not identified.

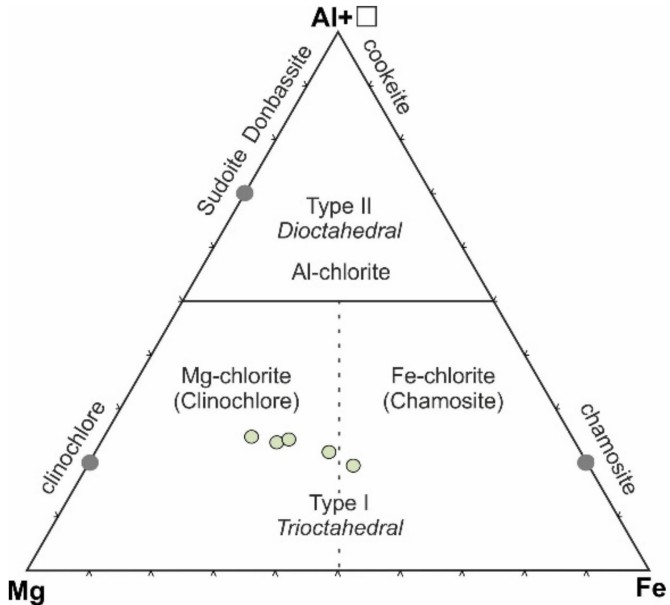

**Figure 24.** Al + □–Mg–Fe classification diagram of chlorite after [84] from the lower Infratatric **Jašter thrust sheet** Upper Cretaceous Couches Rouges type marly slates (JS-2).

The metamorphosed basement and cover rocks of the **Hlohovec Nappe** contain Ph. Figure 25A highlights that all white mica compositions of these rocks project along the muscovite–celadonite mixing-line in the classification diagram. White micas from the Lower Triassic meta-sandstones and meta-quartzites and Middle Triassic to Lower Cretaceous(?) marbles have $SiO_2$ content from 46.0 to 51.5 wt.% (3.22–3.46 Si *p.f.u.*). FeO content in the samples varies between 1.9 and 4.1 wt.%, except in the Lower Cretaceous marble (HC-12) which has increased content between 4.4 and 6.2 wt.%. The white mica in granitoid basement meta-aplite generally has lower $SiO_2$ values between 45.5 and 49.3 wt.% (3.17–3.34 Si *p.f.u.*) with higher FeO values of 4.0–6.2 wt.% and MgO up to 3.5 wt.%. $TiO_2$ content is generally low; up to 0.8 wt.% in all samples. Analyses of the phengitic white mica in Figure 25B mostly fall within the higher anchi-metamorphism to lower greenschist facies fields with alkalis content of 0.9–1.0 K + Na *p.f.u.* and $K_2O$ up to 8.7–11.6 wt.%. Exceptions and marbles (SRB-1) cross both the lower and higher-T field with 0.77–0.95 K + Na *p.f.u.* This diagram compositionally displays two Ph generations in sample SRB-1 (Figure 23C), and this is consistent with petrographic observations.

The metamorphosed rocks of the **Jašter thrust sheet** contain Ilt-Ph. The compositions of the white micas in these rocks project along the muscovite–celadonite mixing-line in the classification diagram (Figure 26A). $SiO_2$ content in white micas from the Upper Cretaceous Couches-Rouges marly slates varies from 46.7 to 52.3 wt.% (3.22–3.45 Si *p.f.u.*). MgO and FeO content is 2.0–3.6 and 1.3–5.0 wt.%, respectively. Upper Cretaceous "flysch" sandstone (JS-5) contains fine-detritic Ms with Si content up to 51.6 wt.% and rather low MgO and FeO content up to 1.8 and 1.6 wt.%, respectively. $TiO_2$ content is up to 0.8 wt.% in all samples (Table S4). Analyses of the **Jašter thrust-sheet** phengitic white mica mostly show values between 0.7 and 0.9 K + Na *p.f.u.* in Figure 26B, typical of lower anchi-metamorphism.

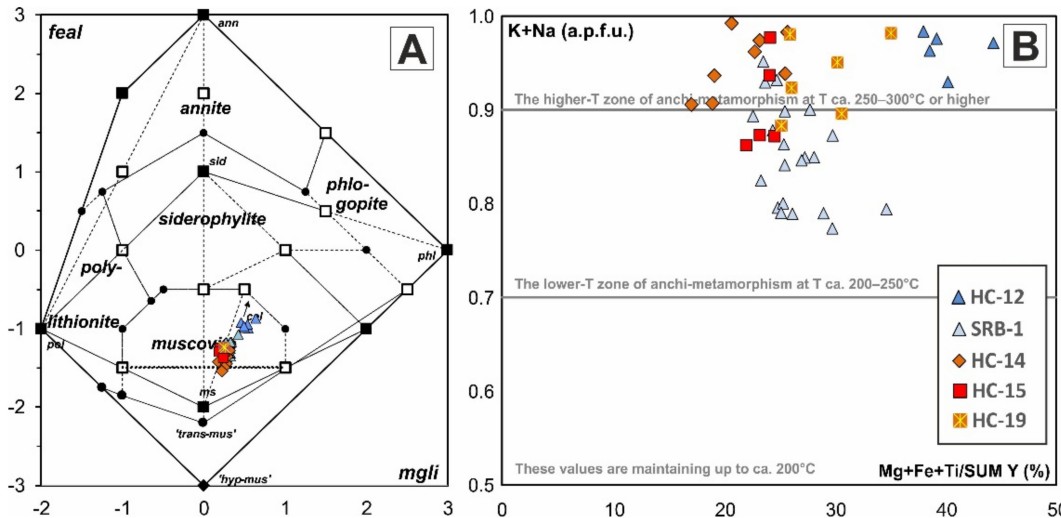

**Figure 25.** Classification diagram of white micas [85] from the Higher Infratatric **Hlohovec Nappe**: (**A**) parameters: *mgli* = Mg-Li, *feal* = (Fe$^{2+}$ + Fe$^{3+}$ + Mn + Ti) − $^{VI}$Al$^{3+}$; (**B**) K + Na vs. Mg + Fe + Ti/SUM Y (%) compositional trend of white micas. Values for lines taken from the works of [69,103,104].

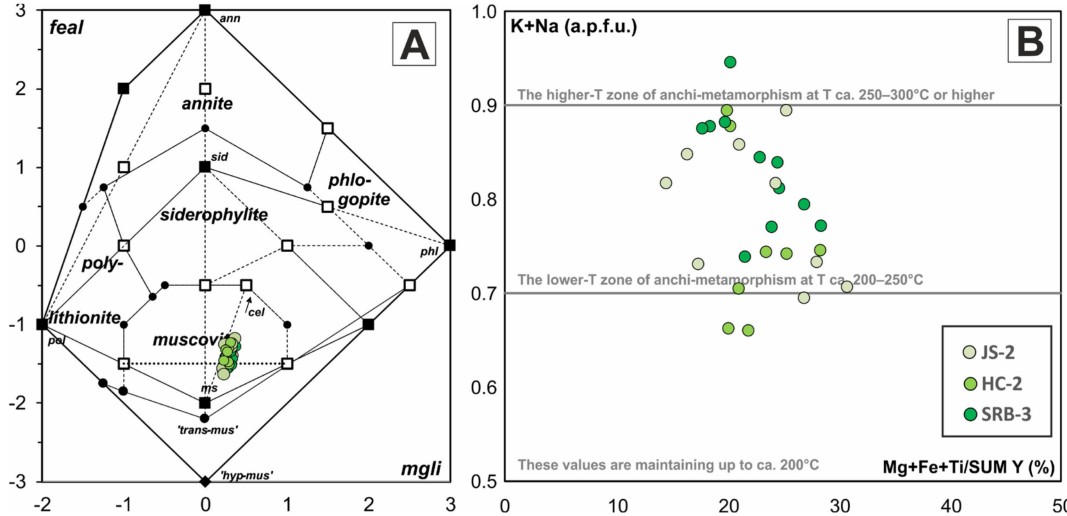

**Figure 26.** Classification diagram of white micas [85] from the lower Infratatric **Jašter thrust sheet**: (**A**) parameters: *mgli* = Mg-Li, *feal* = (Fe$^{2+}$ + Fe$^{3+}$ + Mn + Ti) − $^{VI}$Al$^{3+}$; (**B**) K + Na vs. Mg + Fe + Ti/SUM Y (%) compositional trend of white micas. Values for lines taken from the works of [69,103,104].

The geothermobarometry methods yield very similar P–T values (Table 3). The temperature for **Infratatric Hlohovec Nappe** Lower Triassic quartzites (HC-14) was determined by Ms–Pg thermometry [88] and this yielded 280–283 °C at 600 MPa. Figure 27A (and Table 3) highlights pressure of 600 MPa (HC-14 and HC-15) at estimated temperature of 280–300 °C using Si-in-Ph geobarometry [72]. The inferred recrystalization temperature of 220–250 °C for **Jašter thrust-sheet** Upper Cretaceous Couches-Rouges marly slates (HC-2, SRB-3) yields the pressures of 650–750 MPa based on Si-in-Ph geobarometry (Figure 27B; Table 3). Upper Cretaceous "flysch" sandstone (JS-5) yields undistinct metamorphic overprint at inferred temperature below 200 °C.

Figure 28A highlights that the metamorphic P–T conditions of the aplite vein (HC-19) in the Variscan granodiorite to tonalite in the higher Infratatric **Hlohovec Nappe** achieved 245 °C and 500 MPa using Chl–Ph pairs. Figure 28B constrains the **Jašter thrust-sheet** Upper Cretaceous Couches-Rouges type marly slates (JS-2) P–T conditions to 269–278 °C

and 510–620 MPa. This was accomplished with multi-equilibria Chl–Ph–Qtz–$H_2O$ geothermobarometry [73]. This temperature is consistent with an average temperature estimated by Chl geothermometry (Table 3).

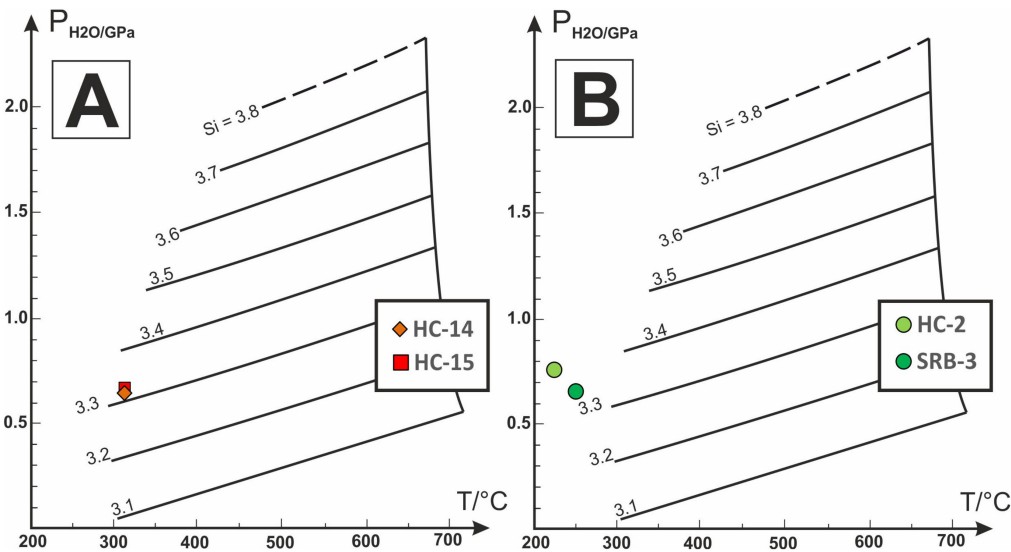

**Figure 27.** Pressure estimates from the Infratatric Hlohovec Block in diagram after [72]: (**A**) higher Infratatric **Hlohovec Nappe**; (**B**) lower Infratatric **Jašter thrust sheet**. Temperatures derived from Chl thermometry by [70,71,73] or estimated from mechanics of quartz dynamic recrystalization [105] in case of missing Chl in sample.

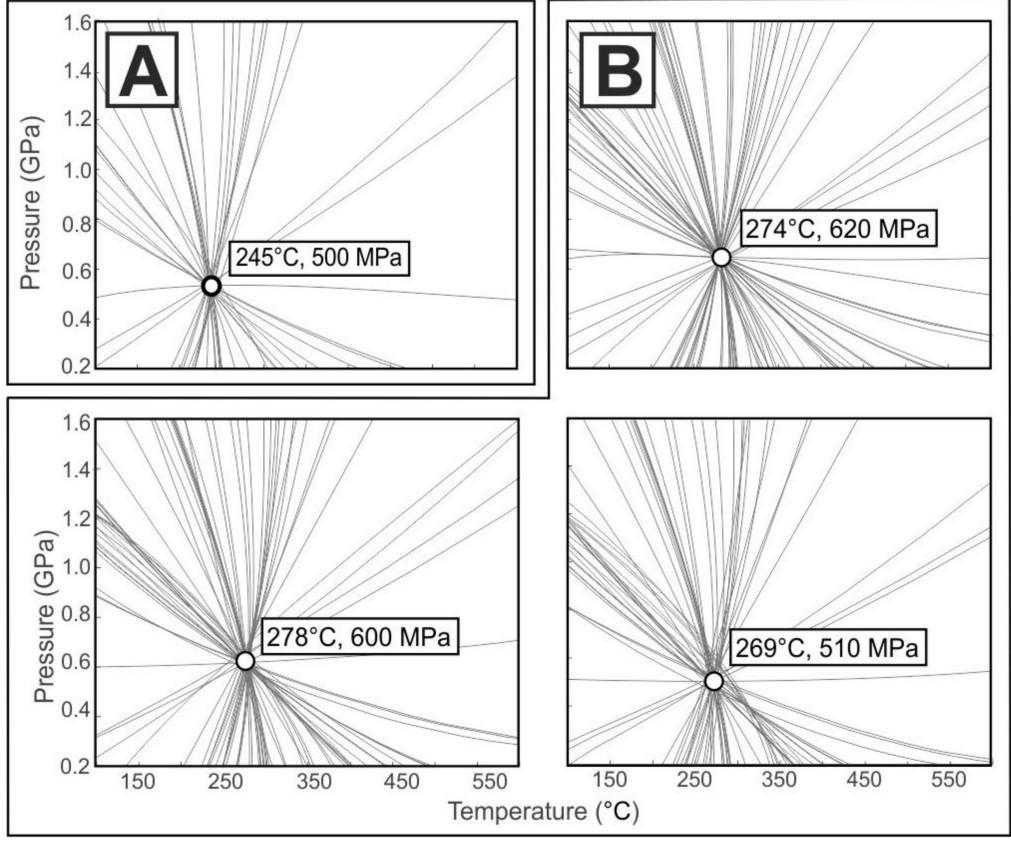

**Figure 28.** P–T diagrams with univariant curves and points intersecting with Chl–Ph pairs at equilibrium P–T conditions: (**A**) meta-aplite (HC-19) from inferred higher Infratatric **Hlohovec Nappe**; (**B**) Upper Cretaceous Couches-Rouges type marly slate (JS-2) from lower Infratatric **Jašter thrust sheet**. Based on Chl–Ph–Qtz–$H_2O$ geothermobarometry [73].

**Table 3.** Estimated P–T conditions from the Považský Inovec Mts. **Hlohovec Block**.

| Sample | Sample Description GPS Coordinates | Chl(C) | Chl(J) | Chl(V) T (°C) ±50 °C | Ms-Pg(B) | T* (°C) | Ph(M) P (MPa) ±200 MPa | Chl-Ms(V) T (°C) ±50 °C | Chl-Ms(V) P (MPa) ±200 MPa |
|---|---|---|---|---|---|---|---|---|---|
| | | | | **Hlohovec Block higher Infratatric Hlohovec Nappe** | | | | | |
| HC-14 | Lower Triassic meta-quartzite N 48°26.426′ E 17°49.579′ | - | - | - | 280–283 | 282 | 600 | - | - |
| HC-15 | Lower Triassic arkosic meta-sandstone N 48°26.421′ E 17°49.621′ | - | - | - | - | 300 | 600 | - | - |
| HC-19 | Meta-aplite N 48°26.575′ E 17°49.528′ | 301 | 307 | 261 | - | 280 | - | 245 | 500 |
| | | | | **Hlohovec Block lower Infratatric Jašter Thrust Sheet** | | | | | |
| JS-2 | Upper Cretaceous C.R. type marly slate clayey-rich layer N 48°26.995′ E 17°48.956′ | 306–331 | 272–304 | 288 | - | 275 | - | 269–278 | 510–620 |
| HC-2 | Upper Cretaceous C.R. type marly slate clayey-rich layer N 48°26.862′ E 17°49.286′ | - | - | - | - | 220 | 750 | - | - |
| SRB-3 | Upper Cretaceous C.R. type marly slate clayey-rich layer N 48°26.846′ E 17°49.404′ | - | - | - | - | 250 | 650 | - | - |

Chlorite thermometry by [70]—(C), [71]—(J). Ms–Pg thermometry after [88]—(B). Si-in-Ph barometry after [72]—(M) based on average Si-in-Ph content. Chl thermometry and Chl–Ms thermobarometry after [73]—(V). *—Average of calculated T or estimated T from mechanics of quartz dynamic recrystalization [105] in case of missing Chl. C.R. = Couches Rouges.

4.1.2. Subautochthonous Fatricum

Tribeč Mts. Zobor Nappe

**Petrography**

The Zobor Nappe granitoid basement has varied deformation recrystalization from macroscopically undeformed or protomylonitic (Figure 29A) to intensively mylonitized zones, with developed S–C structures indicating the NW-ward thrusting and top-to SE extensional sliding (Figure 29B,C). Mylonitic foliation is covered by fine-grained aggregates of white mica and Chl, and the Qtz–Fsp porphyroclasts are prolonged in the NW–SE striking stretching lineation.

The sedimentary cover rocks are distinctly metamorphosed, and the schistosity planes are covered by white mica (Ms to Ph) and Chl. Schistose Middle Triassic marbles (Figure 29D), and Lower Triassic meta-quartzites (Figure 29E–G) are often re-folded with the basement mylonites.

The distinct metamorphic textures were observable in both the basement and cover rocks. Granodiorite to tonalite blastomylonites contain newly formed dynamically recrystalized Qtz, white mica (Ph), Chl, Ab, ±Ep, Czo, Cal (Figure 30A–C). Albitized Fsp porphyroclasts are stretched and the fractures are healed by Qtz, Chl and Ph. Middle Triassic marbles have thin very fine-grained white-mica layers (Figure 30 D). Lower Triassic meta-quartzites to quartzitic schists are rich in metamorphic white mica (Figure 30E–H). The S1 metamorphic schistosity planes are often folded with developed S2 micro-fold axial-plane cleavage (Figure 30E,F). Quartz porphyroclasts are dynamically recrystalized (Figure 30H).

**BSE images, mineral chemistry and P–T estimates**

BSE images of investigated rock textures (Figure 31) document newly formed metamorphic phases which were used for the P–T estimates.

Quartz, albitized Fsp, chloritized Bt and rare Ms porphyroclasts are surrounded by phengitic white mica and Chl aggregates oriented in schistosity planes of tonalite blastomylonites (Figure 31A–C). Middle Triassic marble mylonites contain segmented Ph-rich layers and individual flakes disappeared in recrystalized Cal aggregates (Figure 31D). Lower Triassic meta-quartzites to quartzitic schists and meta-sandstones contain fine-

grained newly formed Ph less Cel-Ms in recrystalized Qtz aggregates (Figure 31E–H). Folded quartzitic schists show syn-metamorphic cleavage planes S2 (Figure 31E,F). Arkosic lithologies contain deformed Ms and Fsp porphyroclasts (Figure 31H).

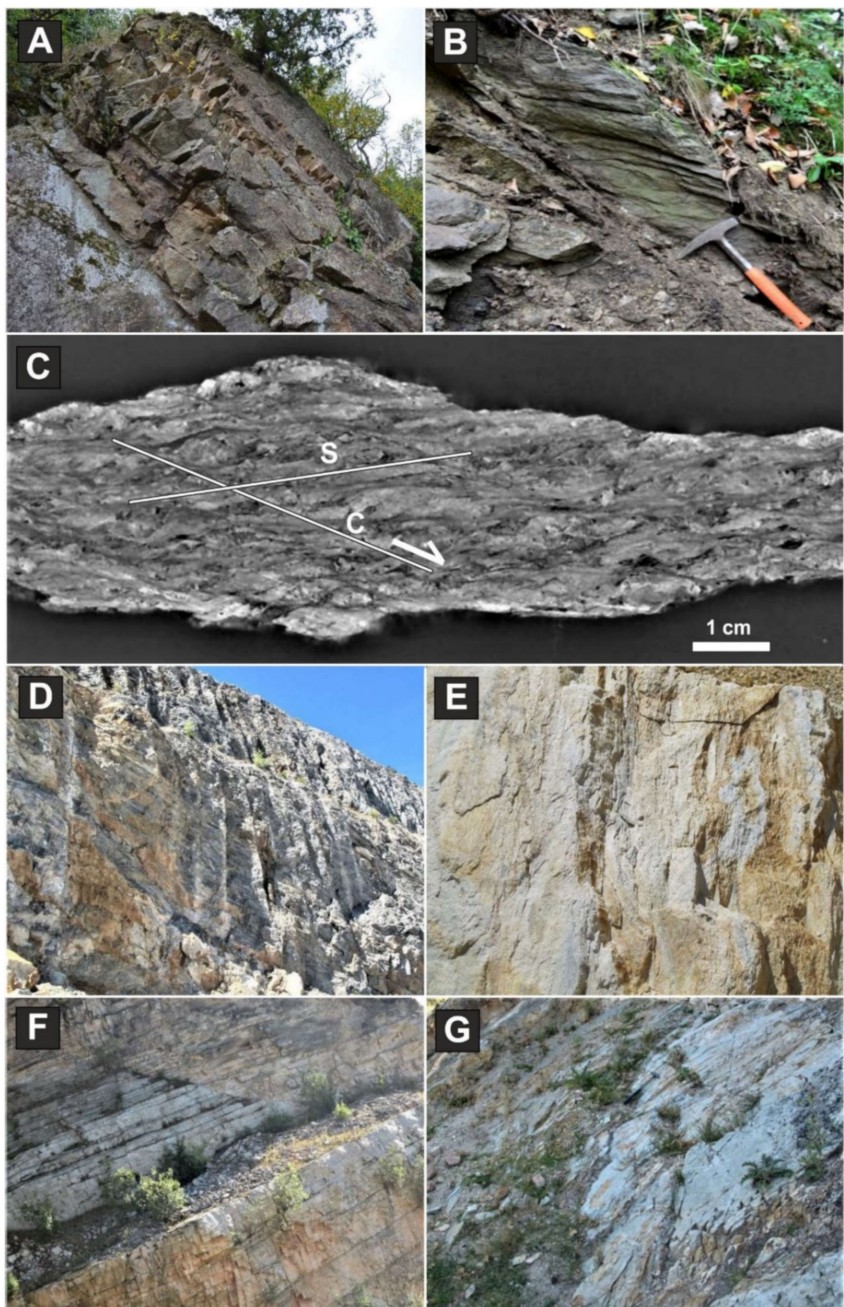

**Figure 29.** Macroscopic features of tectono-metamorphic overprinting in the subautochthonous Fatricum Tribeč Mts. **Zobor Nappe**: (**A**) tonalite protomylonite (NR-1, Nitra—Jazierko); (**B**) tonalite mylonite (ZLT-2, at Zlatno). Subhorizontal mylonitic schistosity S-planes (parallel to C1 in NW-ward thrusting) are crosscut by the SE dipping C(2)-planes related to the SE-ward extensional sliding; (**C**) a detailed view on the asymmetry of the mylonitic S–C structures reveals the top-to-the (E)SE extensional sliding in the present-day co-ordinates (ZLT-2; XZ cut, perpendicular to the foliation and parallel to the stretching lineation); (**D**) Middle Triassic marbles (ZI-4, Žirany quarry). Metamorphic schistosity crossscut by steeply dipping post-metamorphic cleavage; (**E–G**) Lower Triassic schistose meta-quartzites: from Žirany quarry, 8 km NE of Nitra township (E, ZI-3 dated sample), Krnča quarry (F, KR-3) and Hrušov Castle Hill (G, HR-1).

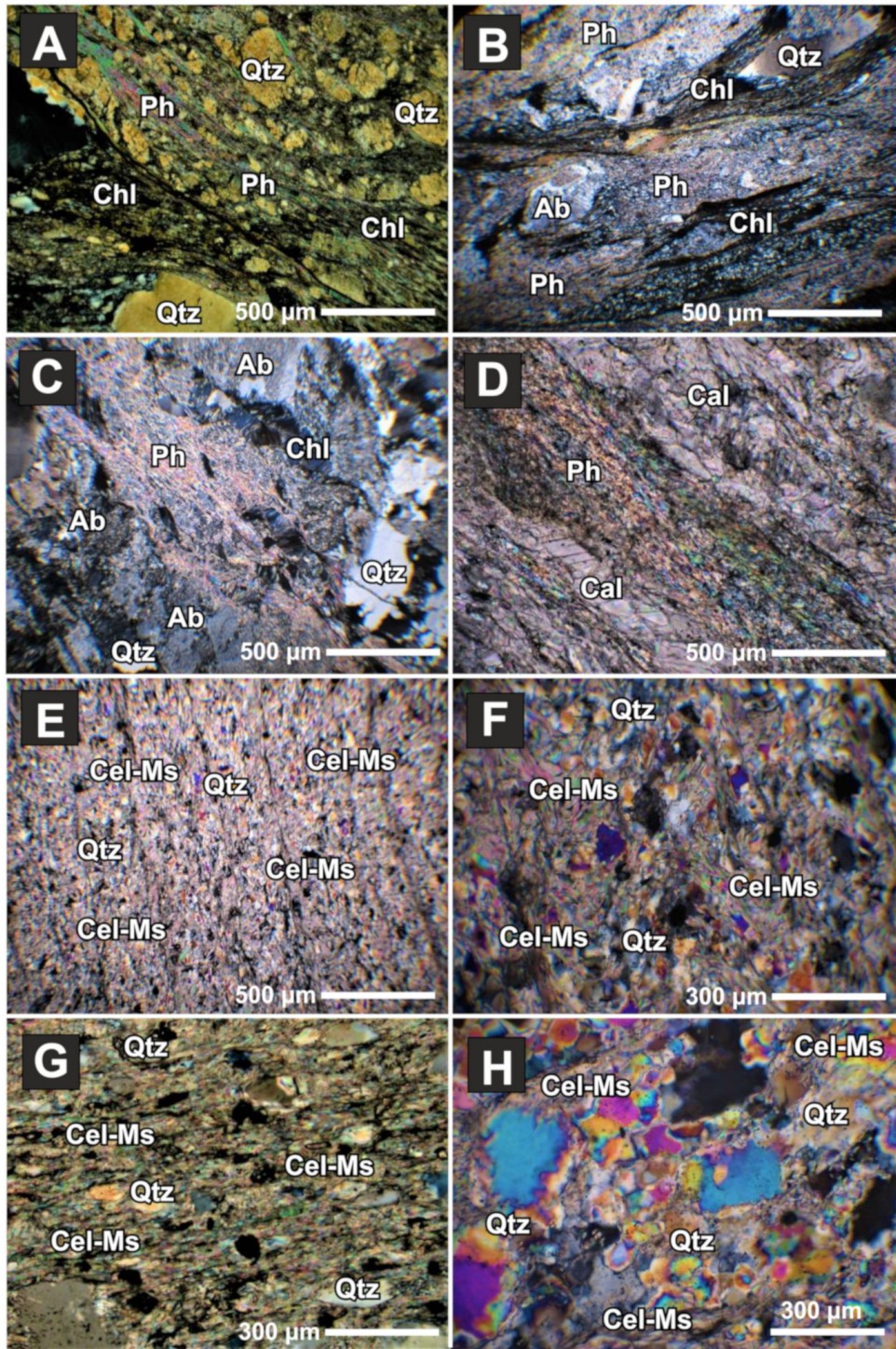

**Figure 30.** Microscopic textures from the subautochthonous Fatricum basement and cover rocks in the Tribeč Mts. **Zobor Nappe**: (**A**) tonalite mylonite (NR-1, Nitra-Jazierko); (**B**) tonalite mylonite S–C structure (ZLT-2) indicating the top-to-the SE extensional sliding; (**C**) protomylonite (KR-1, Krnča quarry); (**D**) Middle Triassic marble (ZI-4, Žirany quarry); (**E**) Lower Triassic quartzitic schist (ZI-3 dated sample, Žirany quarry); (**F**) Lower Triassic quartzitic schist (JEL-1, Jelenec quarry); (**G**) Lower Triassic quartzite (LOV-1, Lovce quarry; (**H**) Lower Triassic meta-sandstone (KR-3, Krnča quarry). All pictures at *X* N.

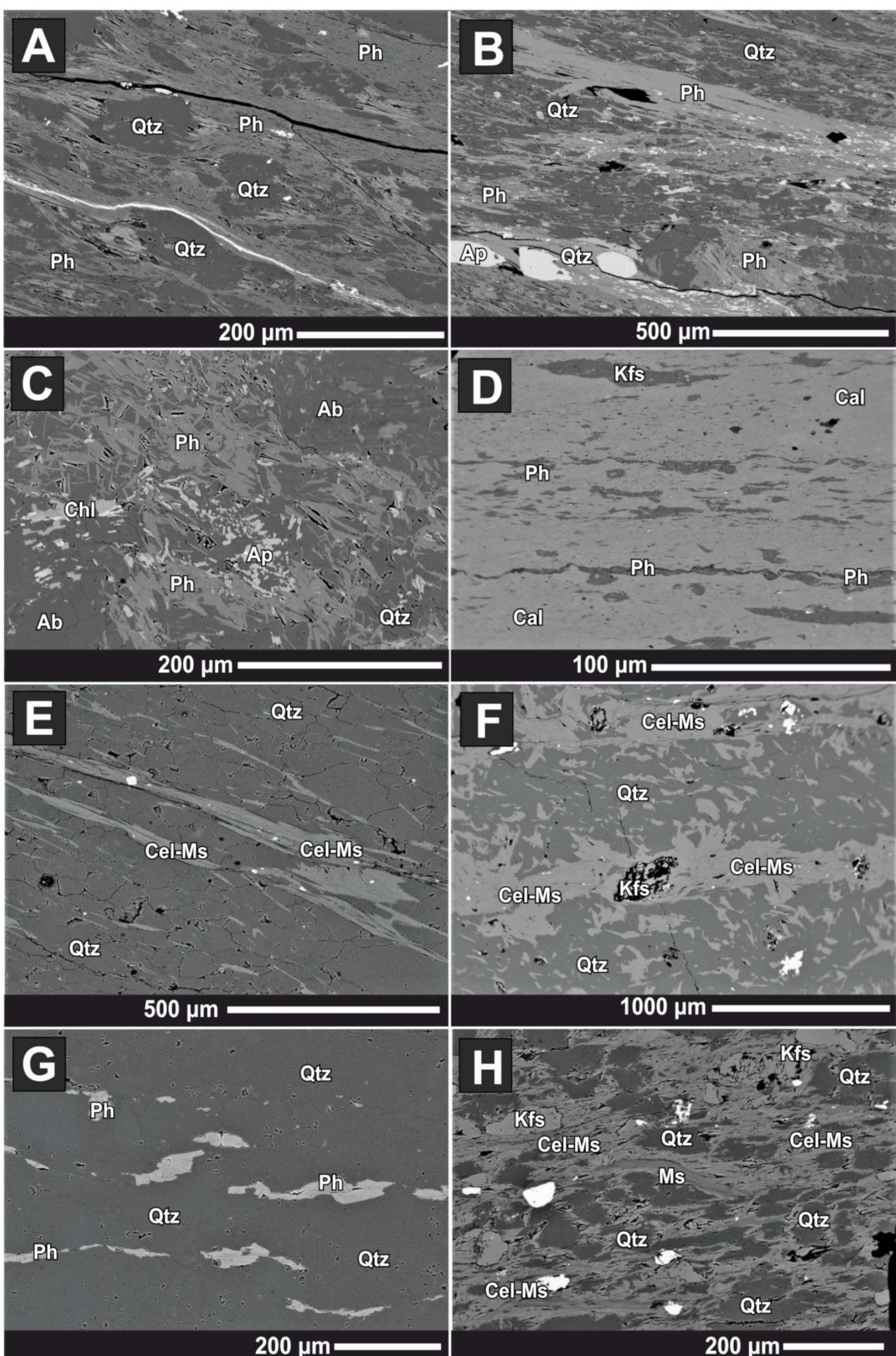

**Figure 31.** BSE images of white mica aggregates (±Chl) from the subautochthonous Fatric rocks of the Tribeč Mts. **Zobor Nappe**: (**A**) tonalite mylonite (NR-1, Nitra-Jazierko); (**B**) tonalite mylonite (ZLT-2, Zlatno); (**C**) granite mylonite (MV-2, Mišov Hill); (**D**) Middle Triassic marble (ZI-4, Žirany quarry); (**E**) Lower Triassic quartzitic schists (ZI-3 dated sample, Žirany quarry); (**F**) microfolds and axial-plane cleavage (S2—horizontal in the figure) in a Lower Triassic quartzitic schist (JEL-1, Jelenec quarry); (**G**) Lower Triassic meta-quartzite (HR-1, Hrušov Castle Hill); (**H**) Lower Triassic meta-sandstone (KR-3, Krnča quarry).

The applied geothermobarometric methods are based on the metamorphic Chl and Ph or Chl–Ph pairs. The chemical analyses of representative minerals from the Zobor Nappe are available in Table S5.

Granitoid basement tonalite blastomylonites (NR-1, ZLT-1, ZLT-2) contain mostly chlorite of clinochlore composition in the classification diagram (Figure 32). Chlorites contain up to 17.4–24.1 wt.% FeO and 11.0–18.1 wt.% MgO. The Fe/(Fe + Mg) ratio is 0.39–0.49. The $^{VI}$Al content reaches 1.23–1.97 *p.f.u.* in the studied sample (Table S5). Chlorite does not occur in quartzitic meta-sediments.

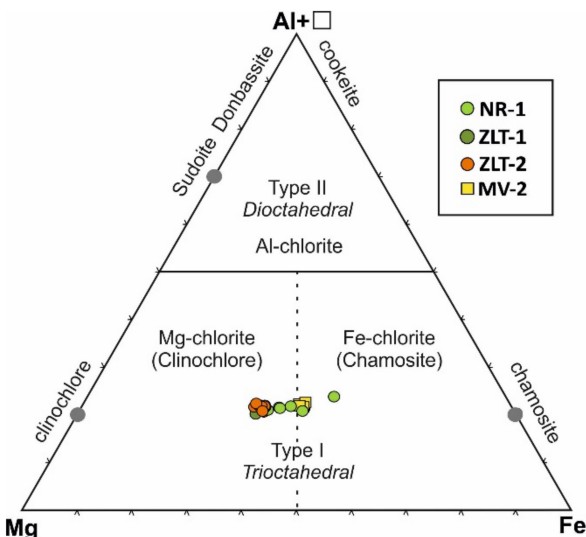

**Figure 32.** Al + □–Mg–Fe classification diagram of chlorite after [84] from the subautochthonous Fatricum of the Tribeč Mts. **Zobor Nappe**.

The metamorphosed basement and cover rocks of the Tribeč Mts. **Zobor Nappe** contain Cel-Ms. Figure 33A highlights that all compositions of the white micas project along the muscovite–celadonite mixing-line in the classification diagram. White micas from tonalite blastomylonites have SiO$_2$ content from 46.4 to 51.1 wt.% (3.22–3.46 Si *p.f.u.*). FeO and MgO content in the samples varies between 1.9–6.5 and 1.7–3.6 wt.%, respectively. Lower Triassic meta-quartzites generally have lower SiO$_2$ content from 45.0 to 47.6 wt.% (3.13–3.6 Si *p.f.u.*). FeO content varies between 1.6–6.5 wt.% with all the highest values linked to sample HR-1 (above 6 wt.%). MgO is relatively low up to 1.55 wt.%. The SiO$_2$ content in the Middle Triassic marbles was 47.1 to 53.6 wt.% (3.19–3.51 Si *p.f.u.*). The FeO content is rather low between 1.9 and 2.6 wt.%. and MgO is 1.5–5.1 wt.% (Table S5). Analyses of the white micas in Figure 33B mostly fall within higher anchi-metamorphism to lower greenschist facies with alkalis content of 0.9–1.0 K + Na *p.f.u.* and K$_2$O up to 8.9–11.9 wt. %. marbles (ZI-2, ZLT-2) exceptionally cross both lower and higher-T field down to 0.87 K + Na *p.f.u.* Sample ZI-3 contained only Ms.

The geothermobarometry for the **Zobor Nappe** basement granitoid rocks yielded a wider pressure interval of 400–700 MPa with the lowest pressures for the tonalite mylonites at Zlatno at average Chl temperatures of 326 °C. Similar lower pressures of 300 MPa were determined from Jelenec (JEL-1) and Krnča (KR-3) Lower Triassic meta-quartzites at estimated temperature of 300 °C. The relatively higher pressures of 700 MPa at estimated 300 °C are indicated in Lower Triassic meta-quartzites from Hrušov Castle Hill (HR-1) S of Skýcov (Figure 34; Table 4).

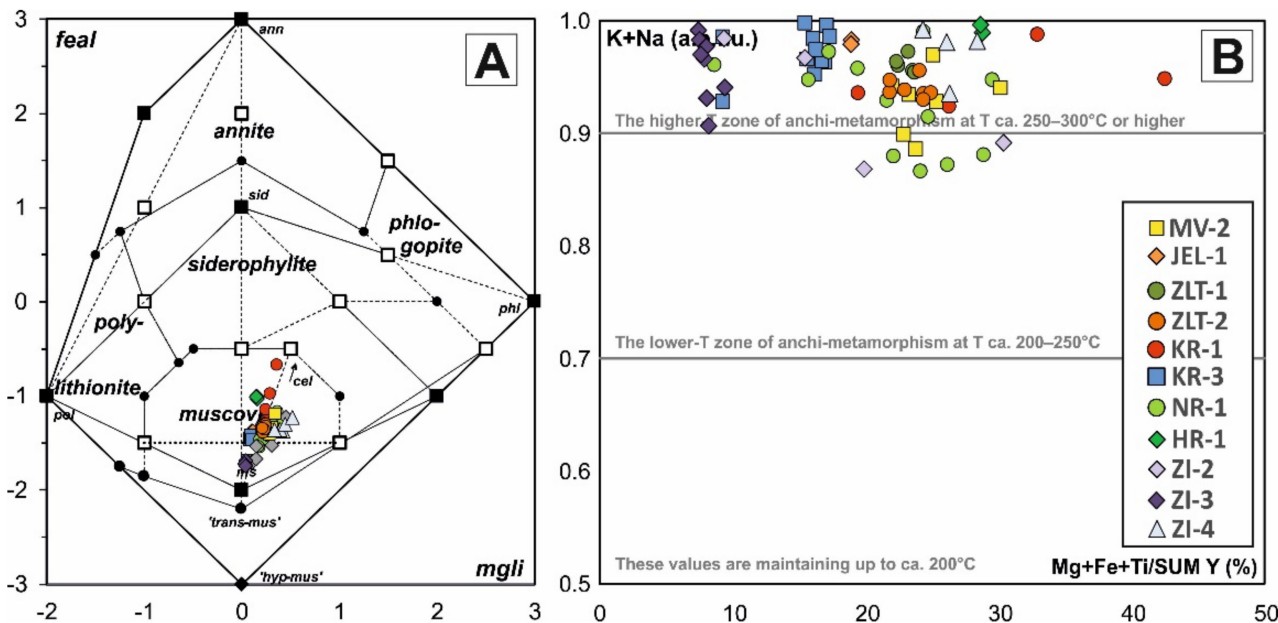

**Figure 33.** Classification diagram of white micas [85] from the subautochthonous Fatricum of the Tribeč Mts. **Zobor Nappe**: (**A**) parameters: *mgli* = Mg-Li, *feal* = (Fe$^{2+}$ + Fe$^{3+}$ + Mn + Ti) − $^{VI}$Al$^{3+}$; (**B**) K + Na vs. Mg + Fe + Ti/SUM Y (%) compositional trend of white micas. Values for lines taken from the works of [69,103,104].

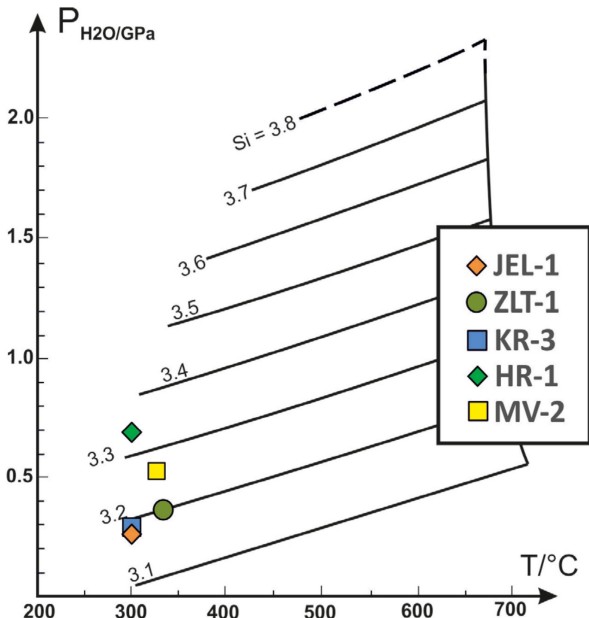

**Figure 34.** Pressure estimates diagram after [72] from the subautochthonous Fatricum of the Tribeč Mts. **Zobor Nappe**. Temperatures derived from Chl thermometry by [70,71,73] or estimated from mechanics of quartz dynamic recrystalization [105] in case of missing Chl in sample.

Figure 35A highlights the metamorphic P–T conditions of meta-tonalite (NR-1) in the Tribeč Mts. **Zobor Nappe** estimated by Chl–Ph pairs at 250–353 °C and 500–700 MPa. Figure 35B constrains P–T conditions of meta-tonalite at Zlatno (ZLT-2) to 308–345 °C and 400–480 MPa. These results were obtained with multi-equilibria Chl–Ph–Qtz–H$_2$O geothermobarometry [73].

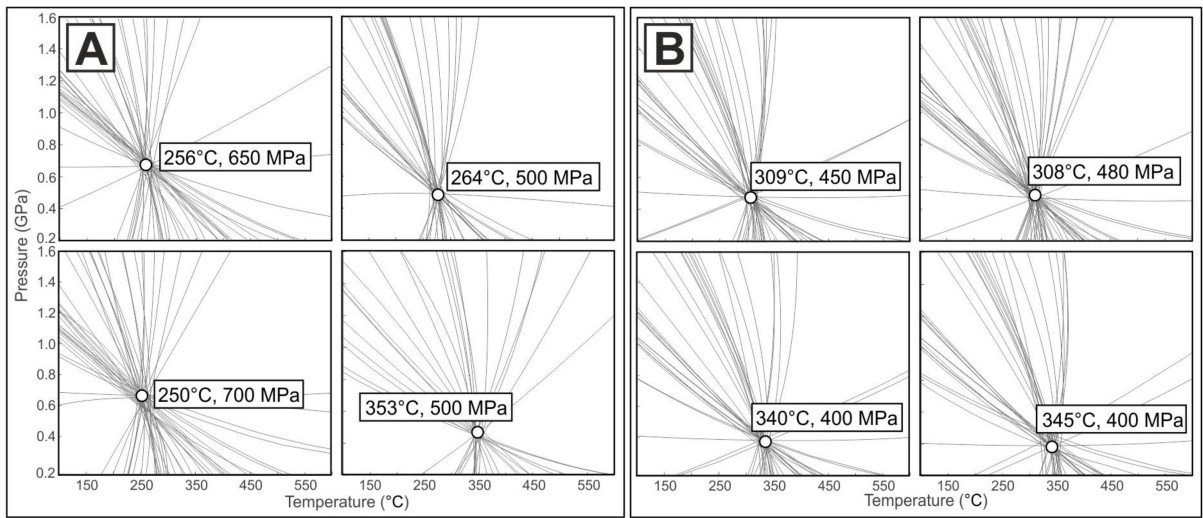

**Figure 35.** P–T diagrams with univariant curves and points intersecting with Chl–Ph pairs at equilibrium P–T conditions from the subautochthonous Fatricum of the Tribeč Mts. **Zobor Nappe**: (**A**) meta-tonalite (NR-1, Nitra-Jazierko); (**B**) meta-tonalite (ZLT-2, at Zlatno). Based on Chl–Ph–Qtz–$H_2O$ geothermobarometry [73].

**Table 4.** Estimated P–T conditions from the Tribeč Mts. **Zobor Nappe**.

| Sample | Sample Description GPS Coordinates | Chl$^{(C)}$ | Chl$^{(J)}$ T (°C) ±50 °C | Chl$^{(V)}$ | T* (°C) | Ph$^{(M)}$ P (MPa) ±200 MPa | Chl-Ms$^{(V)}$ T (°C) ±50 °C | P (MPa) ±200 MPa |
|---|---|---|---|---|---|---|---|---|
| NR-1 | Granodiorite/tonalite protomylonite N 48°20.202′ E 18°5.836′ | 271–336 | 277–340 | 298–324 | 308 | - | 250–353 | 500–700 |
| ZLT-1 | Granodiorite/tonalite mylonite N 48°28.176′; E 018°18.504′ | 286–359 | 288–361 | 256–359 | 326 | 400 | - | - |
| ZLT-2 | Granodiorite/tonalite mylonite N 48°28.176′; E 018°18.504′ | 335–375 | 338–377 | 333–391 | 352 | - | 308–345 | 400–480 |
| MV-2 | Granodiorite/tonalite mylonite N 48°29.354′; E 018°16.178′ | 312–346 | 317–351 | 269–359 | 324 | 500 | - | - |
| JEL-1 | Lower Triassic meta-quartzite N 48°23.059′; E 018°12.574′ | - | - | - | 300 | 300 | - | - |
| HR-1 | Lower Triassic meta-quartzite N 48°28.624′ E 18°25.473′ | - | - | - | 300 | 700 | - | - |
| KR-3 | Lower Triassic meta-sandstone N 48°32.079′ E 18°15.769′ | - | - | - | 300 | 300 | - | - |

Chlorite thermometry by [70]—(C), [71]—(J). Si-in-Ph barometry after [72]—(M) based on average Si-in-Ph content. Chl thermometry and Chl–Ms thermobarometry after [73]—(V). *—Average of calculated T or estimated T from mechanics of quartz dynamic recrystalization [105] in case of missing Chl.

Tribeč Mts. Razdiel Nappe

**Petrography**

The Razdiel Nappe shows distinct tectono-metamorphic overprinting of the basement and cover rock. Our research focused on the Alpine metamorphic overprinting textures in: (1) meta-granites to meta-tonalites (Figure 36A,B); (2) micaschist phyllonites (Figure 36C); (3) the Permian meta-basalt (Figure 36D) and meta-sediments (Figure 36E–G) and (4) Lower Cretaceous slates (Figure 36H) which contain newly formed metamorphic phases.

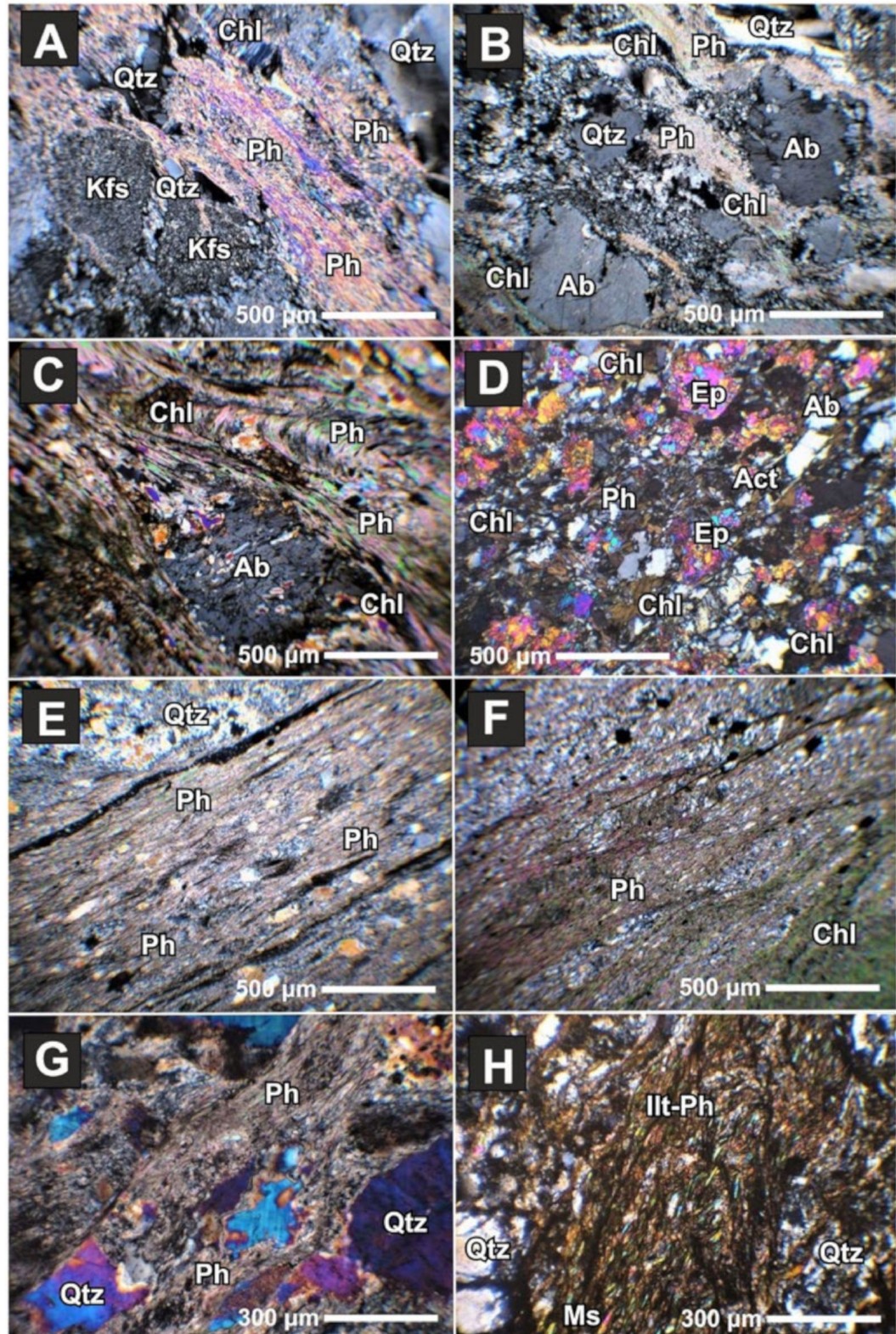

**Figure 36.** Microscopic textures from the subautochthonous Fatricum basement and cover rocks in the Tribeč Mts. **Razdiel Nappe**: (**A**) tonalite mylonite (TRI-21, Chudá Valley); (**B**) granodiorite mylonite (KAM-1, pass E of Kamenná Hill); (**C**) micaschist phyllonite (NES-2, Nestašova Valley); (**D**) Permian meta-basalt (TR-43, Drahožická Valley); (**E–G**) Permian meta-sediments: LAM-2A, W of Lámaniny Hill (**E**); TR-46, Nitrolinská Valley (**F**); VYC-2, Vyčoma Valley (**G**); (**H**) Lower Cretaceous slate (NEM-1, Nemčeky). All pictures at *X* N.

The Razdiel Nappe granitic rocks underwent varied deformational recrystalization to protomylonites and mylonites (Figure 36A,B). Their blastomylonitic schistosity is defined by fine-grained aggregates of white mica (Ph), Chl, Ab, ±Ep, Czo and Cal and the Qtz–Fsp porphyroclasts are prolongated in the NW–SE striking stretching lineation. Quartz, Fsp and rare mica (Ms and chloritized Bt) porphyroclasts are stretched and fractured, and the fractures are infilled with the afore-mentioned newly formed phases. Phyllonitized micaschists are rich in fine-grain aggregates of Chl and white mica, and rotated Ab porphyroblasts (Figure 36C).

The Permian meta-basalt intercalations have greenschist facies mineral assemblages of Chl, Ab, Act, Ep, Czo, ±Ph and Cal (Figure 36D), and the meta-rhyolite consists of newly formed Ph, Chl, Ab, with relics of hypidiomorphic Qtz and sericitized Fsp. The schistosity planes of the Permian meta-sediments are covered by newly formed white mica (Ms to Ph) and Chl (Figure 36E–G). A characteristic feature is the S2 syn-metamorphic crenulation cleavage crosscutting the S1 metamorphic schistosity (Figure 36F). Crenulated metamorphic schistosity is also typical in the Lower Cretaceous slates (Figure 36H).

**BSE images, mineral chemistry and P–T estimates**

BSE images of the rock textures (Figure 37) document the newly formed metamorphic phases used for P–T estimates.

Tonalite protomylonites to blastomylonites (Figure 37A,B) contain relics of albitized magmatic Fsp porphyroclasts, Qtz, chloritized Bt and rare Ms which are surrounded by oriented phengitic white mica and Chl aggregates in the schistosity planes. Phyllonitized micaschists are mostly composed of Qtz, Ph, Chl and Ab (Figure 37C). Layered amphibolites are recrystalized to Act–Ph greenschists and still contain Hbl relics.

The Permian meta-basalts have a greenschist-facies mineral assemblage of Chl, Ab, Act, Ep, Czo, ±Ph, Cal and magmatic Hbl relics (Figure 37D). Permian meta-sediments are rich in newly formed metamorphic Ph and Chl associated with Ab, Qtz and rare Cal (Figure 37E–G). Some of these exhibit S2 syn-metamorphic cleavage (Figure 37F). Similar metamorphic textures also occur in the Lower Cretaceous slates (Figure 37H), with relics of tiny Ms clasts in the newly formed anchi-metamorphic Ilt-Phe matrix.

The geothermobarometric methods were applied to Chl and Ph or Chl–Ph pairs. The compositions of representative minerals are available in Table S6.

Chlorite from the granodiorite to tonalite protomylonites to blastomylonites (TRI-21, KAM-1, LAM-1, CHD-2), Lower Cretaceous slate (NEM-1), micaschist phyllonite (NES-2) and parts of the Permian meta-basalts (TR-43) and hosting meta-sediments (DRD-1) have chamosite composition in the classification diagram (Figure 38). Chlorites contain up to 24.2–33.9 wt.% FeO and 7.9–14.4 wt.% MgO. The Fe/(Fe + Mg) ratio is 0.52–0.70. The $^{VI}$Al content reaches 1.27–1.64 *p.f.u.* in the studied sample. Chlorite from the remaining Permian meta-basalts (TR-35) and basaltic meta-pyroclastics (TR-33), Permian meta-rhyolites (TR-51) and meta-sediments (TR-46, LAM-2) is clinochlore (Figure 38). The chlorites contain up to 11.4–30.3 wt.% FeO and 10.9–25.6 wt.% MgO. The Fe/(Fe + Mg) ratio is 0.20–0.61. The $^{VI}$Al content reaches 1.19–1.49 *p.f.u.* (Table S5**)**.

The metamorphosed basement and cover rocks of the Tribeč Mts. **Razdiel Nappe** contain Cel-Ms. Figure 39A highlights that all Ce-Ms project along the muscovite–celadonite mixing-line. White micas from tonalite blastomylonites and micaschist phyllonites have $SiO_2$ content from 44.1 to 49.1 wt.% (3.07–3.25 Si *p.f.u.*). FeO and MgO content varies in samples between 2.0–7.8 wt.% and 0.7–3.7 wt.%. $TiO_2$ content reaches 2.89 wt. %. The $SiO_2$ content in white micas from the Permian meta-sediments and meta-volcanics varies between 45.3 and 51.3 wt.% (3.05–3.47 Si *p.f.u.*), with higher content linked to meta-volcanics. FeO and MgO vary in samples between 1.0–7.3 wt.% and 0.6–6.4 wt.% with low values mostly in meta-sediments (Table S6). $TiO_2$ content reaches 1.2 wt.%. Analyses of the white micas in Figure 39B mostly fall within higher anchi-metamorphism to lower greenschist facies with alkalis content of 0.9–1.0 K + Na *p.f.u.* and $K_2O$ up to 8.9–11.0 wt. %. Some samples are exceptions and cross both the lower and higher-T field down to the 0.86 K + Na *p.f.u.* level **(Table S6)**.

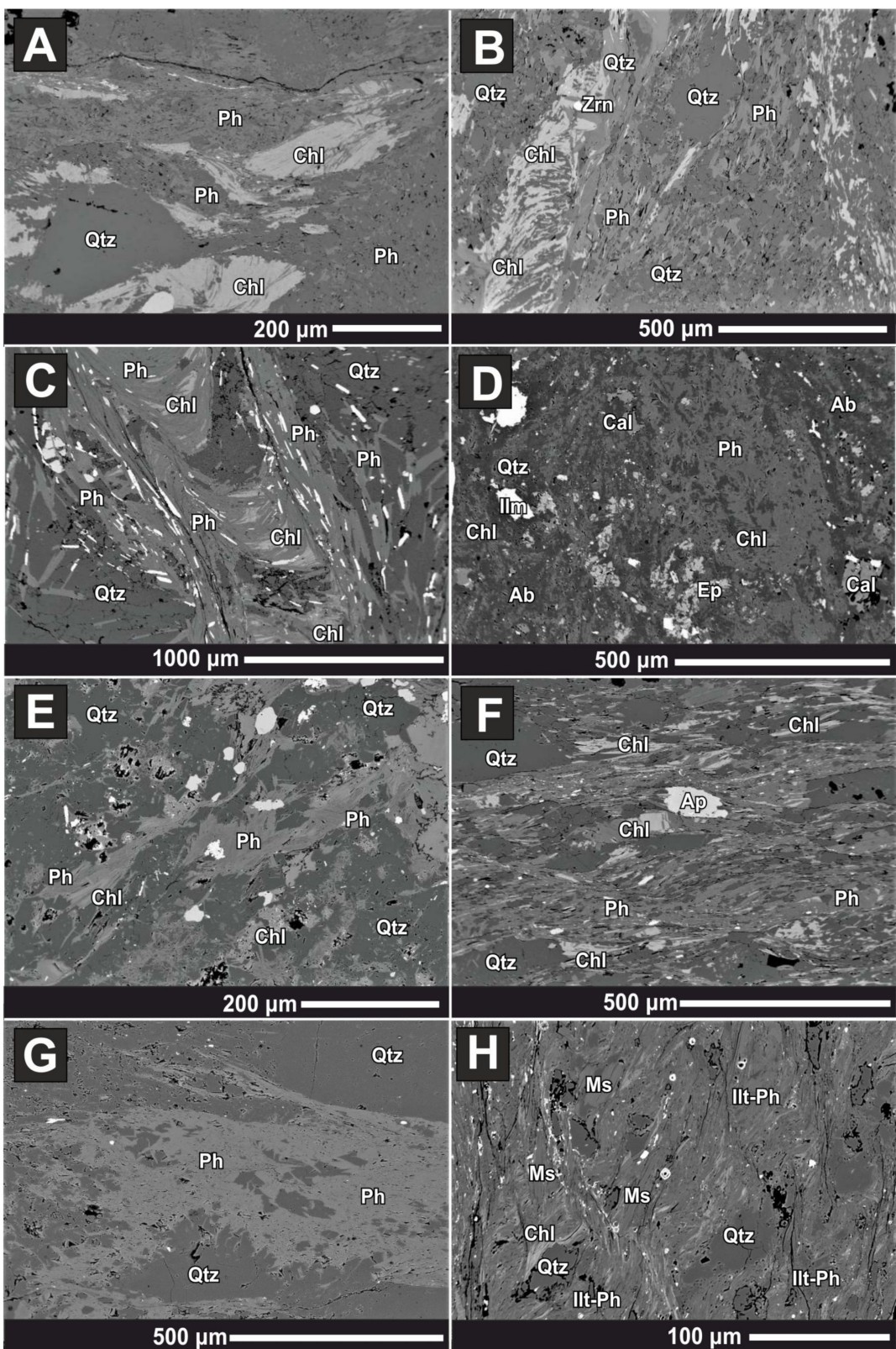

**Figure 37.** BSE images of white mica aggregates and Chl from the subautochthonous Fatric rocks of the Tribeč Mts. **Razdiel Nappe**: (**A**) tonalite mylonite (TRI-21, Chudá Valley); (**B**) granodiorite mylonite (KAM-1, E of Kamenná Hill); (**C**) micaschist phyllonite (NES-2; Nestašova Valley); (**D**) Permian meta-basalt (TR-35, Drahožická Valley); (**E–G**) Permian meta-sediments: TR-46, Nitrolinská Valley (**E**); LAM-2A, W of Lámaniny Hill (**F**); VYC-2, Vyčoma Valley (**G**); (**H**) Lower Cretaceous slate (NEM-1, Nemčeky).

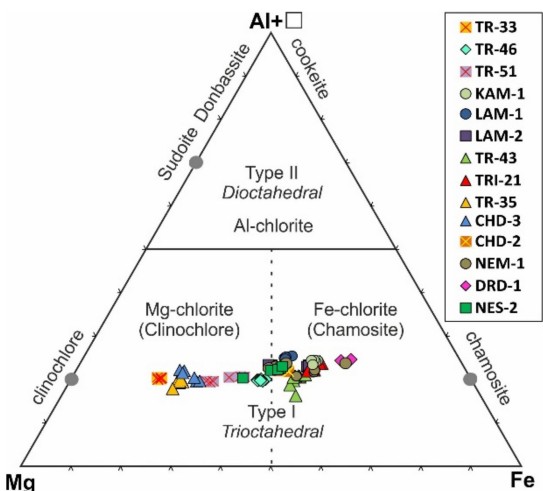

**Figure 38.** Al + □–Mg–Fe classification diagram of chlorite after [84] from the subautochthonous Fatricum of the Tribeč Mts. **Razdiel Nappe**.

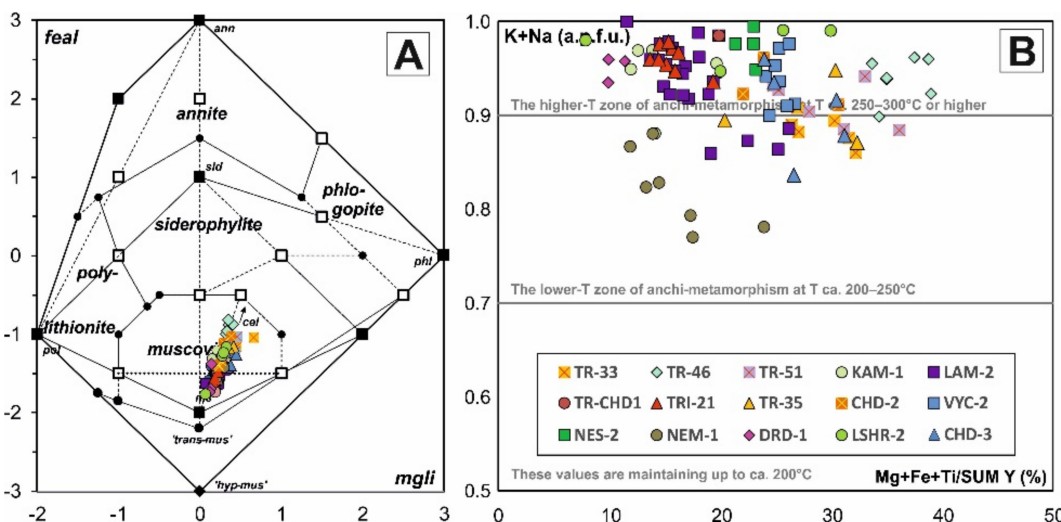

**Figure 39.** Classification diagram of white micas [85] from the subautochthonous Fatricum of the Tribeč Mts. **Razdiel Nappe**: (**A**) parameters: *mgli* = Mg-Li, *feal* = (Fe$^{2+}$ + Fe$^{3+}$ + Mn + Ti) − $^{VI}$Al$^{3+}$; (**B**) K + Na vs. Mg + Fe + Ti/SUM Y (%) compositional trend of white micas. Values for lines taken from the works of [69,103,104].

The geothermobarometry applied to the basement tonalite to blastomylonites of the **Razdiel Nappe** yielded a relatively narrow pressure interval of 400–550 MPa at calculated average Chl temperatures of 308–365 °C (Figure 40; Table 5). Micaschist phyllonites yield 400 MPa pressure at an average 342 °C temperature. Similar pressures were achieved by meta-sediments at 400 MPa and approximately 300 °C temperature (Table 5) based on Si-in-Ph barometry [72]. Chl–Ph pairs in Permian meta-basalt (TRI-35, TR-33) generally yield higher P–T estimates at 700–750 MPa and average Chl temperatures 303–323 °C.

Figure 41A–C highlights the metamorphic P–T conditions of meta-granodiorite (KAM-1) and meta-tonalite (LAM-1, CHD-2) of the **Razdiel Nappe** according to Chl–Ph pairs at 249–270 °C and 400–600 MPa. Micaschist phyllonites (NES-2, Figure 41E) were constrained at 252 °C and 400 MPa. Figure 41D,F constrains the P–T conditions of the Permian cover meta-sediments (schistose layers in meta-arkoses, s. LAM-2A, B) of the afore-mentioned meta-granodiorite (KAM-1) and neighbouring meta-tonalite (LAM-1) to 257–270 °C and 400–600 MPa. Figure 41G shows the inferred Low Cretaceous slate peak at 337 °C and 400 MPa. This was accomplished by multi-equilibria Chl–Ph–Qtz–H$_2$O geothermobarometry [73].

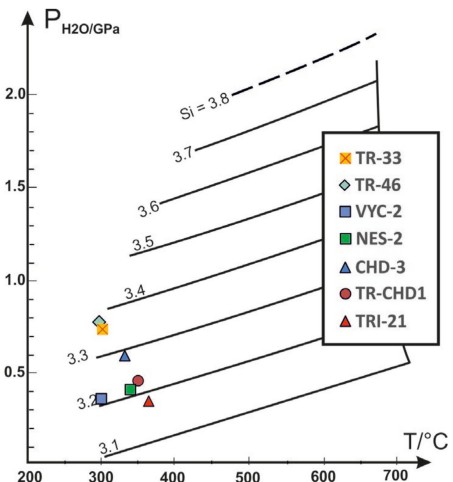

**Figure 40.** Pressure estimates diagram after [72] from the subautochthonous Fatricum of the Tribeč Mts. **Razdiel Nappe**. Temperatures derived from Chl thermometry by [70,71,73] or estimated from mechanics of quartz dynamic recrystalization [105] in case of missing Chl in sample.

**Perple_X pseudosection** modeling determined the D1 stage P–T conditions of the Permian meta-basalt (TR-35) in the greenschist-facies metamorphic overprinting at 370–400 °C at 580–700 MPa pressure, showing the assemblage of Amp–Chl–Ms–Ep–Bt–Ab (Figure 42, Table 5).

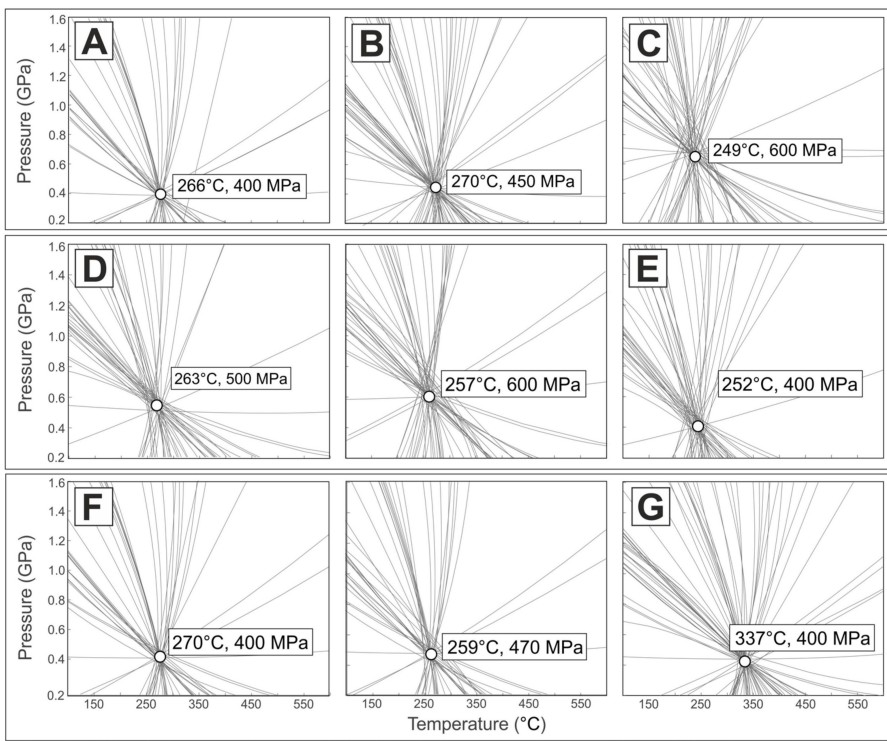

**Figure 41.** P–T diagrams with univariant curves and points intersecting with Chl–Ph pairs at equilibrium P–T conditions from the subautochthonous Fatricum of the Tribeč Mts. **Razdiel Nappe**: (**A**) meta-granodiorite (KAM-1, E of Kamenná Hill); (**B**) meta-tonalite (LAM-1, W of Lámaniny Hill); (**C**) meta-tonalite (CHD-2, Chudá Valley); (**D**) Permian schist (LAM-2A, W of Lámaniny Hill); (**E**) micashist phyllonite (NES-2, Nestašova Valley); (**F**) Permian meta-arkose (LAM-2B, W of Lámaniny Hill); (**G**) Lower Cretaceous slate (NEM-1, Nemčeky). Based on Chl–Ph–Qtz–$H_2O$ geothermobarometry [73].

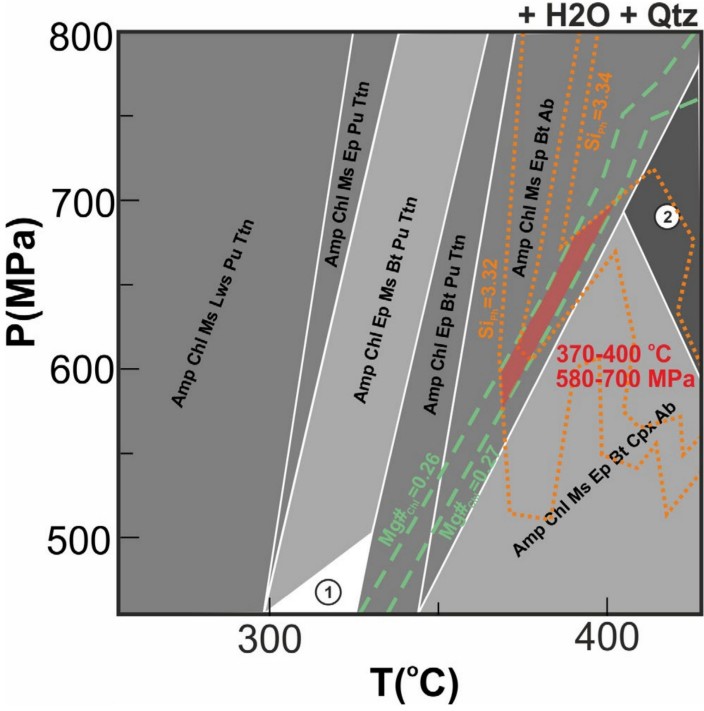

**Figure 42.** P–T pseudosection calculated by Perple_X thermodynamic software [92,93] for the D1 peak metamorphic stage of Permian meta-basalt (TR-35) of the subautochthonous Fatricum of the Tribeč Mts. **Razdiel Nappe**, using isopleths intervals of Si-in-Ph 3.32–3.34 *p.f.u.* and Chl$_{Mg\#}$ 0.26–0.27. Numbers refer to mineral assemblage: **1**—Amp, Chl, Ms, Ep, Bt, Pu, Ab, Ttn, **2**—Amp, Chl, Ep, Ms. Whole-rock composition used for modeling can be found in Table S1.

**Table 5.** Estimated P–T conditions from the Tribeč Mts. **Razdiel Nappe**.

| Sample | Sample Description | Chl$^{(C)}$ | Chl$^{(J)}$ T (°C) ±50 °C | Chl$^{(V)}$ | T* (°C) | Ph$^{(M)}$ P (MPa) ±200 MPa | Chl-Ms$^{(V)}$ T (°C) ±50 °C | P (MPa) ±200 MPa | Perple_X T (°C) ±50 °C | P (MPa) ±100 MPa |
|---|---|---|---|---|---|---|---|---|---|---|
| TRI-21 | Tonalite mylonite N 48°30.412′ E 18°27.384′ | 329–386 | 338–395 | 298–396 | 365 | 400 | - | - | - | - |
| TR-CHD-1 | Tonalite blastomylonite N 48°30.475′ E 18°27.338′ | - | - | - | 350 | 550 | - | - | - | - |
| CHD-2 | Tonalite blastomylonite N 48°49.972; E 018°46.642′ | 328–333 | 273–339 | 311–315 | 308 | - | 249 | 600 | - | - |
| TR-46 | Permian schist N 48°29.309′ E 18°27.726′ | 295–310 | 300–315 | 271–306 | 300 | 750 | - | - | - | - |
| TR-35 | Permian meta-basalt N 48°33.255′ E 18°28.214′ | 377 | 376 | - | 376 | - | - | - | 370–400 | 580–700 |
| TR-33 | Permian basic meta-pyroclastics N 48°33.259′ E 18°28.26′ | 292–307 | 288–303 | 293–337 | 303 | 700 | - | - | - | - |
| TR-43 | Permian meta-basalt N 48°33.316′ E 18°27.729′ | 246–341 | 253–341 | 305–327 | 323 | - | - | - | - | - |
| CHD-3 | Greenschist N 48°49.694′; E 018°47.276′ | 303–355 | 303–359 | 286–369 | 330 | 550 | - | - | - | - |
| KAM-1 | Tonalite mylonite N 48°30.769′; E 018°30.469′ | 362–378 | 370–386 | 351–358 | 362 | - | 266 | 400 | - | - |
| LAM-1 | Tonalite mylonite N 48°30.733′; E 018°30.704′ | 356–377 | 361–383 | 338–383 | 365 | - | 270 | 450 | - | - |
| LAM-2A | Permian schist N 48°30.760′; E 018°30.916′ | 339–351 | 343–356 | 310–366 | 334 | - | 263–270 | 400–500 | - | - |
| NES-2 | Micaschist phyllonite N 48°52.665′; E 018°47.895′ | 324–370 | 327–376 | 285–347 | 342 | 400 | - | - | - | - |
| VYC-2 | Permian meta-arkose N 48°53.349′; E 018°43.234′ | - | - | - | 300 | 400 | - | - | - | - |
| LAM-2B | Permian meta-arkose N 48°30.760′; E 018°30.916′ | 359–371 | 367–379 | 332–398 | 360 | - | 257–259 | 470–600 | - | - |
| NEM-1 | Low Cretaceous slate N 48°48.412′; E 018°50.927′ | 297–370 | 303–381 | 346–350 | 337 | - | 337 | 400 | - | - |

Chlorite thermometry by [70]—(C), [71]—(J). Si-in-Ph barometry after [72]—(M) based on average Si-in-Ph content. Chl thermometry and Chl–Ms thermobarometry after [73]—(V). *—Average of calculated T or estimated T from mechanics of quartz dynamic recrystalization [105] in case of missing Chl.

Eastern Low Tatra Mts. and Upper Hron Valley Vápenica Nappe

### Petrography

The inferred subautochthonous Fatric Vápenica Nappe is composed of thinned granodiorite–tonalite basement and the Permian–Lower Cretaceous cover rocks with distinct tectono-metamorphic overprinting. Our research focused on (1) mylonitized meta-granodiorites to meta-tonalites (Figure 43A) and (2) the Permian meta-sediments and meta-volcanics (Figure 43B–D).

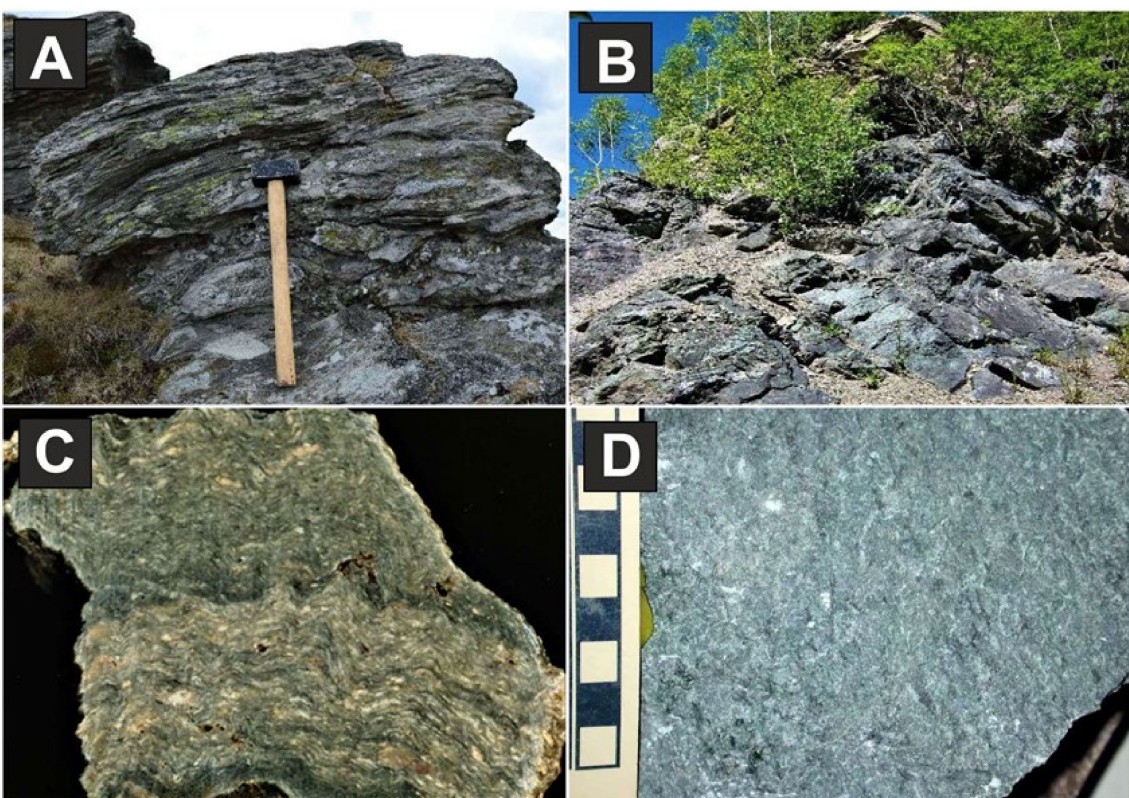

**Figure 43.** Meta-granitoids and Permian meta-sediments and meta-volcanics from the subautochthonous Fatricum **Vápenica Nappe** of the eastern Nízke Tatry Mts. (**A**) and Upper Hron Valley (**B**,**C**): (**A**) tonalite mylonites to blastomylonites (PHK-1, Priehybka pass N of Heľpa); (**B**,**C**) Permian arkosic meta-sandstone (**B**,**C**, HP-22, the top of the quarry and pale debris) with meta-basalt intercalation (**B**,**D**, HEL-1) in in a quarry S of Heľpa.

The Vápenica Nappe sheared and strongly tectonically thickness-reduced granitic sole macroscopically exhibits a variable extent of deformational recrystalization in the granodiorite to tonalite protomylonites, mylonites and schistose blastomylonites (Figure 43A), often with developed S–C structures indicating top-to-SE thrusting over the Veporic front. Blastomylonitic schistosity is covered by macroscopic fine-grained aggregates of white mica, Chl, Qtz, Pl/Ab, ±Ep and Cal and the Qtz–Fsp porphyroclasts prolongation is parallel to the (W)NW–(E)SE striking stretching lineation.

There are Permian pale schistose meta-sandstones (Figure 43B,C) with meta-rhyolitic and meta-basalt (Figure 43D) intercalations in a quarry at Heľpa. Syn-metamorphic medium- to coarse-grained Qtz–Ab veins, sub-parallel to metamorphic schistosity, occur in the arkosic meta-sandstones.

Tonalite and granodiorite mylonites of the Vápenica Nappe (Figure 44A–D) show mylonitic/blastomylonitic schistosity defined by fine-grained aggregates of white mica (Ph), Chl, Ab, Qtz, Ep–Czo, ±Cld and Cal surrounding the porphyroclasts of relic magmatic Pl, Qtz, Ttn and Bt. The Qtz and Fsp porphyroclasts are stretched and fractured, and the

fractures are infilled with the afore-mentioned newly formed phases. The brittle-ductile Fsp behaviour contrasts with ductile flow in the dynamically recrystalized Qtz aggregates.

The Permian sedimentary cover rocks are distinctly metamorphosed, and the metamorphic schistosity planes are defined by newly formed white mica (Ph) and Chl (Figure 44E–G). The recrystalized quartzitic matrix contains partly recrystalized Qtz, Fsp and Ms porphyroclasts. The Permian meta-basalt (HEL-1) intercalations in arkosic meta-sandstones exhibit a greenschist facies mineral assemblage of Chl, Ab, Act, Ep–Czo, Ph, Cal, Qtz (Figure 44H).

**BSE images, mineral chemistry and P–T estimates**

Examples of BSE images of the investigated rock textures (Figure 45) document the newly formed metamorphic phases used for P–T estimates.

Granodiorite to tonalite blastomylonites (Figure 45A) contains relics of albitized magmatic Fsp porphyroclasts, Qtz, chloritized Bt and rare Ms within the planar schistose domains defined by newly formed Ph–Chl aggregates associated with Qtz, Ab and Ep–Czo. The Permian meta-basalt shows the greenschist-facies mineral assemblage is composed of Chl, Ab, Act, Ep–Czo, Ph and Cal (Figure 45B).

The applied geothermobarometric methods are based on using the metamorphic Chl and Ph or Chl–Ph pairs. The chemical analyses of representative minerals from the **Vápenica Nappe** are available in Table S6.

Chlorite from the granodiorite to tonalite protomylonites to blastomylonites (MAD-1, PHK-1, PHK-2) has boundary composition between clinochlore and chamosite in the classification diagram (Figure 46). Chlorites contain up to 18.1–25.9 wt.% FeO and 12.9–17.8 wt.% MgO. The Fe/(Fe + Mg) ratio is 0.36–0.51. The MAD-1 sample has a clinochlore boundary composition with 23.9–26.3 wt.% FeO and 15.1–17.5 wt.% MgO. The $^{VI}$Al content reaches 1.21–1.60 *p.f.u.* in the studied samples. Chlorite from the Permian meta-basalts (HEL-1) and hosting arkosic meta-sandstones is clinochlore (Figure 46). Chlorites contain up to 12.9–19.0 wt.% FeO and 18.9–26.2 wt.% MgO. The Fe/(Fe + Mg) ratio is 0.21–0.34. The $^{VI}$Al content reaches 1.09–1.48 *p.f.u.* (Table S7).

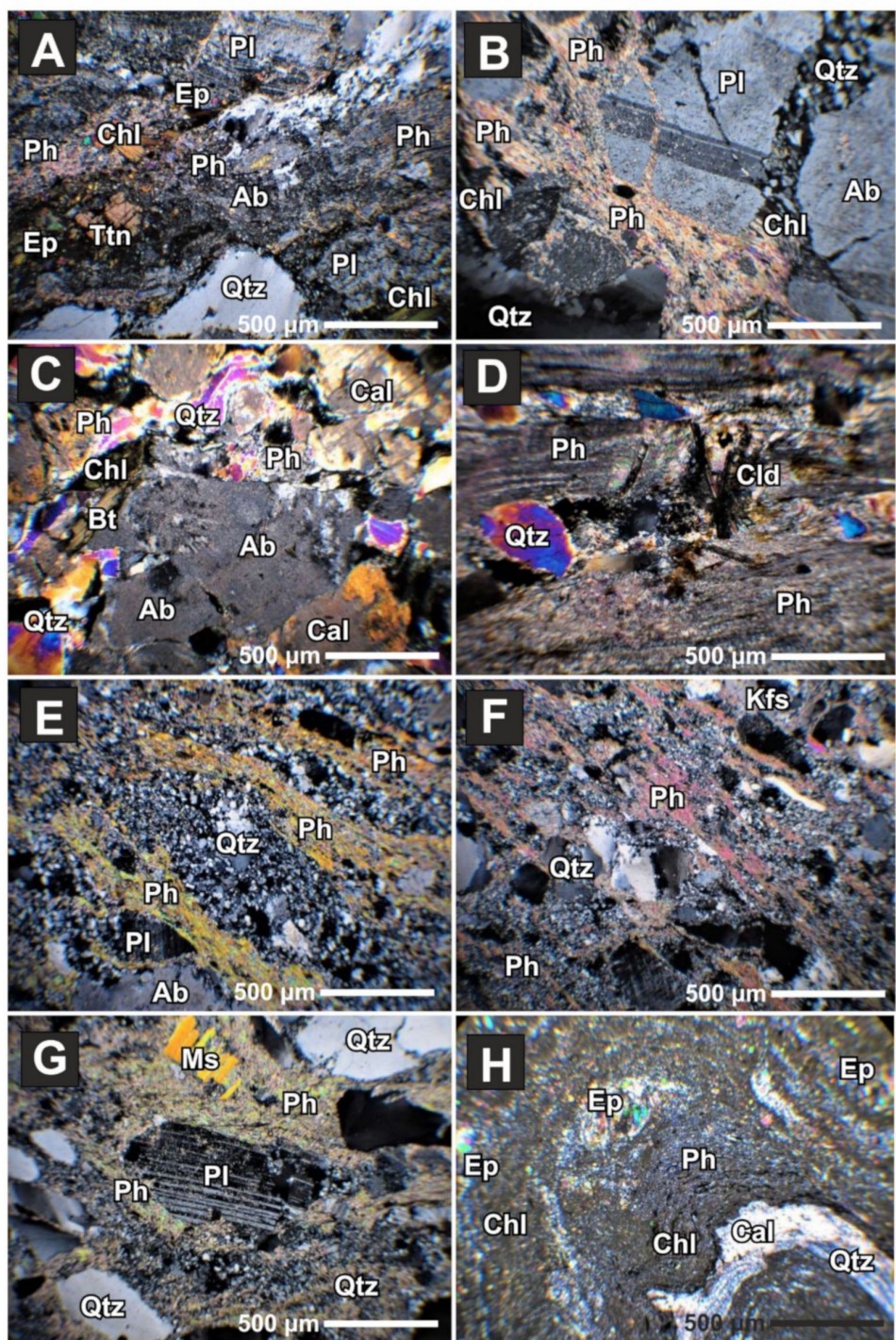

**Figure 44.** Microscopic textures from the subautochthonous Fatricum granitic basement and Permian cover rocks in the eastern Nízke Tatry Mts. **Vápenica Nappe**: (**A**) tonalite mylonite (VV-1, near Veľká Vápenica Hill); (**B**) granodiorite mylonite (DZ-114, N of Kráľova Hoľa Hill); (**C**) tonalite mylonite (MAD-1, Malužiná Valley below Domárka Hill); (**D**) Cld-bearing tonalite blastomylonite (PHK-2, Priehybka pass east of Veľká Vápenica Hill; (**E–G**) Permian schists: VA-23, at Veľká Vápenica Hill (**E**); VA-1, Vápenica Valley (**F**); VA-7, Vápenica Valley (**G**); (**H**) Permian meta-basalt (HEL-1, a quarry S of Heľpa). All pictures at *X* N.

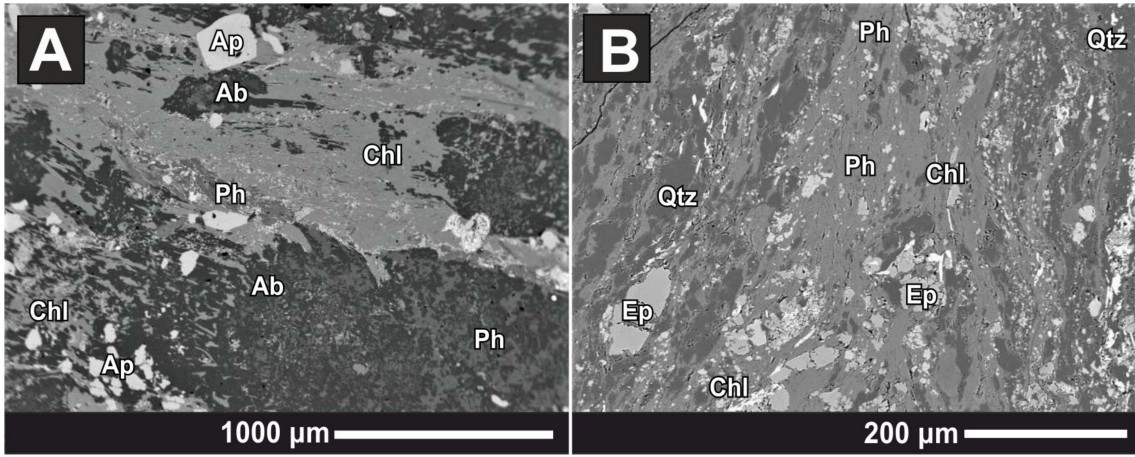

**Figure 45.** BSE images of white mica and Chl aggregates from subautochthonous Fatricum basement and cover rocks in the eastern Nízke Tatry Mts. **Vápenica Nappe**: (**A**) tonalite blastomylonite (PHK-1, Priehybka pass N of Heľpa); (**B**) Permian meta-basalt (HEL-1, a quarry S of Heľpa).

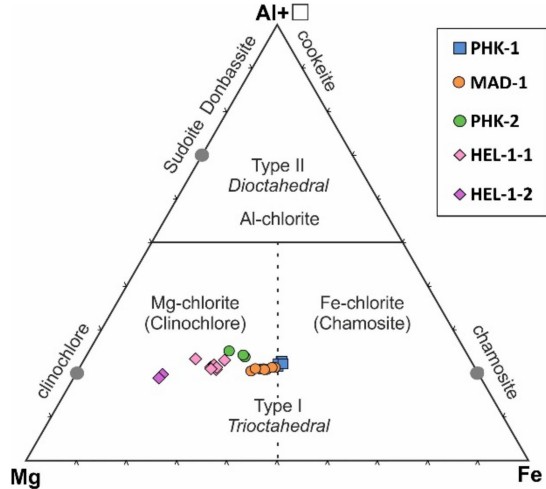

**Figure 46.** Al + □–Mg–Fe classification diagram of chlorite after [84] from subautochthonous Fatricum basement and cover rocks in the eastern Nízke Tatry Mts. and Upper Hron Valley **Vápenica Nappe**.

The metamorphosed basement and cover rocks of the eastern Nízke Tatry Mts. and Upper Hron Valley **Vápenica Nappe** contain Cel-Ms. Figure 47A highlights that all compositions of the white micas project along the muscovite–celadonite mixing-line in the classification diagram. White micas from granodiorite–tonalite blastomylonites have $SiO_2$ content from 45.5 to 51.1 wt.% (3.17–3.42 Si *p.f.u.*). FeO and MgO content varies in samples between 2.3–5.5 wt.% and 1.4–2.5 wt.%. $TiO_2$ content can be up to 1.7 wt.%. $SiO_2$ content in white micas from the Permian meta-sediments and meta-basalts varies between 45.7 and 50.6 wt.% (3.05–3.47 Si *p.f.u.*). FeO and MgO in the samples varies between 2.6–6.6 wt.% and 0.9–4.0 wt.% (Table S7). $TiO_2$ content can be up to 0.8 wt.%. Analyses of the white micas in Figure 47B fall within the higher anchi-metamorphism to lower greenschist facies with alkalis content of 0.9–1.0 K + Na *p.f.u.* and $K_2O$ up to 8.9–11.3 wt.%.

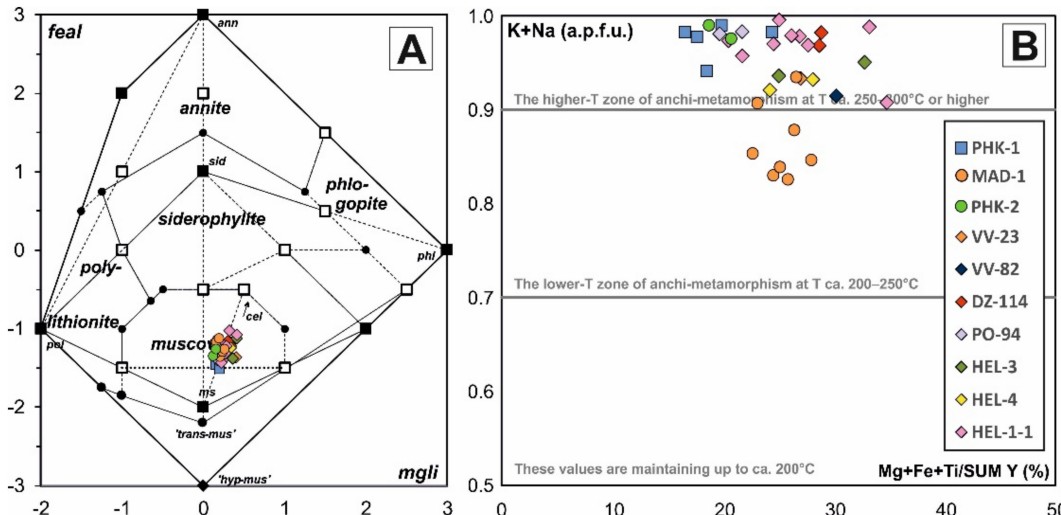

**Figure 47.** Classification diagram of white micas [85] from subautochthonous Fatricum basement and cover rocks in the eastern Nízke Tatry Mts. and Upper Hron Valley **Vápenica Nappe**: (**A**) parameters: mgli = Mg-Li, feal = $(Fe^{2+} + Fe^{3+} + Mn + Ti) - {}^{VI}Al^{3+}$; (**B**) K + Na vs. Mg + Fe + Ti/SUM Y (%) compositional trend of white micas. Values for lines taken from the works [69,103,104].

The geothermobarometry for the **Vápenica Nappe** basement tonalite to blastomylonites yielded 400–700 MPa pressure at the calculated 300–349 °C average Chl or estimated temperatures (Figure 48, Table 5). Similar pressures were achieved by meta-sediments in the 500–600 MPa range at estimated 300 °C temperature (Table 6).

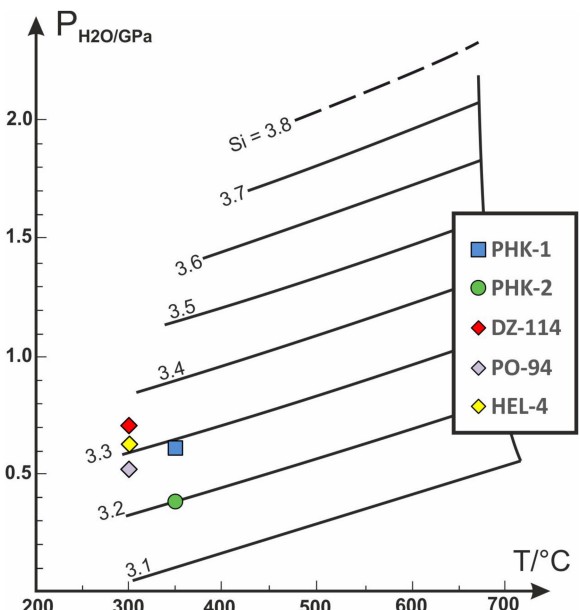

**Figure 48.** Pressure estimates diagram after [72] from subautochthonous Fatricum basement and cover rocks in the eastern Nízke Tatry Mts. **Vápenica Nappe**. Temperatures derived from Chl thermometry by [70,71,73] or estimated from mechanics of quartz dynamic recrystalization [105] in case of missing Chl in sample.

Figure 49 highlights the metamorphic P–T conditions of tonalite protomylonites to blastomylonites (MAD-1) of the **Vápenica Nappe** at 332–345 °C and 350–400 MPa using Chl–Ph pairs. This was accomplished by multi-equilibria Chl–Ph–Qtz–H$_2$O geothermobarometry [73].

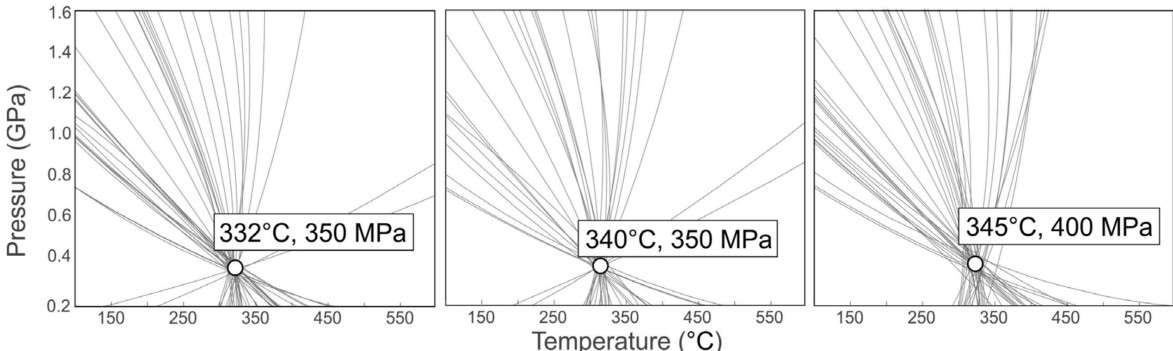

**Figure 49.** P–T diagrams with univariant curves and points intersecting with Chl–Ph pairs at equilibrium P–T conditions from the tonalite protomylonites to blastomylonites (MAD-1) of the **Vápenica Nappe**. Based on Chl–Ph–Qtz–H₂O geothermobarometry [73].

**Perple_X pseudosection** was used to estimate D1 stage P–T conditions of the Permian meta-basalts in the greenschist-facies metamorphic overprinting at 350–410 °C and 375–480 MPa pressure (Figure 50; Table 6). The calculated equilibrium mineral assemblage is Chl–Ms–Ep–Bt–Ttn–Ab. These estimates coincide with results obtained by conventional Chl and Ph geothermometry within the given error.

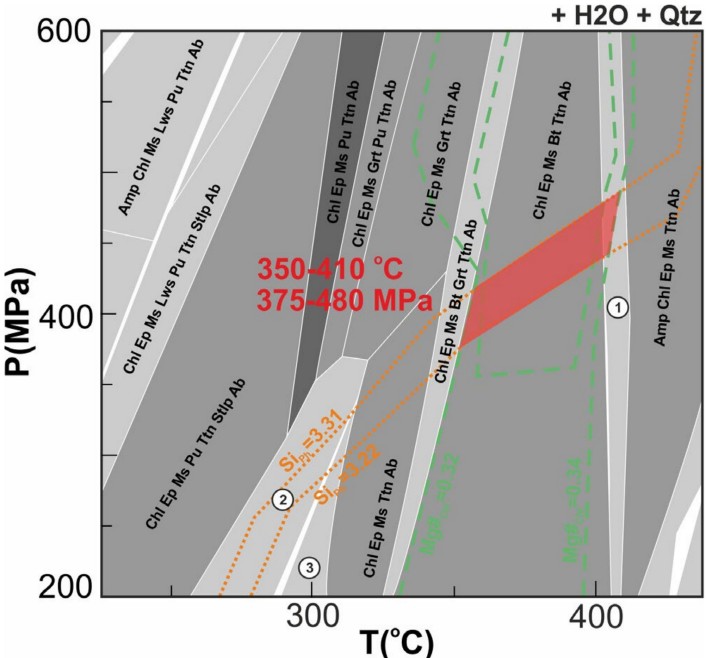

**Figure 50.** P–T pseudosection calculated by Perple_X thermodynamic software [92,93] for the D1 peak metamorphic stage of Permian meta-basalt (HEL-1-1) of the of the **Vápenica Nappe**, using isopleths intervals of Si-in-Ph 3.22–3.31 *p.f.u.* and Chl$_{Mg\#}$ 0.32–0.34. Numbers refers to mineral assemblage: **1:** Amp, Ch, Ep, Ms, Bt, Ttn Ab **2:** Chl, Ms, Ep, Pu, Stlp, Ttn, Ab, **3:** Chl, Ms, Ep, Pu, Pre, Ttn, Ab. Whole-rock composition used for modeling can be found in Table S1.

**Table 6.** Estimated P–T conditions from subautochthonous Fatricum basement and cover rocks in the eastern Nízke Tatry Mts. **Vápenica Nappe**.

| Sample | Sample Description GPS Coordinates | Chl(C) | Chl(J) T (°C) ±50 °C | Chl(V) | T* (°C) | Ph(M) P (MPa) ±200 MPa | Chl-Ms(V) T (°C) ±50 °C | P (MPa) ±200 MPa | Perple_X T (°C) ±50 °C | P (MPa) ±100 MPa |
|---|---|---|---|---|---|---|---|---|---|---|
| PHK-1 | Tonalite blastomylonite N 48°54.6266′; E 019°59.2838′ | 328–353 | 333–358 | 347–357 | 349 | 600 | - | - | - | - |
| MAD-1 | Tonalite blastomylonite N 48°56.0073′; E019°51.7641′ | 331–351 | 335–356 | 333–350 | 342 | - | 332–345 | 350–400 | - | - |
| HEL-1-1 | Permian meta-basalt N 48°50.79′ E 19°57.684′ | 302–346 | 303–346 | 261–396 | 323 | - | - | - | 350–410 | 375–480 |
| DZ-114 | Granodiorite mylonite N 48°54.325′ E 20°7.161′ | - | - | - | 300 | 700 | - | - | - | - |
| PHK-2 | Tonalite blastomylonite N 48°54.6266′; E 019°59.2838′ | 347–362 | 349–363 | 336–359 | 353 | 400 | - | - | - | - |
| PO-94 | Permian meta-arkose N 48°52.505′ E 20°1.891′ | - | - | - | 300 | 500 | - | - | - | - |
| HEL-4 | Permian meta-arkose N 48°50.79′ E 19°57.684′ | - | - | - | 300 | 600 | - | - | - | - |

Chlorite thermometry by [70]—(C), [71]—(J). Si-in-Ph barometry after [72]—(M) based on average Si-in-Ph content. Chl thermometry and Chl–Ms thermobarometry after [73]—(V). *—Average of calculated T or estimated T from mechanics of quartz dynamic recrystalization [105] in case of missing Chl.

## Nízke Tatry Mts. Krakľová Nappe

### Petrography

The south-Fatric Krakľová Nappe has predominantly Middle Variscan Unit micaschist overlain with layered amphibolites of the Upper Variscan Unit base. The Permian–Triassic cover is infolded in phyllonitized crystalline basement rocks. Our research focused on the Alpine metamorphic overprinting textures in (1) micaschist phyllonites (Figure 51A,C,D) and (2) Permian meta-volcanics and meta-granites (Figure 51B,E,F) with the Alpine newly formed greenschist facies metamorphic phases.

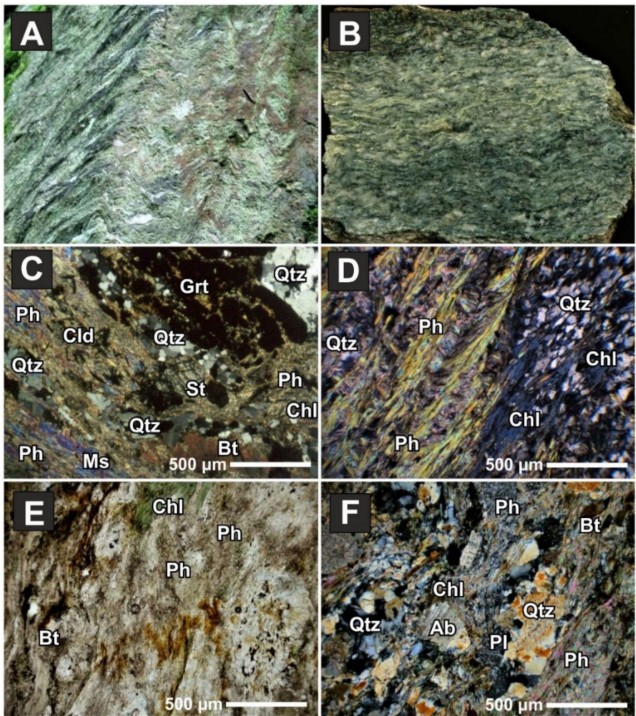

**Figure 51.** Macroscopic (**A**,**B**) and microscopic (**C**–**F**) textures from the subautochthonous southern Fatricum **Krakľová Nappe** in the eastern Nízke Tatry Mts.: (**A**) micaschist phyllonite (PR-2, Priehyba pass N of Heľpa); (**B**) Permian meta-dacite (BAC-1, Bacúch Valley); (**C**) phyllonitized St–Grt–Bt–Ms micaschist to Cld–Ph–Chl–Ab schist (PR-1, Priehyba pass N of Heľpa); (**D**) phyllonite (PR-2, S of Priehyba pass N of Heľpa); (**E**) Permian meta-dacite (BAC-1, Bacúch Valley); (**F**) Permian meta-granite (LED-1, Leňušská Valley at Beňuš). (**C**,**D**,**F**) pictures at *X* N; E picture at *II* N.

The phyllonite S1 schistosity planes are refolded and crosscut by steeply south-dipping S2 axial plane cleavage (Figure 51A). Crenulated metamorphic schistosity is observable in Permian meta-dacite (Figure 51B). Staurolite–garnet micaschists show a gradual transition to Cld–Ph–Chl–Ab schists in phyllonite zones with distinct axial-plane cleavage of syn-metamorphic microfolds (Figure 51C,D). Metamorphic schistosity of the Permian meta-volcanics and meta-granites is defined by greenschist facies minerals such as Chl, Ms and Ph, with Ab and less Ep group (Figure 51E,F).

**BSE images, mineral chemistry and P–T estimates**

Examples of BSE images of the investigated rock textures (Figure 52) document the newly formed metamorphic phases used for P–T estimates.

Micaschist phyllonites (Figure 52A,B) may contain relics of albitized Pl porphyroclasts, Qtz, chloritized Bt and phengitized St and Ms in the schistosity planes defined by newly formed Ph–Chl–Cld–Ab–Qtz aggregates. The Permian meta-dacite and meta-granite contain the greenschist-facies mineral assemblage of Chl, Ab, Act, Ep–Czo, Ph and Cal (Figure 52C,D).

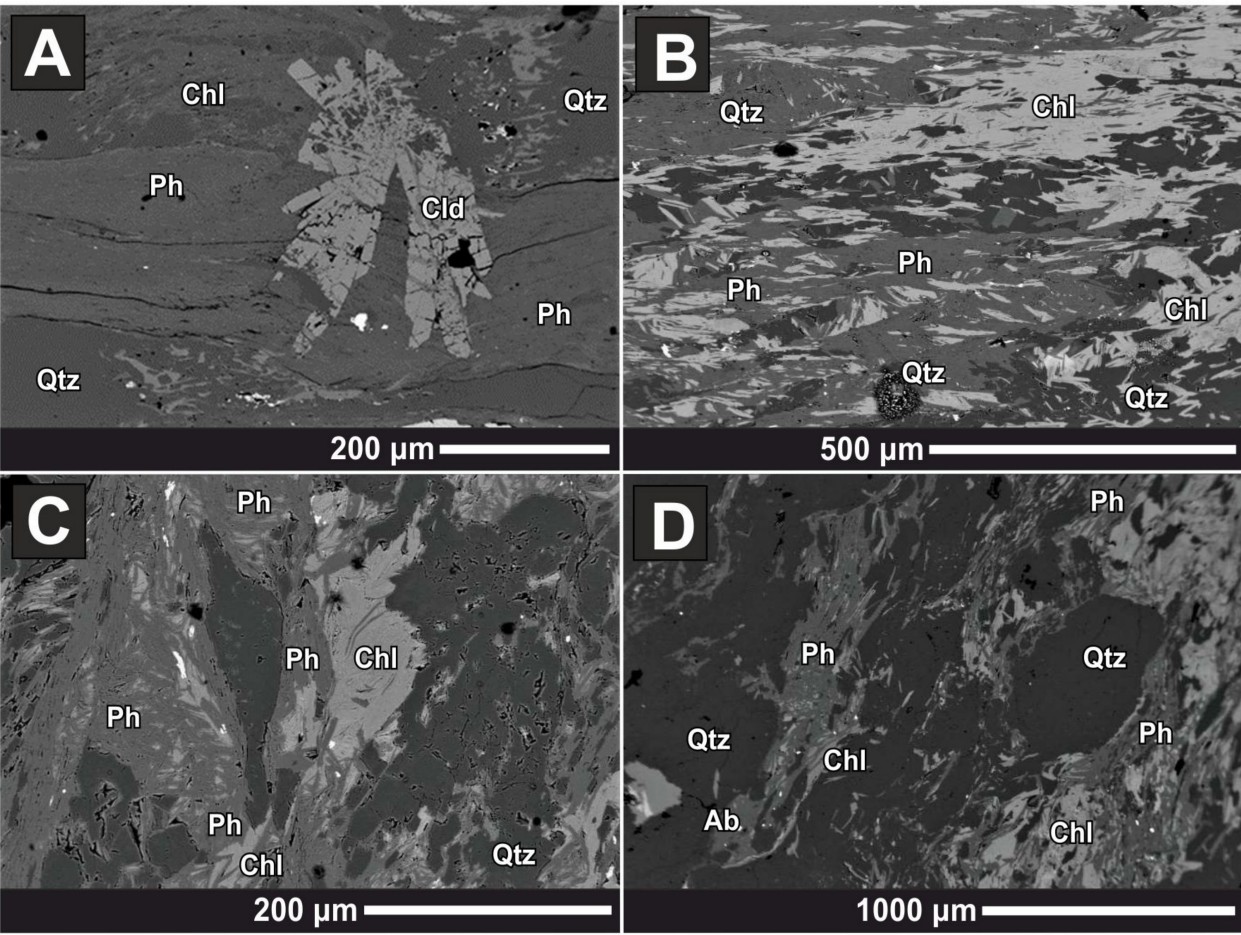

**Figure 52.** BSE images of greenschist facies micaschist phyllonites (**A,B**) and Permian meta-volcanics (**C**) and meta-granites (**D**) from the subautochthonous southern Fatricum **Krakľová Nappe** in the eastern Nízke Tatry Mts.: (**A**) Cld–Chl–Ph-Ab–Qtz phyllonite of a St–Grt–Ms–Bt mica schist (PR-1, Priehyba pass N of Heľpa); (**B**) Chl–Ph–Qtz phyllonite with steeply dipping in outcrop cleavage planes crosscutting the S1 schistosity (PR-2, S of Priehyba pass N of Heľpa); (**C**) refolded metamorphic schistosity of Permian meta-dacite (BAC-1, Bacúch Valley); (**D**) Permian meta-granite (LED-1, Leňušská Valley at Beňuš).

The applied geothermobarometric methods are based on the metamorphic Chl and Ph or Chl–Ph pairs. The chemical analyses of representative minerals from the **Krakľová Nappe** are available in Table S7.

Micaschist phyllonites of **Krakľová Nappe** (PR-2) contain chlorite with chamosite composition (Figure 53). Chlorites contain up to 27.9–32.03 wt.% FeO and 8.0–10.7 wt.% MgO. The Fe/(Fe + Mg) ratio is 0.62–0.64. The $^{VI}$Al content reaches 1.40–1.78 *p.f.u.* in the studied samples. Chlorite from the Permian meta-dacite (BAC-1) and meta-granite (s. LED-1) has transitional composition between clinochlore and chamosite endmembers with 23.6–27.9 wt.% FeO and 12.2–17.8 wt.% MgO. The Fe/(Fe + Mg) ratio is 0.43–0.56. The $^{VI}$Al content reaches 1.14–1.49 *p.f.u.* in the studied samples (Table S8).

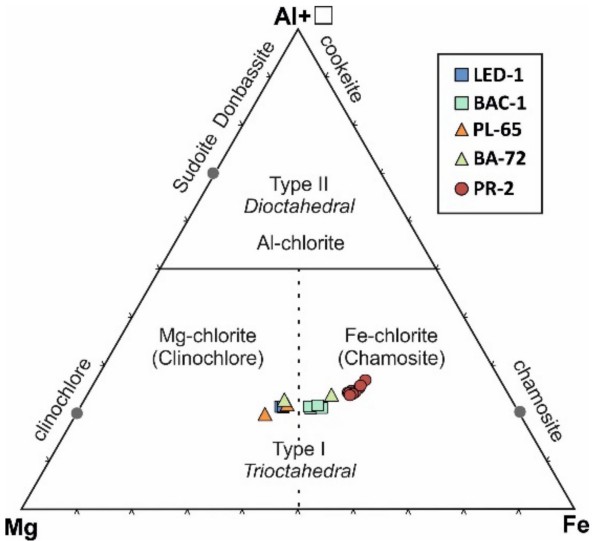

**Figure 53.** Al + □–Mg–Fe classification diagram of chlorite after [84] from subautochthonous southern Fatricum **Krakľová Nappe** in the eastern Nízke Tatry Mts.

The metamorphosed basement and cover rocks of the eastern Nízke Tatry Mts. **Krakľová Nappe** contain Cel-Ms. Figure 54A highlights that all chemical analyses of the white micas project along the muscovite–celadonite mixing-line in the classification diagram. SiO$_2$ content in white micas from the Permian meta-dacite and meta-granite varies between 48.0 and 50.4 wt.% (3.22–3.41 Si *p.f.u.*). FeO content is up to 5.3 wt.%. MgO is relatively low up to 3.6 wt.%. TiO$_2$ content is low in all samples up to 0.6 wt.%. No Cel-Ms was present in sample PR-2, only Ms (Figure 54A,B). Analyses of the white micas in Figure 54B mostly fall within higher anchi-metamorphism to lower greenschist facies with alkalis content of 0.9–1.0 K + Na *p.f.u.* and K$_2$O up to 10.3–11.3 wt. % (Table S8).

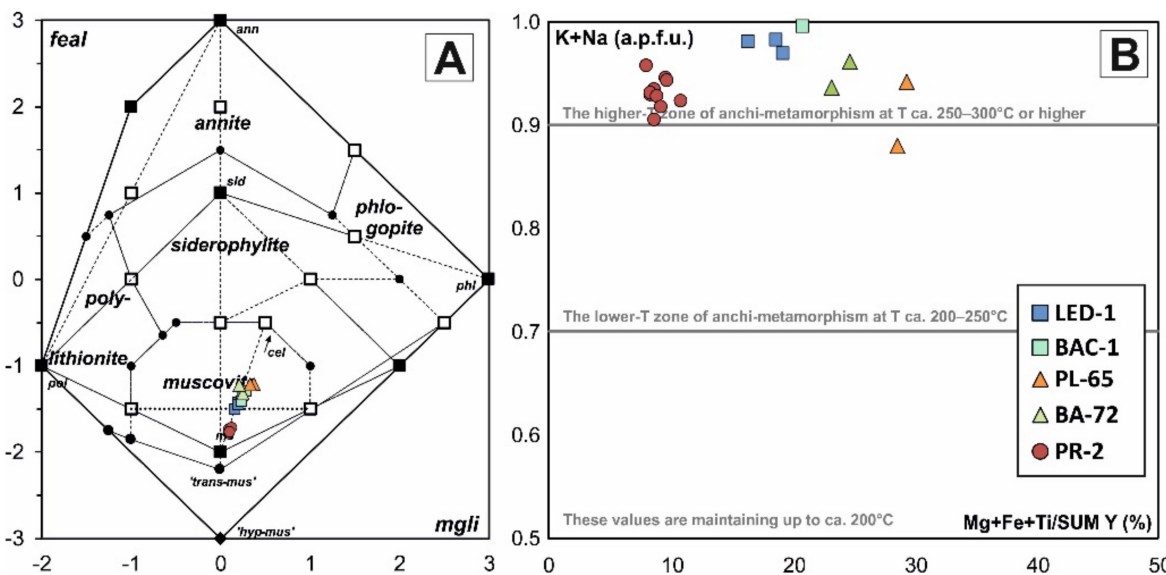

**Figure 54.** Classification diagram of white micas [85] from subautochthonous southern Fatricum **Krakľová Nappe** in the eastern Nízke Tatry Mts.: (**A**) parameters: mgli = Mg-Li, feal = $(Fe^{2+} + Fe^{3+} + Mn + Ti) - VIAl^{3+}$; (**B**) K + Na vs. Mg + Fe + Ti/SUM Y (%) compositional trend of white micas. Values for lines taken from the works of [69,103,104].

The geothermobarometry for the Permian meta-dacite and meta-granite of the **Krakľová Nappe** yielded 500–600 MP pressure at around 290–327 °C Chl crystalization temperature (Figure 55, Table 7).

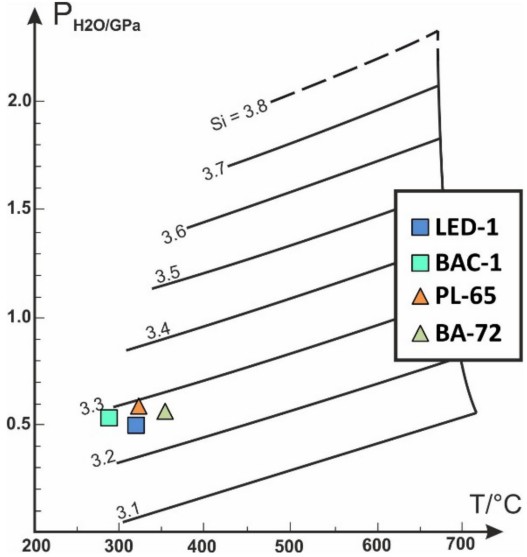

**Figure 55.** Pressure estimates diagram after [72] from subautochthonous southern Fatricum **Krakľová Nappe** in the eastern Nízke Tatry Mts. Temperatures derived from Chl thermometry by [70,71,73] or estimated from mechanics of quartz dynamic recrystalization [105] in case of missing Chl in sample.

**Table 7.** Estimated P–T conditions from subautochthonous Fatricum basement and cover rocks in the eastern Nízke Tatry Mts. **Krakľová Nappe**.

| Sample | Sample Description | Chl(C) | Chl(J) T (°C) ±50 °C | Chl(V) | T* (°C) | Ph(M) P (MPa) ±200 MPa |
|---|---|---|---|---|---|---|
| LED-1 | Permian meta-granite-porphyre N 48°51.8884′; E 019°45.3063′ | 321–334 | 327–338 | 312–324 | 327 | 500 |
| BAC-1 | Permian meta-dacite N 48°53.428′ E 19°48.059′ | 307–331 | 314–336 | 273–311 | 290 | 600 |
| PL-65 | Permian meta-rhyolite N 48°48.447′ E 19°29.871′ | 324–340 | 327–344 | 317–325 | 330 | 600 |
| BA-72 | Permian meta-rhyolite N4 8°53.299′ E 19°47.335′ | 351–369 | 358–373 | 330–342 | 354 | 550 |
| PR-2 | Micaschist phyllonite N 48°54.67′ E 19°56.893′ | 339–414 | 349–423 | 341–384 | 396 | - |

Chlorite thermometry by [70]—(C), [71]—(J). Si-in-Ph barometry after [72]—(M) based on average Si-in-Ph content. Chl thermometry and Chl–Ms thermobarometry after [73]—(V). *—Average of calculated T or estimated T from mechanics of quartz dynamic recrystalization [105] in case of missing Chl.

### 4.2. New Mica $^{40}Ar/^{39}Ar$ Age Data from Subautochthonous Fatricum and Infratatricum and Their Interpretation

Grains of Ph (HC-12 and SRB-1), Cel-Ms (ZI-3), Ms (SRB-1) and Bt (SRB-1) for $^{40}Ar/^{39}Ar$ dating were hand-picked from a 125–212 μm-size fraction (see the Section 3). Cel-Ms in sample ZI-3 seems to represent one deformation/recrystalization event or a D-stage (Figure 56A). Similarly, homogeneous Ph aggregates in sample HC-12 appear to be products of one event or a D-stage (Figure 56B). However, there are also visible dark 10–20 μm relic Ms flakes in the BSE image (Figure 56B). Sample SRB-1 from the hanging wall of a marble thrust sheet overlying C.R. type marly slates (Figure 8A) contains two grain-size Ph aggregates with most likely different ages. The dated coarse-grained Ph1 generation in the S1 schistosity is crosscut by shear bands with fine-grained Ph2 generation along the C-planes (Figure 56C,D). The $^{40}Ar/^{39}Ar$ geochronology results from all dated samples are shown in Figure 57A–E, Table 8, and detailed in Table S2.

White mica of Cel-Ms composition from the ZI-3 meta-quartzite sample yielded a mini-plateau age (with 57% of the total $^{39}Ar$ released) of 62.21 ± 0.31 Ma (MSWD = 0.43, p = 0.95) which is interpreted to constrain the growth of the mica or complete recrystalization of the D1 Ph to Cel-Ms during the D2 to D3 exhumation stages (Figure 57A).

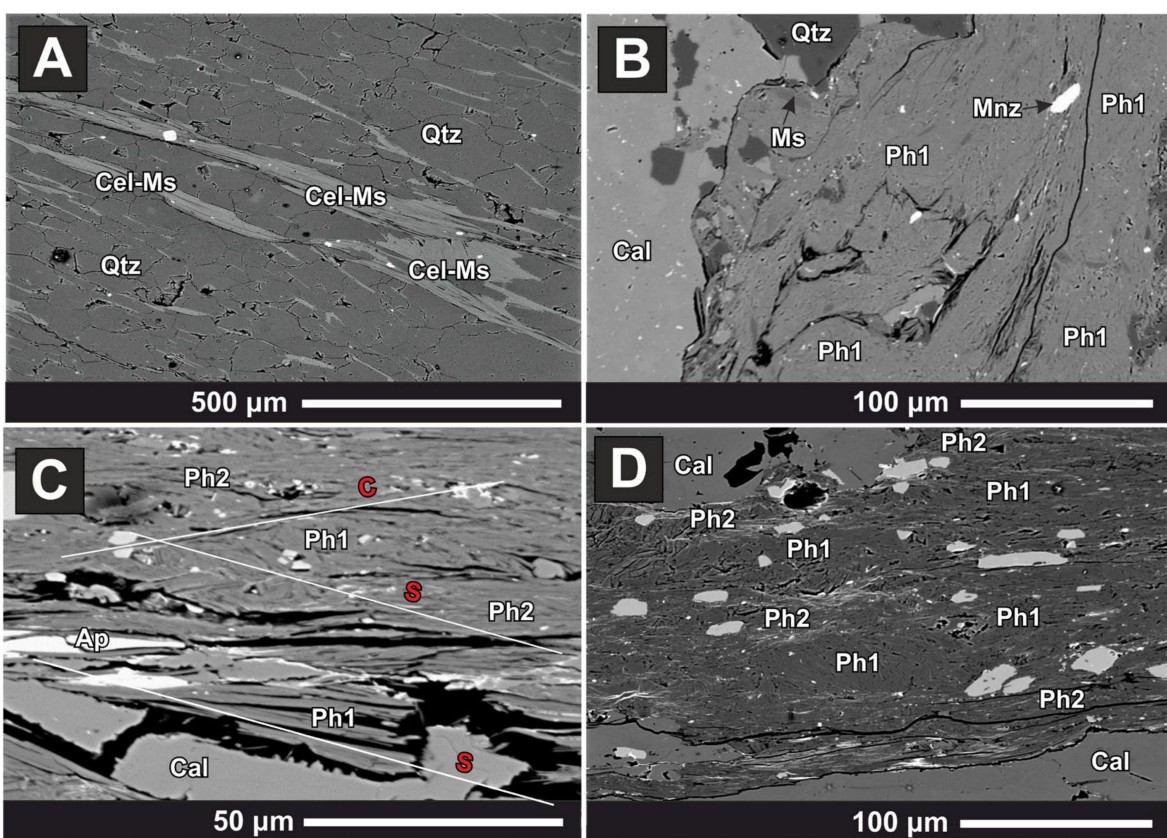

**Figure 56.** $^{40}$Ar/$^{39}$Ar dated white mica textures from the **subautochthonous Fatric Zobor Nappe** in the Tribeč Mts. Žirany quarry (**A**), and **higher Infratatric Hlohohec Nappe**—marble thrust sheet in the Považský Inovec Mts. Hlohovec Block at Hlohovec township (**B–D**). (**A**) ZI-3 Lower Triassic meta-quartzite with Cel-Ms aggregates oriented in metamorphic schistosity S1. The subautochthonou Fatricum Zobor Nappe; (**B**) HC-12 Lower Cretaceous(?) meta-marly carbonates with homogeneous, often refolded Ph aggregates in the S1 schistosity crosscut by the S2 micro-cleavage. The higher Infratatric Hlohohec Nappe; (**C,D**) SRB-1 Middle Triassic schistose marble mylonite with two Ph grain size aggregates. Ph1 oriented in the main S1 schistosity; Ph2 occurs in shear bands parallel to the C-planes. The higher Infratatric Hlohohec Nappe.

Phengite from the HC-12 meta-marl sample failed to produce a plateau age and yielded a stair-shaped $^{40}$Ar/$^{39}$Ar age spectrum with step ages around 85 and 125 Ma (Figure 57B). The sample resolved an initial age of approximately 84 Ma before a gradual increase to around 96 Ma, with 79% $^{39}$Ar released in 13 continuous steps. The step ages do not overlap because of precise errors resolved from high precision measurement. However, repeatable step ages ranging from 85 to 96 Ma from this sample show a consistency with the argon step ages produced from phengite sample SRB-1 (cf. below), before gradually increasing to an age of ca. 125 Ma in the last 6 steps with 12% of $^{39}$Ar released. This type of spectrum is typical in mixed argon gas from different reservoirs, and it may suggest two distinct mica generations, or two distinct tectono-thermal events recorded by a single mica generation.

Phengite from the SRB-1 marble mylonite sample did not yield a plateau age, but produced consistent apparent ages clustered around 90 Ma with 74% $^{39}$Ar released in 10 continuous steps (Figure 57C) which can be tentatively interpreted as recording the phengite growth age. This sample's failure to produce a plateau age is due to the high precision of each step-age analysis and a little overlapping of the precise age errors. The step-ages gradually increased in the last 7 steps with 8% $^{39}$Ar released and produced a stair-shaped $^{40}$Ar/$^{39}$Ar age spectrum reaching approximately 110 Ma. It can be considered that the failure to reveal a plateau with >70% of the total $^{39}$Ar released was due to the presence of excess Ar and/or alteration, or alternatively, the apparent age spectrum may be a mixture between two age members of ca. 110 and 70 Ma which would be removing

any geological significance to the ~90 Ma apparent cluster of step age around ~90 Ma. Furthermore, it is important to note that experimental works have shown that mixture of components with two different ages tend to produce sigmoidal-shape age spectrum such as this one [106]. A mixture interpretation is consistent with the presence of an older metamorphic Ph1 most likely related to the burial (D1) within the Albian–Cenomanian/Turonian accretionary wedge. On the contrary, a younger Ph2 is closely related to the late- to post-Campanian thrust fault formation over the C.R. of the Santonian to Early Campanian age (Figures 8A and 56C,D).

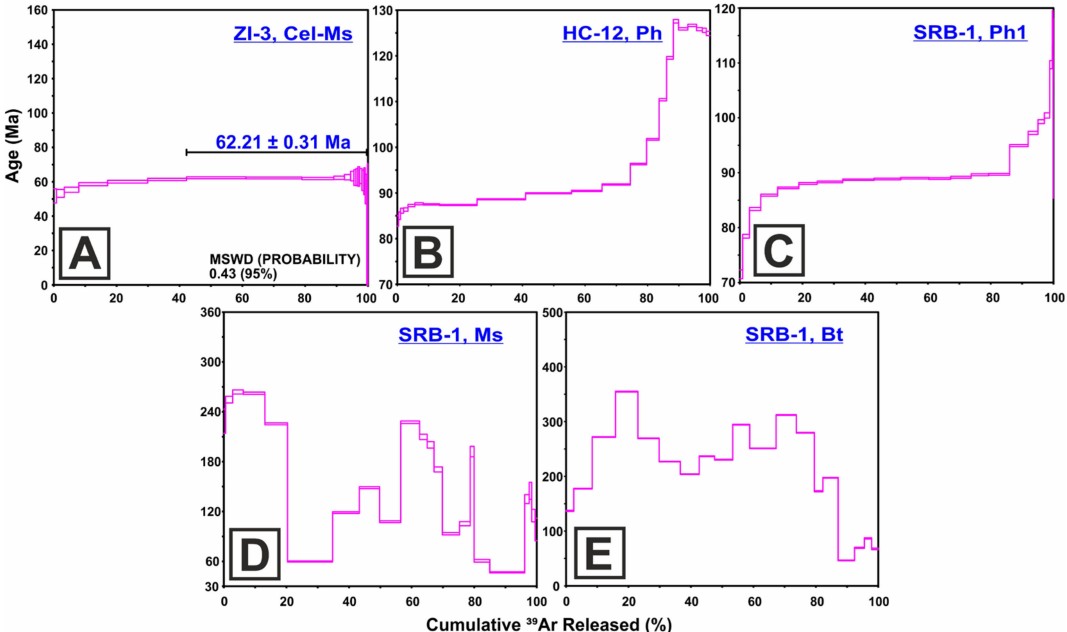

**Figure 57.** $^{40}$Ar/$^{39}$Ar spectra versus cumulative $^{39}$Ar released diagrams of dated samples from the subautochthonous Fatric **Zobor Nappe** in the Tribeč Mts. Žirany quarry (**A**), and higher Infratatric **Hlohohec Nappe**—marble thrust sheet in the Považský Inovec Mts. Hlohovec Block at Hlohovec township (**B**–**E**). (**A**) Cel-Ms of ZI-3 Lower Triassic meta-quartzite; (**B**) Ph of HC-12 Lower Cretaceous(?) meta-marly carbonate; (**C**) Ph(1) of SRB-1 Middle Triassic schistose marble mylonite; (**D**) Ms of SRB-1 schistose Middle Triassic marble mylonite; **E**) Bt of SRB-1 schistose Middle Triassic marble mylonite.

**Table 8.** Summary of $^{40}$Ar/$^{39}$Ar geochronology results. Complete results for all dated samples can be found in Table S2.

| Sample | Mineral | Plateau Age (Ma ± 2σ) | K/Ca | $^{39}$Ar(%) | No. of Steps | MSWD |
|---|---|---|---|---|---|---|
| ZI-3 | Cel-Ms | 62.21 ± 0.31 | 1 | 57 (mini-plateau) | 13 out of 22 | 0.43 |
| HC-12 | Ph | cannot calculate | - | - | 0 out of 22 | - |
| SRB-1 | Ph | cannot calculate | - | - | 0 out of 22 | - |
| SRB-1 | Ms | cannot calculate | - | - | - | - |
| SRB-1 | Bt | cannot calculate | - | - | 0 out of 19 | - |

Biotite and Ms from sample SRB-1 yielded complex, irregular apparent age spectra with no well-defined plateau ages typical for argon mixed from multiple reservoirs during step-heating experiments (e.g., [107]). Although these results prevent robust interpretation and may be disregarded, the apparent age spectra range from ca. 350 Ma (Bt) and ca. 270 Ma (Ms) to ca. 45 Ma. This range may suggest that the analysed samples contained multiple generations (i.e., partial recrystalization) of mica formed during the Variscan to Alpine orogenies. Most likely clastogenic relics of Ms and Bt in SRB-1 marble mylonites may indicate their origin from the Variscan basement rocks such as granites and gneisses. These

micas then have young ages due to Alpine tectono-metamorphic events registered in the Infratatricum.

## 5. Discussion

### 5.1. Applied Methods for Metamorphic P–T Estimates

Determining the P–T condition in low-grade metamorphic rocks (up to "epizone" T < 350 °C) is challenging because of the lack of indexing minerals and other factors (e.g., Bourdelle, 2021). This has mostly been achieved in the past decades by several calibrations of chlorite thermometers and phengite barometers (e.g., [70–72,108]). LT–LP chlorite thermometry, however, is still open to debate [109–111]. Two main approaches are still used today.

The conventional approach using $Al^{IV}$ increase, and octahedral vacancy decrease with increasing temperature originally described by [108] and several re-calibrations using different corrective factors based on the Fe/Mg ratio (e.g., [70,71,112]). A new thermo-dynamic approach has also been developed for low-grade metamorphic rocks [73], and this mostly involves a chlorite thermometer which enables $XFe^{3+}_{Chl}$ estimation at fixed pressure [73,113]. This approach is most rigorous, but it has problems with lacking thermo-dynamic data for Si-rich members which limits its use to chlorites with Si under 3 *p.f.u*.

A similar development is described for pressure estimates. Widening the misci-bility gap between muscovite and celadonite with either rising pressure or decreasing temperature was initially reported by Velde [114,115], thus confirming the pressure depen-dency of $Si^{4+}$ in Ph used in the geothermobarometry of phengite-bearing metamorphic rocks [72,116,117]. The reaction of this barometer ($3Qtz + 2Kfs + Phl + 2H_2O = 3AlCel$) depends on the coexistence of Ph with Phl, Kfs and Qtz and water activity ($P_{tot} = P_{H2O}$). We assume that the $H_2O = 1$ condition was met for the studied samples because they all contain a greater number of water-bearing minerals. The required assemblage was not present in all samples. However, according to [72], we can derive at least minimum pressures from the Ph that coexists with other Fe, Mg silicates such as Chl, and not with this limiting assemblage [72]. We selected the samples that seemingly follow both re-quirements for Si-in-Ph barometry. This method appears satisfactory for most of our studied samples with reasonably calculated pressures (Figures 19, 27, 34, 40, 48 and 55; Tables 2–7). However, some meta-sandstones (i.e., VV-82, VV-23) and clayey-rich layers in C.R. type marly slates (i.e., HC-2, PI-39) exhibit systematically very high or overestimated pressures (Figures 19, 27 and 48, Tables 2, 3, 6, S4, S5 and S7). We assume that this can be caused by the composition deviating from the barometer calibration or the very Si-rich nature of the protoliths which could cause additional Si-enrichment of the Cel-Ms (Ph) in some samples. Like chlorite, a thermodynamic approach on K-white mica thermobarome-try was employed by [91] which enabled P–T derivation by combining the Chl–Qtz–$H_2O$ thermometer and K–white mica Qtz–$H_2O$ barometer [73].

We therefore employed a combination of the described methods to avoid the limita-tions in these approaches and obtain the biggest possible dataset of P–T estimates. All the studied chlorites have the low $Al^{IV}$ typical for low-temperature crystalization [108] and all results from conventional thermometry are consistent within the error ($\pm 50$ °C) with the multi-equilibria thermometry by [73]. The results in Tables 2–5 show that some temperatures calculated by Chl–Qtz–$H_2O$ ($Chl^V$) equilibria give different temperature interval (usually lower) than the combination of Chl–Ph–Qtz–$H_2O$ ($Chl$–$Ms^{(V)}$) equilibria. We assume that this is due to the following: (1) The Chl–Qtz equilibrium is constrained by two independent equilibria while the Chl–Ph–Qtz equilibria are constrained by four equi-libria. Hence, when we combine these minerals, we increase P–T estimate robustness but simultaneously increase result-error. (2) The Chl–Qtz equilibrium is at lower temperatures than Chl–Ph–Qtz equilibria. This method was primarily calibrated for meta-pelitic rocks, so it is reasonable to assume that it is best for samples such as meta-sandstones, meta-arkoses and flysch slates and samples that are compositionally close to these lithotypes (Figures 18, 28 and 41); we were mostly unable to get realistic results, or none, when we

used this method for other samples. Moreover, the pressure estimates calculated by Chl–Ph–Qtz–H$_2$O geothermobarometry appear more reasonable if we expect the [72] barometry results to reflect maximum pressures for the studied samples mostly within 200 MPa error and greater than minimum pressure.

The achieved anchi-metamorphic to low-greenschist condition of the studied samples is further confirmed by typical transitional Ilt-Ph composition with decreased K + Na and K$_2$O contents (Figures 18, 25, 26, 33, 39 and 47, Tables S3–S8) [69,104]. Since many samples lack Chl, this and dynamical recrystalization of Qtz [105] were the only options to obtain at least estimated temperatures. The correlation between illite crystallinity and mica K$_2$O content was described from the diagenetic grade to the epizone by [103], where the occupancy of the interlayer position of the micas gradually increases and the K + Na content of the illite–muscovite increases [118–120]. In the sample SRB-1 with the two Ph generations (Figure 56C,D) the coarse-grained Ph1 shows the higher Si (3.4–3.5 *p.f.u.*) and Na + K values in comparison with the deformed Ph1 porphyroclasts and the recrystalized fine-grained Ph2 in the shear bands which show a decrease in Si to ca. 3.2 *p.f.u.* and Na + K (Figure 25). An increase of Fe + Mg in Ph2 (Figure 25) is likely also related to the deformational recrystalization (e.g., [121]).

Perple_X pseudosection modeling of the Považský Inovec Mts. Permian meta-basalt sample (PI-B3) was published by [3]. However, the changes in database and related solution models can produce smaller to larger differences in the calculated pseudosections e.g., [122], therefore we decided to update the pseudosection modeling with [95,96] solution models and the related [94] database to keep it in par with other modelled samples (HEL-1-1, TR-35, Figures 42 and 50). The calculated results of 330–345 °C and 460–580 MPa (Figure 21) overlap within the given errors with conditions of 308–315 °C and 490–610 MPa reported in [3]. Calculated stable mineral assemblages also match (Amp–Chl–Ms–Ab–Pu–Bt) except for Cpx which often tended to be over stabilized (especially in meta-basic rocks) when the older Cpx(HP) model was used [92,123]. This then caused most of the changes in the topology of both pseudosections.

The metamorphic conditions were most precisely estimated by Perple_X pseudosection modeling and these, however, are related only to the D1 syn-collision burial stage of the three chosen structural units, one from the Infratatricum (PI-B3, Figures 21 and 58) and two from the subautochthonous Fatricum (TR-35 and HEL-1-1, Figures 42, 50 and 58) to provide evidence on their participation in the inferred Albian–Cenomanian/Turonian accretionary wedge constrained by the new $^{40}$Ar/$^{39}$Ar geochronology data (Figure 57B,C). The estimated P–T conditions of the many remaining structural units using the various geothermobarometry methods discussed above are reviewed in Tables 1–7, and here we recommend considering the average values within the error.

The medium-pressure lower-temperature greenschist-facies metamorphic conditions were determined from the deepest buried fragments of the basement and the Upper Paleozoic cover rocks with similar mechanical behaviour. Some Lower Triassic quartzite fragments with newly formed Ph show similar burial depths compared to those containing metamorphic Ms instead of higher-pressure Ph. Similarly, most of the Middle Triassic marbles are Ph bearing. These rocks may record the D1 stage metamorphic conditions constrained by several white mica (Ph) plateau ages at ca. 100–90 Ma [3,7], including the newly obtained ages from the HC-12 and SRB-1 Infratatric samples. In contrast, most of the Jurassic to Cretaceous slates contain newly formed Ilt-Ph in addition to remnants of Ms microclasts, thus indicating relatively shallower burial depths and maximum higher anchi-metamorphic conditions. However, the Upper Cretaceous "flysch" sediments contain only Ilt to Ilt-Ph and reflect only the boundary diagenetic/lowest anchi-metamorphic conditions (Figure 58A–H).

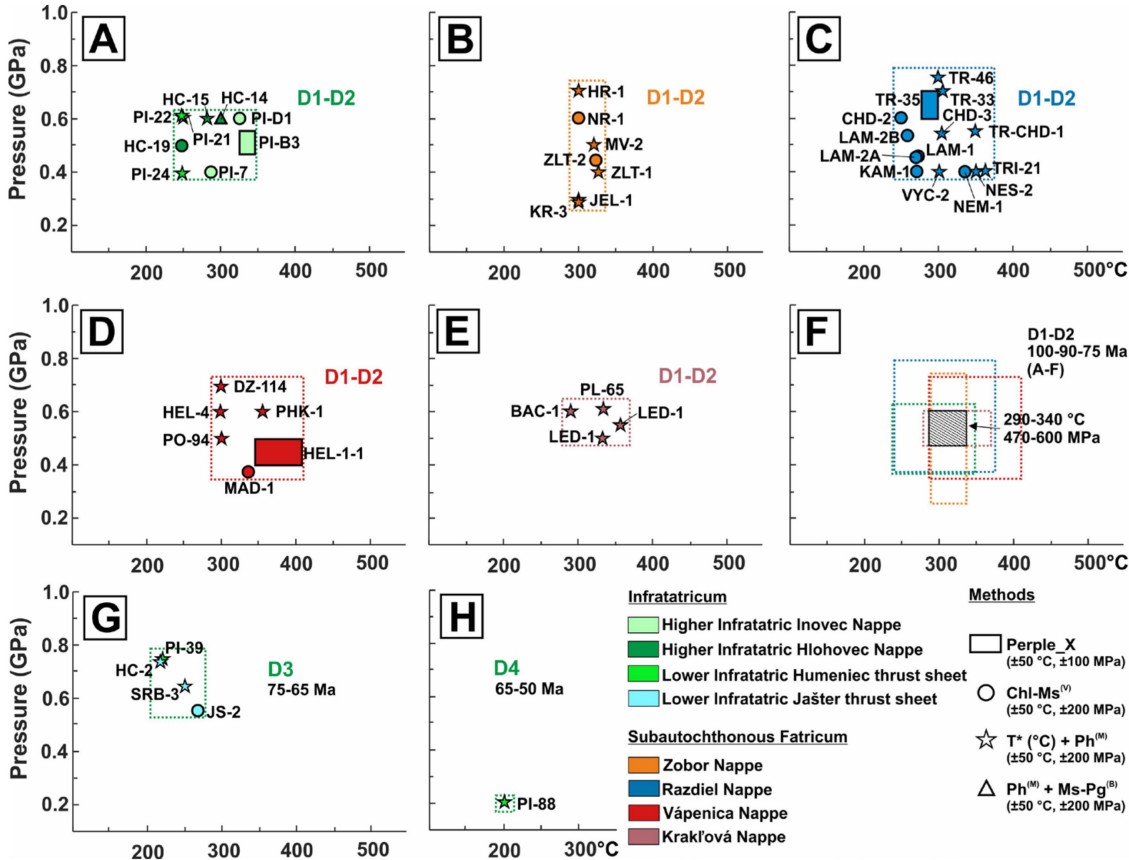

**Figure 58.** Review diagrams of P/T estimates from the Infratatricum (**A**,**G**,**H**) and subautochthonous Fatricum (**B**–**E**) structural units during the D1 to D4 evolutionary stages. (**A**–**E**) estimated the D1–D2 stages metamorphic P/T conditions from the investigated structural units; (**F**) summary diagram of the D1–D2 stages metamorphic P/T estimates in the higher anchi-metamorphic to lower greenschist facies; (**G**) constrained the D3 stage anchi-metamorphic conditions from the C.R. slates; (**H**) estimated the D4 stage lowest anchi-metamorphic conditions from "flysch" and legend to structural units. Ph[M]—Si-in-Ph barometry after [72], Chl-Ms[V]—Chl–Ms thermobarometry after [73], Ms-Pg[B]—Ms–Pg thermometry after [88], T*—average of calculated T or estimated T from mechanics of quartz dynamic recrystalization [105] in case of missing Chl.

We focused mainly on structural units exhibiting the metamorphic structures representative of the D1 stage (Figure 58A–E), although we locally could document also the metamorphic or mylonitic textures of the D2 exhumation. Despite some overlapping the P/T we can conclude that the pressure is decreasing from the wedge rear subautochthonous Fatricum to the Infratatricum wedge front. Cel-Ms from ZI-3 meta-quartzite of the subautochthonous Fatricum Zobor Nappe may document D1 shallower burial depths, while Ph from HR-1 meta-quartzite indicates much deeper part of this nappe. However, the age of the ZI-3 sample Cel-Ms of ca. 62 Ma does not fit to the D1 stage and may record a late D3 exhumation stage. Similar trend of the decreasing temperature towards the front is also characteristic of the accretionary wedges. This trend indicates a relatively deeper burial of the subautochthonous Fatricum due to inferred distinctly thinned a pre-metamorphic basement. Moreover, the exhuming ductilely deformed Veporic basement was a more effective overload in the rear part of the D1 wedge in comparison with thicker and mechanically harder Tatric sliver overthrusting the Infratatricum later, in the D3 stage (Figure 60, cross-section D). Thus, Figure 58F highlights partly overlapping P/T conditions of all accretionary wedge structural units for the D1–D2 stages. The C.R. marly slates record the D3 stage at higher anchi-metamorphic temperatures and medium pressures (Figure 58G). They occur as individual tectonic bodies of the lower Infratatric thrust sheets (Figures 5–8) due to inferred thinned rugged basement detachment and underthrusting below the higher

Infratatric nappes. Figure 58H highlights indistinct metamorphic overprinting of a frontal "flysch" basin infill during the D4 stage.

*5.2. Fatric–Tatric–Infratatric Evolutionary Stages Constrained by Published and New $^{40}Ar/^{39}Ar$ Ages*

The Fatricum in our model also includes the so-called "northern Veporicum" area, and these terms are synonymous for the same Late Cretaceous structural zone. We therefore recommend avoiding the use of "northern Veporicum" and substitute it by Fatricum as a structural unit equivalent to the Infratatricum, Tatricum, Veporicum and Gemericum which are composed of basement–cover structural units such as nappes, structural complexes or thrust sheets. We can distinguish these IWC northern zones paleogeographically with the following Jurassic–Cretaceous basin cover successions: Borinka Succession of the inferred southern Penninic oceanic margin, Orešany Succession of the Infratatricum continental margin facing the Penninic Realm, intra-continental Tatra and Šiprúň basin successions of the Tatricum and Vysoká, Manín, Zliechov, Veľký Bok and Humienec(?) type successions of the intra-continental Fatric Basin which transformed into a number of structural complexes described in this paper.

In addition, this "northern Veporicum" term does not fit with the mountain geography being part of the Nízke Tatry (Low Tatra) Mts., and the Veporicum occurs between the Late Cretaceous Pohorelá and Lubeník thrust faults in the Vepor Mts. The Foederata cover of the Veporicum is different to that of the Fatricum/"northern Veporicum" in the missing Jurassic–Cretaceous succession, most likely due to the commencing Jurassic compression and the formation of the southern Neotethyan wedge. Even the Upper Triassic successions are different (e.g., [8]). However, the typical allochthonous Fatricum occurs in the form of the Krížna Nappe system overlying the Tatricum; typically, in the Veľká and Malá Fatra Mts.

There is no lithostratigraphic or geochronologic evidence for "northern Veporicum" (~Fatricum) participation in the southern Neotethyan wedge with the Veporicum south of the Pohorelá line. In direct contrast, participation of the southern Veporicum in the Neotethyan Meliatic–Gemeric–Veporic wedge [4 and references therein] is constrained by the phyllonite white mica $^{40}Ar/^{39}Ar$ plateau age of ca. 124 Ma (124.2 ± 3.6 Ma; [7]) and the Permian meta-rhyolite in situ EPMA monazite age of ca. 130 Ma from the Vepor Mts. Fabova Hoľa Massif [35]. These ages suggest that only the Veporicum south of the Cretaceous Pohorelá thrust fault was incorporated in the southern wedge contemporaneously with the growing Fatric Basin towards the north. The formation of the southern Neotethyan wedge terminated at ca. 100 Ma [3,4,7,34], and the compression-deformation regime moved northwards to the Fatricum. Consequently, the Fatric Basin's extensionally thinned basement was underthrust beneath the Veporicum, thus initiating Veporic exhumation between approximately 100 and 85 Ma according to the white-mica $^{40}Ar/^{39}Ar$ plateau ages reported from the "southern" Veporicum [7,24]. The underthrust Fatricum may have blocked inferred an orthogonal N-ward collision which changed to a sinistral transpression in the exhuming south-Fatric rear (e.g., [41,124,125]), while the Veporicum south of the Pohorelá line underwent top-to-ESE(SE) structural unroofing [125,126]. Moreover, this important tectonic boundary indicates a change from the Early Cretaceous thick-skinned tectonics in the newly defined Veporicum to the Late Cretaceous thin-skinned tectonics in the Fatricum [41,43,125,127]. Thus, the Pohorelá (master) thrust fault is one of the major tectonic boundaries of the IWC separating the southern Neotethyan and the northern Atlantic (Alpine) Tethys wedges or continental microplates. Consequently, the whole zone between the Pohorelá and Čertovica thrust faults can be considered the root zone of huge Fatric nappe system.

The maximum (D1) white mica $^{40}Ar/^{39}Ar$ plateau ages of 93.2 ± 1.5 and 91.9 ± 2.3 Ma [7] in the newly defined Fatricum of the eastern Nízke Tatry Mts. north of the Pohorelá line overlap with the exhumation (D2) ages of the Veporicum south of the Pohorelá line, and these $^{40}Ar/^{39}Ar$ ages are consistent with the Fatric Basin closure and the Fatric nappe

formation. Therefore, the Veporicum may have acted as a "back-stop" domain during the closure of the Fatric Basin and the southward underthrusting of the Fatric basement–cover complexes underneath the Veporicum.

Both the Tribeč Mts. Zobor and Razdiel nappes could be the exhumed subautochthonous Fatricum. The Razdiel Nappe, composed of the strongly blastomylonitized basement and the Veľký Bok type Veľké Pole Permian–Mesozoic cover [45] most likely derived from the southern basin margin, resembling the Fatric, although finally south-vergently thrust Vápenica Nappe over the south-Fatric Ľubietová and Krakľová nappes. In contrast, the variably deformed protomylonitic to blastomylonitic granitic basement, with the metamorphosed shallow-water Mesozoic cover of the Tribeč Mts. Zobor Nappe, may be the northern basin margin, facially resembling the south-Tatric swell. The central deep-water Zliechov Succession was transformed into the superficial Fatric Krížna Nappe system because the strongly thinned central Fatric basement was almost completely underthrust below the Veporicum. An approximately 10-km thick reflection horizon of this underthrust inferred strongly blastomylonitized basement was recorded in reflection seismic cross-section 2T [127].

Both Tribeč nappes achieved a similar medium pressure greenschist facies tectono-metamorphic overprinting grade, with the $^{40}Ar/^{39}Ar$ white mica plateau ages of ca. 80–70 Ma [7,27]. The $^{40}Ar/^{39}Ar$ plateau age of 80.2 ± 1.1 Ma of Ph from the Razdiel Nappe blastomylonitized tonalites (CHD-1; [7]) is younger than the white mica plateau ages from the south-Fatric Krakľová Nappe (ca. 93–84 Ma; [7]), but older than the white mica plateau age from the blastomylonitized granitoids of the Zobor Nappe (71.4 ± 0.7 Ma; [27]). The age sequence between ca. 90 and 70 Ma may document gradual exhumation (D2 to D3) of the subautochthonous Fatricum, most likely in a transtensional regime and general SE-ward Veporicum sliding. The Zobor Nappe thus resembles an exhumed core-like complex from below the Razdiel Nappe which slid with the inferred Veporicum above them. Although the small Razdiel Nappe Mesozoic cover thrust sheets overlie the Zobor Nappe, the typical detached unmetamorphosed Fatric nappes generally overlie only the Tatricum towards the north ([18]; Map 1:50,000).

The new Cel-Ms $^{40}Ar/^{39}Ar$ mini-plateau age of ca. 62 Ma from the Zobor Nappe (ZI-3 sample of Lower Triassic meta-quartzite from Žirany quarry in the Tribeč Mts.) is the youngest age reported so far for the subautochthonous Fatricum. This age may suggest the wedge rear Fatricum tectono-thermal reactivation contemporary with the Infratatric trench-like Belice "flysch" trough closure, and the Infratatricum accretion to the Tatricum (Figures 59 and 60). The overall compression/transpression regime finally led to the (late-D3) fan-like structure formation of the subautochthonous Fatricum rear which bivergently overthrust the Tatricum and Veporicum. However, the ductile regime of the Vápenica Nappe back-thrusting towards the SE/ESE is limited by the ZFT ages of the southern Fatricum (former northern Veporicum) at ca. 75–65 Ma [77,128], the latter ages indicating the already advanced exhumation and cooling of the northern wedge hinterland.

The extensional exhumation (D2) of the subautochthonous Fatricum at ca. 90/85–75 Ma constrained by white mica $^{40}Ar/^{39}Ar$ data occurred during the collapse of the Albian–Cenomanian/Turonian accretionary wedge front, contemporary with the start of the south-Pennic subduction beneath the Infratatricum in the north. This agrees with the widespread regional extension related to Gosauian basin formation in the Eastern Alps (e.g., [24,129]) and it is most likely that this type of basin, including the Belice Basin, also formed in the IWC above the inferred Penninic–Váhic subduction zone.

The Tribeč Mts. ZFT and AFT ages are in the 70–55 Ma and 45–35 Ma intervals, respectively [76,77]. The pseudotachylytes in granodiorite from the Tribeč Mts. Zobor Nappe, dated at 58 and 46 Ma by $^{40}Ar/^{39}Ar$ document a significant Eocene seismogenic event [130] which we relate to the collapse of the IWC Eocene orogenic wedge.

The central Tatric sliver north of the subautochthonous Fatricum has weak tectono-thermal overprinting according to the typical ductile-brittle deformation structures. An older apparent ZFT age of ca. 102 Ma was determined from deformed granitoids of the

Žiar Mts. [131]. A similar $^{40}$Ar/$^{39}$Ar step age cluster of ca. 100 Ma was determined from the hanging wall blastomylonites of the Tatric Panská Javorina Nappe overlying the higher Infratatric Inovec Nappe in the middle part of the Považský Inovec Mts. [7]. These rare apparent ages may indicate burial of some parts of the central Tatricum sliver below the overthrusting early Fatric nappes in the forming Albian–Cenomanian/Turonian wedge. In contrast, the approximately 80–73 Ma $^{40}$Ar/$^{39}$Ar plateau ages from the Infratatricum of the Malé Karpaty Mts. [7] may indicate the formation of the frontal Infratatric edge adjoining the inferred subducted south-Penninic (Borinka Unit) oceanic margin.

A similar D1 stage at 101 $\pm$ 2.9 Ma plateau $^{40}$Ar/$^{39}$Ar and 102 $\pm$ 4 Ma ZFT ages were found from Permian meta-sandstone of the higher Infratatric Inovec Nappe [3] in the central part of the Považský Inovec Mts. The increased $^{40}$Ar/$^{39}$Ar plateau age of 114 $\pm$ 2.4 Ma determined from a Lower Cretaceous slate olistolith in the Upper Cretaceous "flysch" of the lower Infratatric Humienec thrust sheet most likely reflects recoil effect (l.c.).

The two new white mica samples with apparent $^{40}$Ar/$^{39}$Ar ages clustering at ca. 90 Ma from the higher Infratatric Hlohovec Nappe marble thrust sheet (HC-12 and SRB-1; Figure 56B,C and Figure 57B,C) may still belong to the D1 stage, indicating the present-day Infratatricum participation in the Albian–Cenomanian/Turonian wedge. This age is however, interpretable as a mixture from the D1 and D3 tectono-metamorphic stages recorded in the Ph-bearing D1 and D3 microstructures (Figure 56C,D). The $^{40}$Ar/$^{39}$Ar plateau ages of ca. 100–90 Ma [3,7] are consistent with termination of the Jurassic to Lower Cretaceous inferred Fatric Humienec Succession and stratigraphic hiatus before the Upper Cretaceous C.R. type marls and "flysch" deposition already in the Tatricum foreland Infratatricum. Moreover, metamorphosed olistolithic and tectonic fragments of the basement micaschists and the Permian to Lower Cretaceous cover rocks occur in almost unmetamorphosed "flysch" pelagites [3,57,132].

The formation of a fore-arc type Infratatric Belice Basin (D2) may suggest the southward subduction of the southern Penninicum beneath the Albian–Cenomanian/Turonian wedge at ca. 90–75 Ma that is the time of the Belice Basin formation. The change in hemipelagic sedimentation of the Couches-Rouges type marls (C.R.) to deep-water calciclastic and siliciclastic "flysch" sediments most likely indicates the change from an extension to compression regime in the latest Cretaceous and the transformation (D3) of the Belice Basin to a "flysch" trough. The D3 compression may have been accompanied by underthrusting of the C.R. marls underneath the higher Infratatric Inovec and Hlohovec nappe system and the lower anchi-metamorphic slaty cleavage formation in the C.R. This deformation regime change might have been related to ceasing foreland Penninic subduction, and thus the supra-subduction extension termination. However, direct superposition of the "flysch" on the C.R. type marls was not observed, and the marble thrust sheet of the Hlohovec Nappe, for example, directly overlies (D3) the C.R. anchi-metamorphosed marls. Similarly, the Inovec Nappe Hradisko Hill Triassic cover thrust sheet overlies the "flysch" and the lower anchi-metamorphosed C.R. marls fragments north of Humienec Hill (Figures 5–8). In addition, the differences in the metamorphic overprinting may indicate different burial ages of these Upper Cretaceous successions: the late Campanian to Maastrichtian at ca. 75–65 Ma for the C.R. marls (D3), and the Paleocene to Eocene at ca. 65–50 Ma for the "flysch" (D4). A lower anchi-metamorphic overprinting of the C.R. contrasts with the diagenesis to lowest anchi-metamorphic overprinting of the "flysch".

Finally, during the D4 stage, the higher anchi-metamorphic to low-grade basement and cover rocks of the higher Infratatric Inovec and Hlohovec nappes and the hanging wall Upper Cretaceous anchi-metamorphosed C.R. overthrust "flysch" sediments. The rugged basement fragments of the lower Infratatric Humienec and Jašter thrust sheets often occur directly in the Upper Cretaceous sediments, and the whole arrangement thus has a mélange character after the D4 stage, observable for example, in the Hlohovec Block and north of the Humienec and Hradisko hills in the Selec Block (Figures 5–8).

The Infratatricum exhumation at ca. 50–40 Ma (D5) was determined by the 48.3 $\pm$ 2.2 Ma Cel-Ms $^{40}$Ar/$^{39}$Ar plateau age from the Tatricum hanging wall leucogranitic blastomy-

lonites (PI-17HZ; [7]), and the 57–37 Ma ZFT ages from the Infratatricum footwall [3]. The latter event is most likely related to collapse of the Eocene wedge and development of wedge-top basins in response to subduction of the north-Penninic Magura Basin of the Outer Western Carpathians underneath the IWC (cf. [133]). The main interval of ZFT ages of ca. 55–35 Ma [3,77] suggests the D5 Eocene exhumation and cooling of the Infratatricum in the Selec Block. The AFT ages of 21–13 Ma [76,77] indicate final Miocene exhumation and cooling of the Infratatricum and the overlying Tatricum of the Bojná Block. ZFT ages are missing in the Hlohovec Block, and the AFT ages are in the 50–40 Ma interval [76,77].

Characteristic fold-and-thrust tectonic style is observable in the Infratricum and sub-autochthonous Fatricum with variable metamorphic overprinting grade (Section 4.1) of structural units (Figure 58). The lower-temperature thin-skinned tectonics [134,135] is typical of the Albian–Cenomanian/Turonian accretionary wedge. It is inferred that the syn-collisional burial (D1) was accompanied by detachment faulting and decoupling of the basement and cover rocks due to different mechanical behaviour and buoyancy. Moreover, several mechanically weak lithological zones may have been utilized at the formation of the Mesozoic Fatric nappes [15]. Blastomylonitic zones in the basement rocks are indicators of such zones typically developed in the subautochthonous Fatricum, Infratatricum and the base of the Tatricum (Sections 2 and 4).

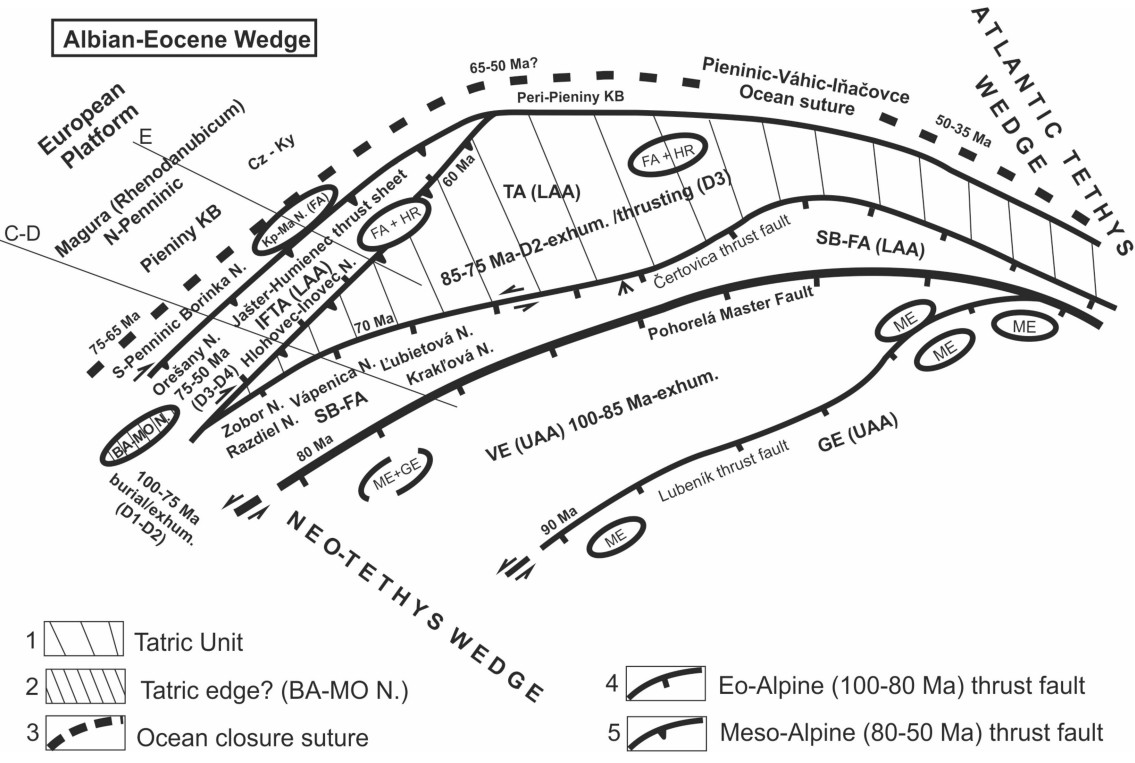

**Figure 59.** Palinspastic sketch of the Albian–Eocene orogenic wedge of the Inner Western Carpathians. KB—Klippen Belt, Cz—Czorsztyn Unit, Ky—Kysuca Unit, Kp-Ma—Klape–Manín Nappe, FA—Fatric Nappe, HR—Hronic Nappe, GE—Gemericum, IFTA—Infratatricum, TA—Tatricum, BA–MO N.—(Hainburg/)Bratislava–Modra Nappe, SB-FA—subautochthonous Fatricum, VE—Veporicum, ME—Meliaticum, LAA—Lower Austroalpine analogues, UAA—Upper Austroalpine analogues. C, D, E—cross-section lines to evolutionary stages of the Albian–Eocene orogenic wedge in Figure 60.

The Infratatricum resembles the "supratatric" subautochthonous Fatricum in many aspects, especially for example when considering the similarities in the tectonic structures set out in the Sections 2 and 4, and the D1 stage metamorphic conditions depicted in Figure 58. Such close relationships between the Infratatricum and subautochthonous Fatricum could be evidence of a close tectonic position of the higher Infratatric and subautochthonous Fatric basement-cover nappes in the Albian–Cenomanian/Turonian wedge during the D1 stage.

So far geological interpretations could not explain the older ca. 100 Ma Ar/Ar ages in the north (in the higher Infratatric nappes) within this N-ward progressing accretionary wedge. Moreover, considering Tatricum overload on the Infratricum at ca. 100 Ma contradicts the stratigraphic extent of the northern Tatricum autochthonous Mesozoic cover up to the Turonian in the Tatra and Veľká Fatra Mts. [64,65]. In addition, the central Tatricum sliver as mechanically strong and internally without a distinct tectono-metamorphic overprint could hardly been separating two domains of similar but much more distinct overprint in the wedge D1 stage. This is fact that the D1 Ph ages: 101.2 ± 2.9 Ma plateau age (with 94% $^{39}$Ar released) determined from the higher Infratatric Inovec Nappe Permian meta-sandstone and 106.2 ± 3.7 Ma plateau age (with 81% $^{39}$Ar released) from the Inovec Nappe Lower Cretaceous slate olistolith in Upper Cretaceous "flysch" of the lower Infratatric Humienec thrust sheet are about 10 Ma older than the approximate 90 Ma ages determined from the Krakľová Nappe in the south [3,7]. This may indicate that the present-day higher Infratatric nappes and the Klape nappe with the Albian–Cenomanian "flysch" containing the Neotethyan wedge material [3,5,13] both derived from the area south of the Tatricum (Figure 59). Although the Albian–Cenomanian "flysch" in the Infratatricum of the Považský Inovec Mts. is missing, it occurs in the close Infratatric foreland Klape Unit (Figure 60), and these inferred early Fatric nappes must have overthrust the Tatric sliver terminated edges before the Belice Basin opening in the late Turonian/Coniacian. Evidence for this is provided by the C.R. type marls occurring only north of the Tatricum which secondarily overthrust the Infratatricum in the D3 stage (Figures 59 and 60). This is also suggested by the (pre-D3) deep-water Turonian–Santonian succession [17] deposited at the Tatricum northern edge which is now exposed in the middle part of the Považský Inovec Mts. Bojná Block. Therefore, a separated (Humienec-type?) part of the Fatric Basin s.l. in direct proximity to the Neotethyan wedge cannot be excluded. The Pohorelá thrust fault may be a suture zone which formed after this basin closure and expulsion of the present-day higher Infratatric and the Klape nappes. The Humienec Basin may then have been part of a more complex intracontinental Fatric riftogenic basin system along the Tatricum southern margin, while an oceanic Váhic–S-Penninic riftogenic basin ran around the Tatricum northern margin, and both basin systems formed part of the Atlantic (Alpine) Tethys. Although, this paragraph's speculations of the first author may be not soundly based at this investigation stage, this is an alternative interpretation of the very close structural, metamorphic, and geochronological relationships between the Infratatricum and the "supratatric" subautochthonous Fatricum in the proposed evolutionary model depicted in Figure 60.

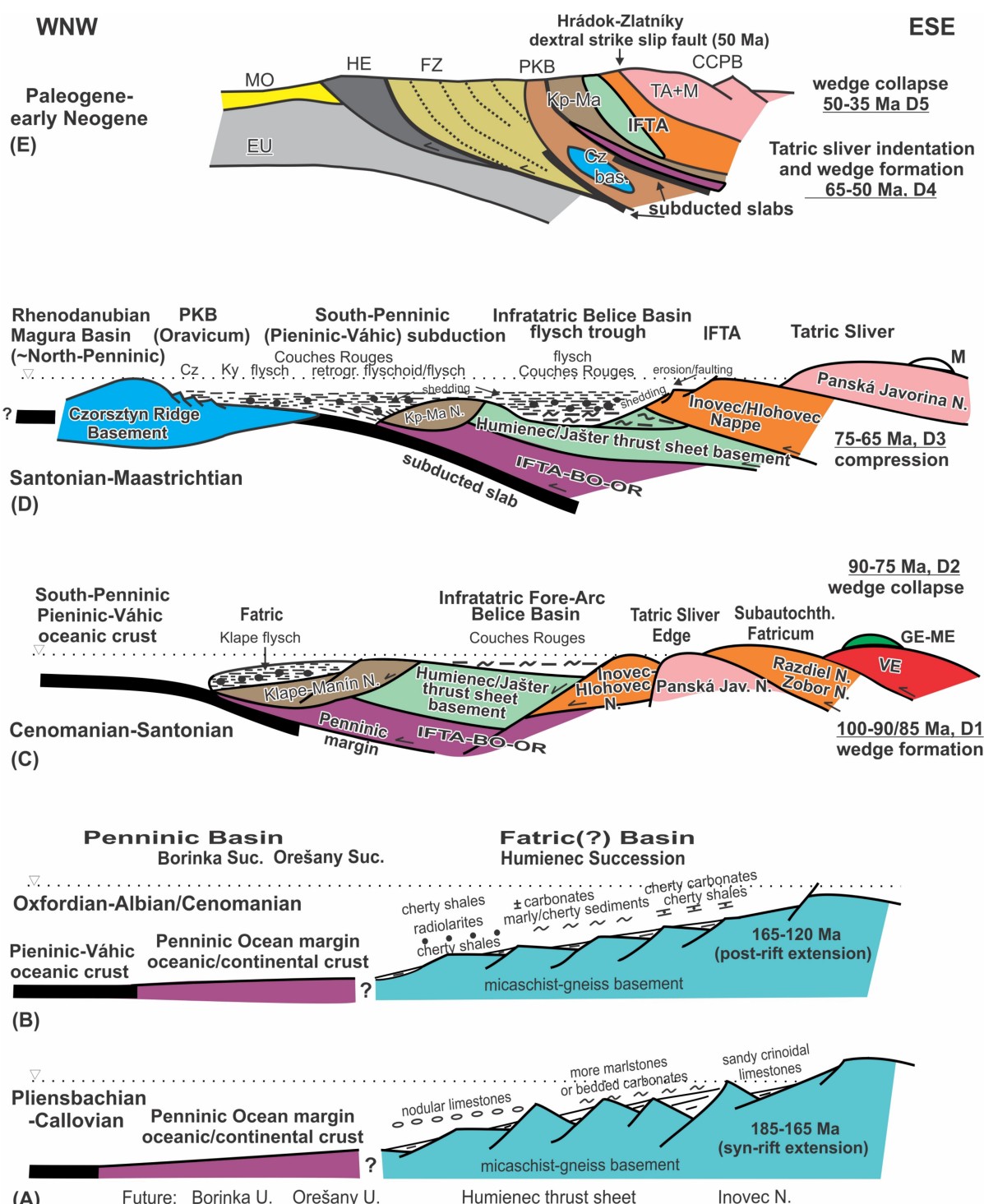

**Figure 60.** Evolutionary model of the Albian–Eocene orogenic wedge formation in the IWC, approximately along cross-sections in Figure 59. EU—European Plate; MO—Neogene molasse; HE—Helveticum; FZ—Outer Fysch Zone; CCPB—Central Carpathian Paleogene Basin; PKB—Pieniny Klippen Belt; Cz—Czorsztyn Unit; Cz bas.—Czorsztyn Ridge basement; Ky—Kysuca Unit, Kp—Klape Unit; Ma—Manín Unit; Penninic margin—inferred adjacent continental zone to oceanic crust; TA + M—Tatricum and Mesozoic Fatric and Hronic nappes; M—Mesozoic Fatric and Hronic nappes; IFTA—Infratatricum; TA—Tatricum, VE—Veporicum; GE—Gemericum; ME—Meliaticum. A–E—cross-sections of evolutionary stages of the Albian–Eocene orogenic wedge formation. Cross-section lines of evolutionary stages C–D and E are depicted in Figure 59.

## 6. Conclusions

- Thermodynamic modeling, geothermobarometry and white mica $^{40}$Ar/$^{39}$Ar dating constrain the metamorphic conditions and evolutionary (D) stages of the structural units which formed due to closure of the Jurassic–Cretaceous Atlantic (Alpine) Tethys basin system and subsequent opening and closure the Upper Cretaceous basin system in the northern part of the IWC (Figures 58–60).

- The metamorphic conditions determined from the Permian meta-basalts of the Infratatricum and subautochthonous Fatricum by Perple_X modeling are very similar (within the error of the applied methods), indicating the D1 stage or syn-collisional burial in the Albian–Cenomanian/Turonian accretional wedge. The subautochthonous Fatric structural units provided the pressures of 400–700 MPa at 250–410 °C by the Perple_X modeling (Razdiel and Vápenica nappes), Chl–Ph geothermobarometry, Chl geothermometry and Ph geobarometry. The higher Infratatric nappes achieved 400–600 MPa at 250–350 °C that was constrained by the Perple_X modeling (Inovec Nappe), Chl–Ph geothermobarometry, Chl and Ph–Pg geothermometry and Ph geobarometry. Such small differences in the estimated P/T conditions for most of the nappes after the D1 stage indicate their neighbouring position in the Albian–Cenomanian/Turonian wedge, in general, south of the Tatricum and beneath the Veporicum (Figure 60, cross-section C). The present-day higher Infratatric Inovec and Hlohovec nappes and the Manín–Klape nappe units achieved the Infratatric position in the D1 wedge front by the overthrusting the Tatricum terminated edges yet before the opening of the Belice Basin. The Infratatric nappes were secondarily overthrust by the Tatricum during the D3 stage (Figures 59 and 60, cross-section D).

- The D1 event was $^{40}$Ar/$^{39}$Ar dated on phengitic white mica to two ca. 90 Ma apparent step ages from the higher Infratatric Hlohovec Nappe marble thrust sheet (SRB-1 and HC-12 samples). We interpret this age of ca. 90 Ma as a mixture of Ph ages from the D1 and the D3 tectono-metamorphic stages. A Cel-Ms age of ca. 62 Ma from the subautochthonous Fatric Zobor Nappe (ZI-3 meta-quartzite sample) indicates advanced (late-D3) exhumation of the D1 wedge rear.

- The average metamorphic conditions determined from the lower Infratatric Humienec and Jašter thrust sheets C.R. type marly slates with newly formed Ilt-Ph are ca. 220–270 °C and 600–750 MPa by the Chl–Ph geothermobarometry and Ph geobarometry. These conditions are related to the compressional D3 stage or burial of the C.R. underneath the exhumed (D2) higher Infratatric nappes at ca. 75–65 Ma. The newly formed Ilt to Ilt-Ph in the "flysch" pelagites indicates the lowest anchi-metamorphic conditions of the D4 stage.

- The lower Infratatric Humienec and Jašter thrust sheets with the C.R. slates, incorporating the rugged thinned basement (micaschists or granitoids with the cover rocks), underlie the higher Infratatric Inovec and Hlohovec nappes after the D3 stage. The Belice "flysch" trough closure at 65–50 Ma was followed by the Infratatricum accretion to the Tatricum and the rear bivergently structured subautochthonous Fatricum in the Eocene orogenic wedge.

**Supplementary Materials:** The following are available online at https://www.mdpi.com/article/10.3390/min11090988/s1. Table S1: Bulk-rock composition of PI-B3, HEL-1-1 and TR-35 in oxide weight percentages. Table S2: The $^{40}$Ar/$^{39}$Ar geochronology results for samples ZI-3, SRB-1, HC-12. Table S3: Representative analyses of chlorites and white micas from the Považský Inovec Mts. Selec Block. Table S4: Representative analyses of chlorites and white micas of the Považský Inovec Mts. Hlohovec Block. Table S5: Representative analyses of chlorites and white micas from the Tribeč Mts. Zobor Nappe. Table S6: Representative analyses of chlorites and white micas of the Tribeč Mts. Razdiel Nappe. Table S7: Representative analyses of chlorites and white micas from the Fatricum basement and cover rocks in the eastern Nízke Tatry Mts. Vápenica Nappe. Table S8: Representative analyses of chlorites and white micas from subautochthonous southern Fatricum Krakľová Nappe in the eastern Nízke Tatry Mts.

**Author Contributions:** Conceptualisation, M.P., O.N. and M.D.; field investigation, M.P., P.R. and A.M.; methodology, M.P., O.N. and F.J.; software, O.N. and F.J.; validation, Č.T. and J.S.; formal analysis, M.P. and M.D.; resources, M.P., P.R., Č.T., J.S. and A.M.; data curation, M.P., O.N. and M.D.; writing—original draft preparation, M.P., O.N. and F.J.; writing—review and editing, M.P., O.N., F.J., M.D., J.S., Č.T., P.R. and A.M.; supervision and project administration, M.P.; funding acquisition, M.P. and F.J. All authors have read and agreed to the published version of the manuscript.

**Funding:** This research was funded by the Slovak Research and Development Agency (contracts APVV-15-0050, APVV-19-0065) and VEGA Agency (No. 1/0151/19, No. 2/0013/20).

**Data Availability Statement:** Not applicable.

**Acknowledgments:** Constructive reviews of two reviewers are greatly appreciated. We thank R. Marshall for reviewing the English content. We would like to express our special thanks to O. Vidal for his help with Chl–Ph–Qtz–H2O multi-equilibria calculations. M.D. was supported by Australian Research Council Discovery funding scheme (DP160102427) and Curtin Research Fellowship.

**Conflicts of Interest:** The authors declare no conflict of interest. The funders had no role in the design of the study; in the collection, analyses, or interpretation of data; in the writing of the manuscript, or in the decision to publish the results.

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
