# Peer review of "Formation of a Composite Albian–Eocene Orogenic Wedge in the Inner Western Carpathians: P–T Estimates and 40Ar/39Ar Geochronology from Structural Units"

_minerals, doi:10.3390/min11090988_

Round 1
Reviewer 1 Report
Overall, this is a thoughtful and data-rich manuscript. However, the text is very long, the terminology is complex, and it lacks a summary figure comparing the metamorphic P-T results for the nappes. This results in a difficult manuscript to read.
1 A summary P-T-t diagram with uncertainties is needed for a more direct comparison of the data from each nappe.
2 My compilation and understanding of the P-T history of each nappe was hindered by the dense terminology for the sedimentological packages, structural blocks, and nappes. The manuscript would benefit from simplified terminology.
Presumably the P-T data are only a small part of the data used for constructing the tectonic history and perhaps my compilation is wrong. But, I extracted the following:
Inovec Nappe: .4 to .9 GPa / 4-6 kbar 290-320 C
Hlohovec Nappe; .3-.8 GPa / 5-8 kbar 270-280 C
Razdiel Block .25-.75 GPa / 6-7 kbar 380-400 C
Nízke Tatry Mts. Vápenica Nappe .4-.8 GPa / 4-5 kbar 400-500 C
Fatricum Krakľová Nappe .5 GPa / 5-8 kbar 290-395 C
All of the pressures above overlap in range without considering uncertainty. So, what does this indicate about the structure? Is it possible that the mineral assemblages all achieved peak T [with most recrystallization] at essentially the same P? Alternately, is it possible that the range in P estimates for each nappe is meaningful?
3 The authors make an attempt to set up a comparison of ‘conventional’ and ‘new’ thermobarometers. The data for the two techniques are presented and there is considerable overlap and variation in the results, but no comprehensive discussion of these is provided and the paper is focused on tectonics. A complete comparison would be a separate manuscript; therefore, I recommend deleting all reference to the phengite barometer and instead using only the Chl-Ph-Qtz- water barometer. Finally, the conclusions rely on the P estimates from Chl-Ph univariant curve intersections. If the Si in phengite barometry are going to be ignored then there is not need to present them.
4 Figure 1 would benefit from additional info to indicate the exact location of the main map on inset figure.
5 Figs. 2, 3, 4, 5, 6, and 7 need lat and longitude.
6 Ar age presentation needs modification to reduce inferred precision. Also, make sure that the percent of Ar used for ages is at least 60% of total Ar released
7 Many of the table and figures are mislabeled with incorrect units for P and tables state that the units are different for the magnitude and the uncertainties. Tables and text should be modified to use the same units of magnitude throughout. For example, all P should be in MPa or kbar.
8 The location of the figure 10 cross section should be shown on a map.

Author Response
Please, see attachment

Reviewer 2 Report
The manuscript by Putis et al. is a great compilation of the existing evidence and a significant portion of new structural, thermobarometric and geochronological data for the Inner Western Carpathians, which notably extend the understanding of Alpine accretionary tectonics in the area. The manuscript is exceptionally well prepared and organized, written in a excellent English, has all necessary supplementary materials, which justify the conclusions well enough. Overall, I see the manuscript ready for publication, and I have only three remarks for the content, which might aid the improvement process.
- The authors should double-check through the text, figure captions etc. for minor grammatical and spelling errors, which are present sometimes.
- I would suggest reducing the manuscript size, which is quite huge and consistent more with a review type of papers more than with that of a regular article. However, this may be optional and should be decided by the associated editor based on the journal rules and limitations.
Author Response
Please, see attachment

Round 2
Reviewer 1 Report
The revised manuscript is much improved and the additional summary figure (58) is useful. However, this figure tends to confirm that there are only very small differences in the estimated pressures for most of the nappes. Only JS-5 and PL88 have significantly lower pressure estimates and these are for the D4 event.
Author Response
Please, see the attachment,
